# A modified fluctuation-test framework characterizes the population dynamics and mutation rate of colorectal cancer persister cells

Mariangela Russo [1,2,5], Simone Pompei [3,5], Alberto Sogari [1,2,5], Mattia Corigliano [3,4,5], Giovanni Crisafulli [1,2], Alberto Puliafito[1,2], Simona Lamba[2], Jessica Erriquez[2], Andrea Bertotti[1,2], Marco Gherardi [3,4], Federica Di Nicolantonio [1,2], Alberto Bardelli [1,2,6 ✉] and Marco Cosentino Lagomarsino [3,4,6 ✉]

Compelling evidence shows that cancer persister cells represent a major limit to the long-term efficacy of targeted therapies. However, the phenotype and population dynamics of cancer persister cells remain unclear. We developed a quantitative framework to study persisters by combining experimental characterization and mathematical modeling. We found that, in colorectal cancer, a fraction of persisters slowly replicates. Clinically approved targeted therapies induce a switch to drug-tolerant persisters and a temporary 7- to 50-fold increase of their mutation rate, thus increasing the number of persister-derived resistant cells. These findings reveal that treatment may influence persistence and mutability in cancer cells and pinpoint inhibition of error-prone DNA polymerases as a strategy to restrict tumor recurrence.

When patients with cancer are treated with targeted agents, tumor relapse is often observed after an initial response[1,2]. Emergence of resistance after prolonged response and disease stabilization is also frequent[3,4]. Indeed, when cancer cells are exposed to lethal doses of targeted therapies, the emergence of a subpopulation of drug-tolerant 'persister' cells prevents tumor eradication[5–10]. Unlike genetically resistant cells, persisters tolerate drug pressure through reversible, nongenetic, noninheritable mechanisms of resistance[5,10,11]. However, it is unclear whether persisters enter a quiescent state or slowly progress through the cell cycle. It is also unknown whether the persister phenotype is drug induced or pre-exists. In addition, the population dynamics governing persister evolution to resistance are only partially elucidated[8].

Exposure of colorectal cancer (CRC) cells to targeted therapies induces DNA damage and impairment of DNA repair proficiency, a phenotype recently confirmed in multiple studies[12,13]. Drug treatment leads to error-prone DNA replication in cancer cells, suggesting that mutability of persisters could increase during therapy-induced stress[14].

Measuring mutational processes by DNA sequencing is challenging, owing to tumor heterogeneity and the difficulties of tracking lineages[12]. A complementary strategy is represented by the 'fluctuation test' developed by Luria and Delbrück to characterize the onset of resistance in bacterial populations[15]. This assay exploits multiple replicates of clonal populations to bypass lineage-tracking issues and provides an elegant strategy to estimate mutation rates.

The fluctuation test has been previously modified to infer the acquisition of resistance to therapy in tumors[16–19], particularly for the evolution of pre-existing resistant cells before treatment initiation and estimation of cancer cells' mutation rate in basal conditions[20].

However, it is not designed to quantify mutation rates in cancer persisters during drug treatment.

In the present study, we present a general quantitative methodology to characterize the transition of cancer cells to persisters and measure their population dynamics during drug treatment. We deployed a two-step fluctuation test to quantify phenotypic mutation rates of CRC cells. Importantly, our assay discriminates pre-existing resistant clones from persister-derived ones, allowing a quantification of spontaneous (that is, in untreated conditions) and drug-induced mutation rates.

## Results

**Growth dynamics of CRC cells.** We first aimed to quantify how CRC growth dynamics parameters were affected by drug treatment. Our experiments included: (1) growth rates in standard conditions, (2) population dynamics under treatment and (3) population dynamics of persisters (Supplementary Note).

We used two microsatellite-stable (MSS) CRC cell models, *RAS/RAF* wild-type DiFi and *BRAF* V600E-mutated WiDr, which are respectively sensitive to the anti-epithelial growth factor receptor (EGFR) antibody cetuximab alone[13,21] or in combination with a BRAF inhibitor[13], two clinically approved regimens for CRC[22–24]. To reduce the possibility that pre-existing resistant cells were present in the populations at the start of the assay, we isolated individual clones for each cell model, with growth and drug-sensitivity profiles comparable to those of the parental population (Supplementary Fig. 1a,b).

We measured birth and death rates of clones in standard cell-culture conditions (Supplementary Fig. 2, Supplementary Table 1 and Supplementary Note). We collected data from two

[1]Department of Oncology, University of Turin, Candiolo, Italy. [2]Candiolo Cancer Institute, FPO-IRCCS, Candiolo, Italy. [3]IFOM Foundation, FIRC Institute of Molecular Oncology, Milan, Italy. [4]Department of Physics, University of Milan and INFN, Milan, Italy. [5]These authors contributed equally: Mariangela Russo, Simone Pompei, Alberto Sogari, Mattia Corigliano. [6]These authors jointly supervised this work: Alberto Bardelli, Marco Cosentino Lagomarsino. ✉e-mail: alberto.bardelli@unito.it; marco.cosentino-lagomarsino@ifom.eu

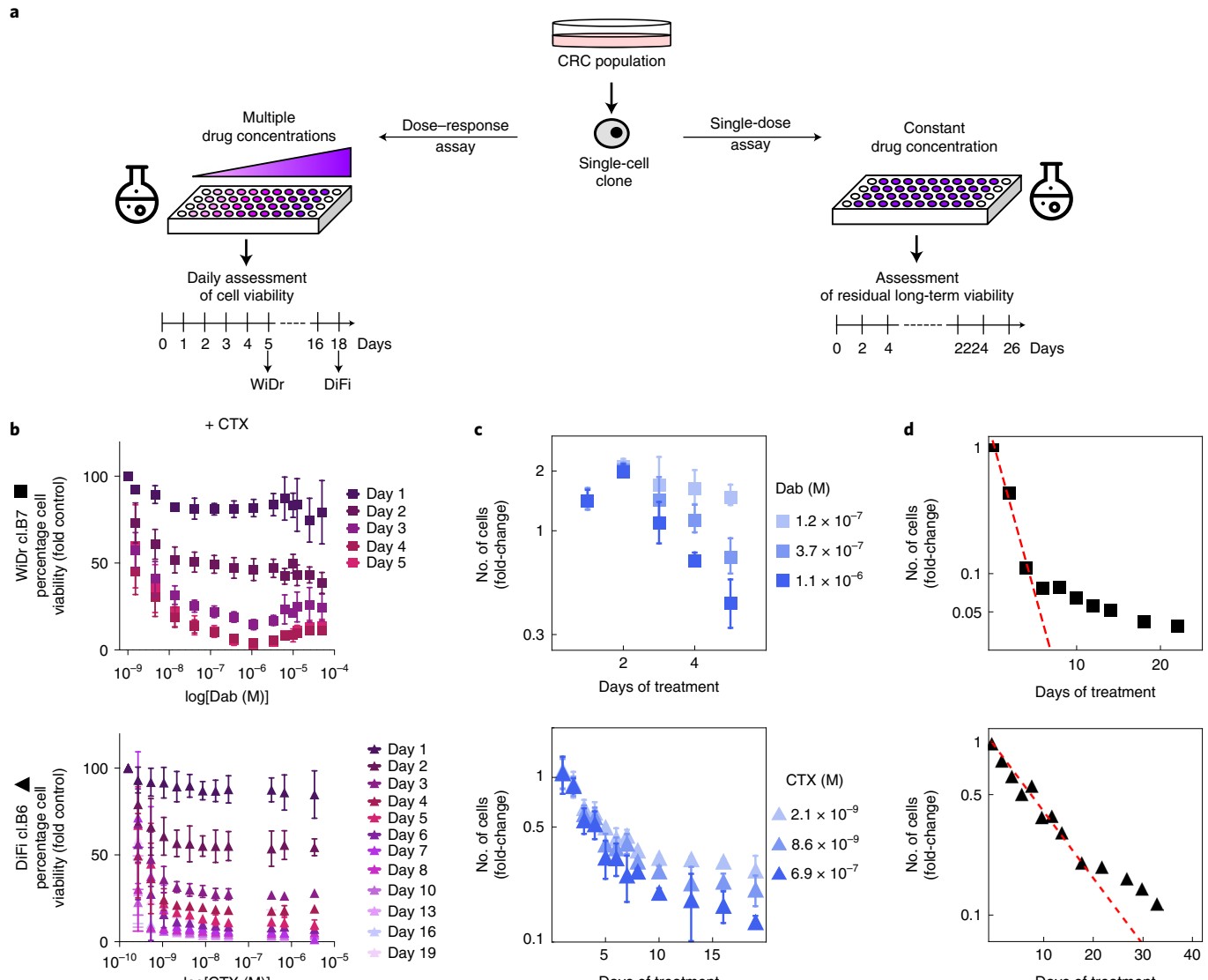

**Fig. 1 | Population dynamics of CRC clones in response to targeted therapies. a**, Schematic representation of the drug-screening growth curve assays performed on CRC clones. **b**, In the dose–response assay, WiDr cells were treated with increasing concentrations of dabrafenib (Dab) + 50 μg ml⁻¹ of cetuximab (CTX), whereas DiFi cells received increasing concentrations of CTX. Cell viability was measured by the ATP assay at the indicated timepoints. Results represent the average ± s.d. ($n = 3$ biologically independent experiments for WiDr; $n = 5$ biologically independent experiments for DiFi). **c**, Growth curves of the indicated cells under treatment, reported as fold-change of viable cells (log scale) versus time of drug exposure, calculated from dose–response assay data, by normalizing cell viability at the indicated timepoints using the viability measured at day 0. Growth curves for three different drug concentrations for each clone are shown as the average ± s.d. ($n = 3$ for WiDr; $n = 5$ for DiFi). **d**, Fold-change of viable cells (log scale, assessed by ATP assay) versus time of drug exposure for indicated cells in the single-dose assay. The total number of viable cells is compatible with an exponential decay with two time scales, supporting the outgrowth of persisters (the dashed line indicates the initial slope). Symbols indicate means ($n = 2$ biologically independent experiments).

sets of drug–response growth assays (Fig. 1a). The first, the dose–response assay in which CRC clones were exposed to increasing concentrations of targeted therapies (Fig. 1b and Extended Data Fig. 1a), was used to analyze growth curves defined as the number of live cells versus time and drug concentration (Fig. 1c and Extended Data Fig. 1b) and to quantify the intertwined processes of growth and transition to persister state on drug treatment. Supplementary Fig. 3 shows the normalization process of the dose–response assay data used to obtain growth curves (Supplementary Note). The second, the single-dose assay, consisting of 3 weeks of exposure to a constant drug concentration, highlighted a biphasic (two time-scale) killing curve (Fig. 1d and Extended Data Fig. 1c), characterized by a rapid decline of sensitive cells followed by a slower decline[25,26],

a hallmark of the emergence of persisters in bacteria[25]. The fraction of surviving persisters displayed a slow but measurable decay in cell number (Extended Data Fig. 2), suggesting a tendency for persisters to slowly die over time.

**CRC persister cells slowly replicate during drug treatment.** We next sought to elucidate the dynamics of persister proliferation and death under drug treatment. Staining of CRC persisters with carboxyfluorescein succinimidyl ester (CFSE), a cell-permeable fluorescent dye allowing flow-cytometric monitoring of cell divisions[27], and 5-ethynyl-2′-deoxyuridine (EdU), a modified thymidine analog that is efficiently incorporated into DNA during active DNA replication, revealed that a fraction between 0.2% and 2.5% of

persisters slowly replicates during treatment (Extended Data Fig. 3a–c and Supplementary Note), in line with recent data[28]. We next used a live cell microscopy imaging assay (Supplementary Note). Although most CRC persisters were nonreplicating, cell division events were visible in all the CRC clones analyzed (Extended Data Fig. 3d and Supplementary Videos 1–4). The cell division was occasionally successful and viable (Extended Data Fig. 3d). We also detected cell death events after cell division and in nondividing cells (Extended Data Fig. 3e,f).

**The persister phenotype is induced by targeted therapies in CRC.** To quantify cell population dynamics during drug treatment, we developed a mathematical model of the transition of CRC cells to the persister state, which we term the 'transition-to-persister' (TP) model. This model incorporates birth–death parameters and phenotypic switching in the deterministic limit (that is, neglecting fluctuations due to stochastic demographic effects, see Supplementary Note)[29,30]. Figure 2a summarizes the TP model dynamics. We exploited the model to quantify the transition rate and assess whether a subpopulation of persisters predated drug administration or whether the persister phenotype emerged on drug treatment. The TP model considers three possible fates for drug-treated cells: (1) death, (2) replication and (3) switching to persister state at a rate, $\lambda$, in the presence of the drug; it further considers the pre-existence of an arbitrary steady fraction $f_0$ of persisters (Fig. 2a).

The following equations define the dynamics of sensitive ($X(t)$) and persister ($Z(t)$) cells according to the TP model (under drug treatment):

$$\begin{cases} \frac{d}{dt}X(t) = (B - D([M]))\,X(t) - \lambda([M])\,X(t) \\ \frac{d}{dt}Z(t) = -D_P Z(t) + \lambda([M])\,X(t) \end{cases} \quad (1)$$

where $B$ and $D([M])$ are, respectively, the birth- and drug-dependent death rates of sensitive cells, whereas $[M]$ is the drug concentration. Persisters emerge with a drug-dependent transition rate $\lambda([M])$. Under drug treatment, persisters die with a rate $D_P > 0$ (Extended Data Figs. 2 and 3). The model assumes that persisters that attempt to divide before acquiring drug-resistance mutations die; therefore a possible back-switching from persister to sensitive in the presence of drug would effectively contribute to the death rate (Supplementary Note).

The initial condition that specifies the solution to equation (1) is key for quantifying to what extent the transition to persister state is induced by the drug treatment. Specifically, if the sensitive-to-persister transition is fully drug induced, then untreated populations would not contain any persisters, that is, $f_0 = \frac{Z(t_0)}{N(t_0)} = 0$.

Conversely, if some persisters pre-exist, then the initial fraction of persister cells has a finite positive value ($f_0 > 0$). If $f_0$ is very small, some persisters may pre-exist, but the transition is mainly drug induced. If $f_0$ is actually comparable to the fraction of residual persisters after weeks of treatment, then the transition to persistence is not drug induced.

To determine the most likely scenario, we used experimental data collected from drug–response assays (Fig. 1). Using results from the dose–response assay, we defined parameters governing the dynamics of the model over a short timeframe, such as the initial fraction of persisters $f_0$ and the effective growth rate of treated cells. Similarly, the single-dose assay was used to quantify model parameters that affect long-term dynamics, such as the transition rate of sensitive to persister cells ($\lambda$) and the effective death rate of persisters ($D_P$). By constraining model parameters from experimental data, we established which scenario would best describe the cell-based results. The inferred parameters are compatible with the values obtained by live cell microscopy assay, supporting a balance

between proliferation and cell death skewed slightly toward the latter (Extended Data Fig. 3g and Supplementary Note).

On treatment, the number of cells started to decline within 1–3 d ($t_0$), depending on the initial seeding density (Supplementary Fig. 4). The observed cell dynamics were coherent in experiments with different seeding densities once the growth curves had been scaled (both in time and in measured viability) to the maximum value reached at $t = t_0$ (Supplementary Fig. 4).

The parameters of the TP model were inferred with a standard Bayesian inference framework for both cell lines (Supplementary Table 2 and Supplementary Note). DiFi displayed slower 'dying' dynamics compared with WiDr. In light of this, in WiDr we performed a joint fit of both the dose–response and the single-dose datasets, whereas in DiFi we assessed growth curves in response to multiple doses of targeted therapies for up to 19 d, which allowed performing a model fit based on the dose–response dataset only (Supplementary Note).

We identified the best-fit TP model parameters given the experimental data, considering different values of the initial number of persisters ($f_0$). The best fit between the inferred TP model and experimental data occurs when $f_0 = 0$, whereas the concordance decreases when $f_0$ increases; we note that a value of $f_0$ of 10% already leads to substantial deviations from the data (Fig. 2b). Therefore, the TP model is consistent with the persister phenotype being predominantly drug induced. In addition, the model properly describes the dynamics of the single-dose assay (Fig. 2c).

To further confirm the validity of the TP model, we next focused on the Bayesian statistics of the two model parameters describing the dynamics of persisters: the transition rate $\lambda$ and initial fraction of persisters $f_0$. The joint posterior distribution of the Bayesian inference of these two parameters is shown in Fig. 2d and Supplementary Fig. 5. We found that the transition rate to persistence $\lambda$ estimated by the model fit does not vary when considering different values of the initial fraction of persisters $f_0$ (Fig. 2d). The marginalized posterior probability of $f_0$ peaked at zero (Fig. 2d, bottom panel), and its upper boundary is much smaller than the ratio between the persister population size (after all persister cells have emerged) and the total population size at the start of treatment. This implies that the inferred value of transition rate $\lambda$ is independent from $f_0$ and the best concordance of the TP model to the experimental data is obtained for $f_0 = 0$.

Finally, to compare the scenarios $f_0 = 0$ and $f_0 > 0$, we used the Bayesian information criterion (BIC) and Akaike's information criterion (AIC), which are standard Bayesian criteria used for model selection. According to both, the TP model with $f_0 = 0$ is preferred over $f_0 > 0$ (Extended Data Fig. 4, Supplementary Table 3 and Supplementary Note). We summarize the best-fit TP model parameters in Supplementary Table 4. It is interesting that we found the transition rate of WiDr and DiFi cells to persisters to be drug dependent (Extended Data Fig. 4 and Supplementary Fig. 6). These results indicate that, even if few persisters exist in the population before drug treatment, most of them must have transitioned to the persister phenotype after drug exposure. Our finding that WiDr cells show a transition rate to persistence that increases with drug concentration could be applied to design innovative strategies to restrict persister evolution. Notably, our analysis predicts that a linear increase of drug concentration, compared with a constant dosage, might reduce the number of persisters (Extended Data Fig. 5).

**Persisters distribution supports a drug-induced scenario.** We then measured how the number of persisters varied across multiple wells, because the distribution of this parameter is expected to be different between a drug-induced and a pre-existing scenario[31]. We seeded DiFi cl.B6 and WiDr cl.B7 in multiple 96-well plates and quantified the distribution of persisters (residual cell viability) among >400 independent wells after 3 weeks of drug treatment

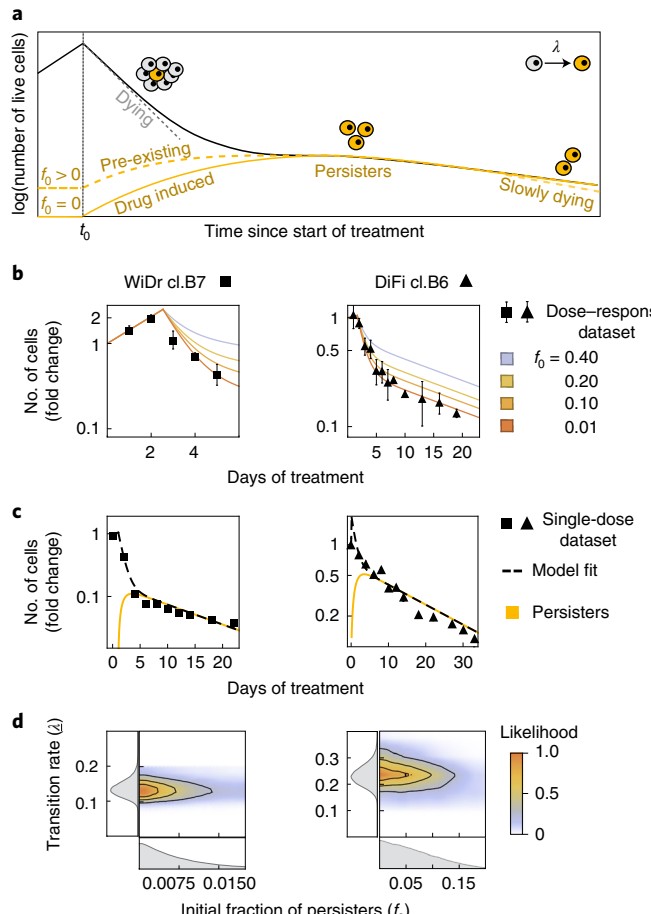

**Fig. 2 | Emergence of CRC persister cells is predominantly drug dependent. a**, Schematic representation of cell population dynamics under constant drug treatment. When cancer cells are exposed to targeted therapies, the number of viable cells starts to decline. A homogeneous population of sensitive cells (gray cells) would shrink exponentially to extinction (gray dashed line). Some cells survive drug treatment due to the transition to a persister phenotype (dark-yellow cells, dark-yellow lines) at a rate $\lambda$ and residual cells (solid black line) show a biphasic decay. A finite fraction of persister cells ($f_0$) might be present in the population before drug treatment ($f_0 > 0$, dashed dark-yellow line) or not ($f_0 = 0$, solid dark-yellow line). Persister cells show a reduced death rate during treatment, which results in a slow exponential decline of the cell population (dark-yellow dashed line). **b**, Growth curves of CRC clones under treatment, calculated from the dose–response assay. Black symbols and bars represent the average ± s.d. of the dose–response dataset ($n = 3$ biologically independent experiments for WiDr, $n = 5$ biologically independent experiments for DiFi). The continuous lines indicate the TP model fit to the experimental data for different values of the initial fraction of persister cells ($f_0$, color coded). **c**, Fold-change of viable cells versus time of drug exposure for the indicated cells, assessed based on the single-dose dataset. Black symbols represent averages of the experimental data ($n = 2$ biologically independent experiments). The black dashed line indicates the fit of the TP model to the data, whereas the expected fraction of persister cells is shown by the dark-yellow solid line. **d**, Joint posterior distribution (contour plot, color coded with the normalized likelihood function) and marginalized posterior distributions (left and bottom panel, gray area indicates the probability density function) of TP parameters describing the dynamics of persister cells: (1) the initial fraction of persister cells ($f_0$, bottom panel) and (2) transition rate of sensitive to persister cells ($\lambda$) induced by the drug treatment. The likelihood function measures the agreement of the model with the experimental data as a function of the value of the parameters considered.

(Extended Data Fig. 6a). The observed abundance distribution across wells was consistent with a Poisson distribution (Extended Data Fig. 6b), supporting a drug-induced scenario, as confirmed by computational simulations (see ref. [31], where a similar method was used for mutational processes, and Extended Data Fig. 6c). These numerical simulations show that, provided that persisters do not exist before drug treatment, the number of persisters emerging from sensitive cells under treatment is Poisson distributed. Conversely, pre-existing persisters would be generated with a constant rate from an exponentially expanding population before treatment administration. Hence the number of pre-existing cells is not Poisson distributed, but is described by a Luria–Delbrück[15] distribution (with variance ≫ mean). We found that the final distribution of persisters across wells is a Poisson distribution, in line with emergence after drug treatment.

**A fluctuation assay quantifies persisters' mutation rates.** Measurement of mutation rates in the absence or presence of anti-cancer drugs required the development of a second model, hereafter the 'mammalian cells–Luria–Delbrück' or 'MC–LD' model. The MC–LD model is a fully stochastic birth–death branching process, describing the growth of resistant cells before and during drug treatment (Fig. 3a). We designate with $\mu$ the effective rate at which one individual (cell) develops resistance whereas $\mu_s$ and $\mu_p$ indicate mutation rates of sensitive (untreated) and persister cells, respectively (Fig. 3b and Supplementary Note).

WiDr and DiFi cells were seeded in 20 96-multiwell plates each and allowed to grow for a fixed number of cell divisions in drug-free standard culture conditions (Fig. 3c); afterwards, a constant, clinically relevant drug concentration was applied (Fig. 3d). The number of wells, the initial population size in each well and the time of cell replication in the absence of drug treatment were set by theoretical considerations incorporating the population dynamics parameters that we previously measured (Supplementary Tables 1 and 4 and Supplementary Note).

In accordance with our previous work[6], a small number of early emerging resistant colonies was detected after 3–4 weeks of treatment (Fig. 3d,e). Conversely, in the vast majority of the wells sensitive cells died, whereas drug-tolerant persisters survived, as detected by measurement of residual cell viability (Extended Data Fig. 6)[6]. After several weeks of constant treatment of the residual persister cells (Fig. 3d), late-emerging resistant colonies appeared in a subset of wells in which persisters had previously been detected (Fig. 3d,e).

We ran multiple MC–LD model simulations, with input parameters inferred with the TP model, and found that resistant clones emerging at late time points (>4 weeks of treatment) are unlikely to originate from pre-existing resistant cells (Fig. 4a and Extended Data Fig. 7). In accordance with previous work[6], we considered the resistant colonies that became microscopically visible within the first 4 weeks of drug treatment (early emerging resistant) as those representing pre-existing resistant cells, that is, mutant cells that emerged during the expansion phase by spontaneous mutation. We also reasoned that resistant colonies that slowly emerged after ≥10 weeks of drug treatment (late-emerging resistance) in persister-containing wells could have developed drug-resistance mutations through the adaptive mutability process that we and others have observed[12,13] (Fig. 4a and Extended Data Fig. 7).

As in a standard fluctuation test, the mutation rate can be inferred from the observed fraction of wells containing resistant cells. In the model, this fraction corresponds to the expected probability of observing a resistant clone in a well in a given time interval $[0, T]$. To compute this probability in the MC–LD model, we assumed that resistant cells divide with rate $b$ and die with rate $d$, just like untreated cells. Supplementary Fig. 7 supports the stability of the inferred values of the mutation rates against variation of the division rate of resistant cells. As a result of reproductive

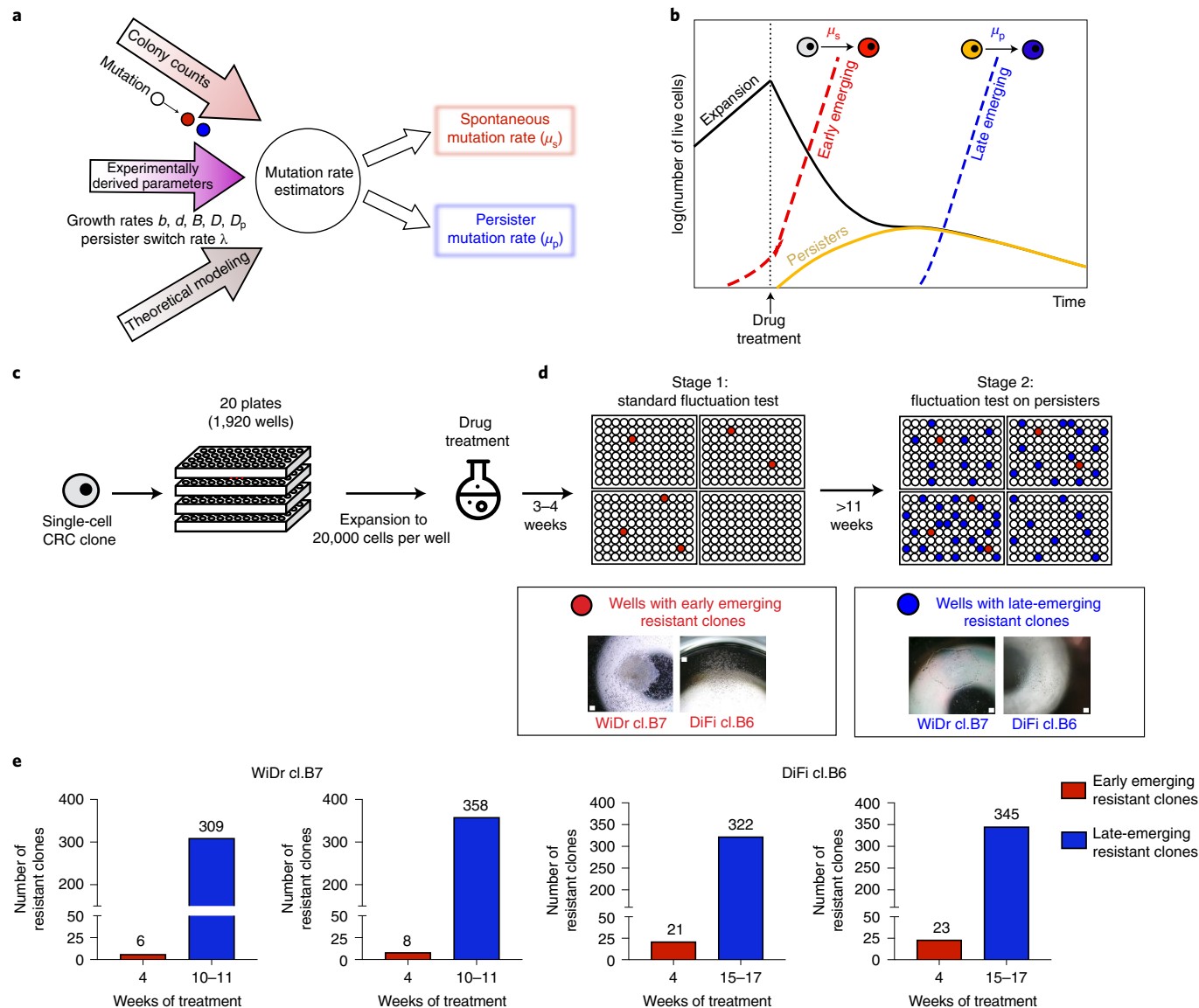

**Fig. 3 | A modified Luria–Delbrück fluctuation test to measure mutation rates in mammalian cells. a**, The modified fluctuation test, based on the inferred population dynamics and the MC–LD model, allows estimation of spontaneous ($\mu_s$) and persister ($\mu_p$) mutation rates. **b**, Schematic representation of cell population dynamics of CRC cells during the fluctuation test. During the initial expansion in the absence of drug treatment, CRC cells mutate with the spontaneous mutation rate ($\mu_s$). When cells are exposed to targeted therapies, pre-existing resistant cells are selected by the drug and give rise to early emerging resistant colonies (red dashed line), whereas sensitive cells start to decline in number (black solid line) and switch to the persister state (dark-yellow solid line). Resistant cells derive from persisters with a mutation rate $\mu_p$ and give rise to late-emerging resistant colonies (blue dashed line). **c**, Schematic representation of the experimental design underlying the fluctuation assay. WiDr and DiFi cells were seeded in 20 96-multiwell plates, for a total of 1,920 wells, and allowed to expand in the absence of drug for about 8 generations (reaching ~20,000 cells per well). After the expansion, all the wells were treated with targeted therapy (100 µg ml⁻¹ of cetuximab for DiFi and 1 µM dabrafenib + 50 µg ml⁻¹ of cetuximab for WiDr). **d**, Two sets of resistant clones were identified during the MC–LD experimental assay: the early emerging resistant clones grown after 3–4 weeks (stage 1) and the late-emerging resistant clones arising after >10 weeks (stage 2) of constant drug treatment. Scale bar, 100 µm. **e**, Bar graphs listing the number of resistant clones counted at the indicated timepoints during the MC–LD experiment for each CRC clone. Red bars indicate early emerging resistant clones (appearing in the first 4 weeks of drug treatment); blue bars indicate late-emerging resistant clones (appearing after ≥10 weeks of drug treatment). Results of two independent biological replicates for each clone are shown.

fluctuations (genetic drift), cells carrying drug-resistance mutations can still become extinct, and only a fraction of the mutants, which we refer to as 'established mutants', survive stochastic drift. The probability of surviving stochastic drift in a time interval $\Delta t$, denoted here as $\psi(\Delta t)$, is a well-known result of the birth–death process[32,33] (Supplementary Note).

We derived analytically an approximate solution of the model, by considering that the number of mutant cells established in the time interval $[0, T]$ follows a Poisson distribution with an expected value $\mathcal{M}(T)$. Consequently, the probability of having at least one mutant is given by:

$$P(T) = 1 - e^{-\mathcal{M}(T)}. \tag{2}$$

To quantify the spontaneous mutation rate of cancer cells before drug administration, we focused on the resistant cells established by

the time $T_{\text{treat}}$ before treatment administration. The expected number of resistant cells that emerged from sensitive cells in this time interval reads:

$$\mathcal{M}_{\text{sensitive}}\left(T_{\text{treat}}\right) = \mu_{\text{s}} \int_0^{T_{\text{treat}}} X\left(t\right) \psi\left(T_{\text{treat}} - t\right) \mathrm{d}t, \qquad (3)$$

where $\mu_{\text{s}}$ is the mutation rate of sensitive cells.

To quantify the mutation rate of persister cells $\mu_{\text{p}}$, we consider resistant cells that emerged by the time $T$ since the beginning of the drug treatment. The expected number of resistant cells that emerged from persister cells reads:

$$\mathcal{M}_{\text{persisters}}\left(T\right) = \mu_{\text{p}} \int_0^T Z\left(t\right) \psi\left(T - t\right) \mathrm{d}t. \qquad (4)$$

We emphasize that equations (2)–(4) are connected to the solution of the TP model, equation (1). Hence, the solution of the MC–LD model is defined in terms of the same parameters that were estimated with the TP model (Supplementary Note).

We used this solution of the MC–LD model to derive estimators of mutation rates of sensitive cells $\mu_{\text{s}}$ (encompassing the fraction of wells with early emerging resistant cells) and of persister cells $\mu_{\text{p}}$ (corresponding to the fraction of wells with late-emerging resistant clones).

**Persisters show an increased mutation rate under treatment.** Data collected with two-step MC–LD fluctuation tests for each clone allowed inferring mutation rates of sensitive ($\mu_{\text{s}}$) and persister ($\mu_{\text{p}}$) cells. We conservatively evaluated the mutational processes as chronological (measured in mutations per day) rather than replicative (mutations per generation). This choice is safe, as the ratio between replicative mutation rates of cells displaying the two phenotypes must always be higher than for chronological rates, because (beyond any uncertainty) measured cell division in persister cells was very low compared with that of untreated cells.

We found that mutation rates were increased by a factor of 7- to 50-fold in cells that survived and tolerated for several weeks doses of targeted therapies that were lethal for most of the parental population (Fig. 4b and Supplementary Table 5). This result was consistent across multiple biological replicates, both in DiFi and WiDr cells and in response to clinically relevant concentrations of either EGFR blockade or EGFR/BRAF concomitant inhibition, respectively (Fig. 4b and Supplementary Table 5).

To further validate the consistency of the mutation rate inference based on the MC–LD model, we ran multiple simulated replicates of the experiment, using a set of sensitive ($\mu_{\text{s}}$) and persister ($\mu_{\text{p}}$) mutation rates, and we then used the MC–LD estimators on the synthetic data. Figure 4c compares boxplots of the estimated mutation rates across replicates of simulated experiments, with the actual values of mutation rates used as inputs to the simulations. The agreement between these values validates our estimates.

We next assessed whether and to what extent the inferred value of the mutation rate is affected by the presence of different numbers of pre-existing persister cells $f_0$ using our estimators within a Bayesian framework (Fig. 4d). This approach returns the mutation rate, considering a range of realistic values of $f_0$ and their probability. We obtained the fold increase of the mutation rate of persister cells as a function of $f_0$, in the entire range of values that are compatible with the dynamics observed in the growth curve assays experimentally assessed in Figs. 1 and 2. Figure 4d summarizes the results of this inference. We found that, considering all representative values of $f_0$ that are compatible with our experimental data, the increase of mutation rate in persister cells remains strongly supported.

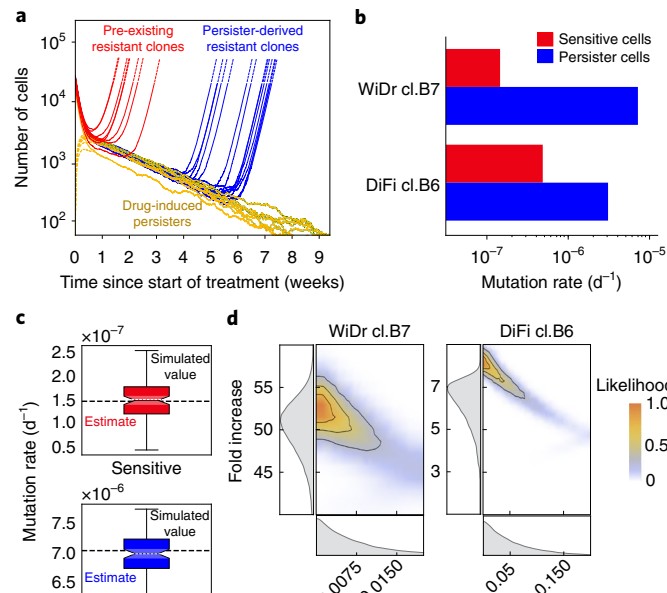

**Fig. 4 | Quantification of mutation rates in persister cells. a**, Simulated data for the assay described in Fig. 3. The experimentally measured MC–LD model parameters and the model-derived estimators of mutation rate, for sensitive and persister cells (dark-yellow solid lines), were used to simulate the time of appearance of pre-existing (red) and persister-derived (blue) resistant cells. **b**, Quantification of mutation rates for sensitive (red) and persister (blue) cells in the MC–LD experiment. The indicated cell models were seeded and treated as described in Fig. 3. Mutation rates were calculated from the experimental data, based on population parameters and the number of pre-existing (early emerging) and persister-derived (late-emerging) resistant clones as described in Fig. 3. Results represent inferred mutation rates (for each clone of sensitive and persister cells) with bar plots showing the mean of the posterior distributions of the mutation rates ($n = 2$ biologically independent experiments). Here, the bar chart is used as a graphic representation of inferred mutation rates (see Supplementary Table 5 for the corresponding numerical values). **c**, Validation of mutation rates estimator with model simulations. The boxplots represent the distribution of the estimated mutation rates for $n = 100$ independent simulations of the entire experiment using the parameters reported in Supplementary Tables 1 and 4. Red and blue boxes indicate the interquartile ranges (25th and 75th percentiles) of the estimated mutation rates of sensitive and persisters, respectively, whereas the upper and lower whiskers represent the maximum and minimum values of the distribution. The median of the distribution, reported as a black line in the center, is shown together with its 95% confidence interval (nuanced area), indicated by the notches on both sides of the box. The mean of the distribution and the input value of the mutation rate used in the simulation are reported as a dashed white line and a black dashed line, respectively. **d**, Joint posterior distribution (contour plot, color coded with the normalized likelihood function) and marginalized posterior distributions (left and bottom panel, gray area shows the probability density function) of (1) the initial fraction of persister cells ($f_0$, bottom panel) and (2) fold increase of the mutation rate of persister cells compared with mutation rate of sensitive cells ($\mu_{\text{p}}/\mu_{\text{s}}$). The likelihood function measures the agreement of the model to the experimental data as a function of the value of the parameters considered.

To corroborate these results we replicated the full set of experiments and ran the analysis pipeline for two additional clones, one for each cell model (WiDr cl.B5 and DiFi cl.B3), thereby confirm-

ing our findings and excluding a clonal bias effect (Extended Data Figs. 8 and 9). Molecular profiling of persister-derived resistant clones isolated from the fluctuation assays revealed acquisition of single-nucleotide variations (SNVs) or copy-number alterations (CNAs) in genes involved in the RAS-MEK pathway, which are known drivers of resistance to anti-EGFR/anti-BRAF inhibitors in CRC[21,34,35] (Supplementary Fig. 8 and Supplementary Table 6).

We propose a quantitative model for the evolutionary dynamics of CRC cells exposed to targeted therapies (Fig. 5a). Untreated cancer cells replicate and spontaneously acquire mutations that can confer resistance to targeted therapies (pre-existing resistant mutations) at a replicative mutation rate $\mu_s$. However, when cells are exposed to targeted therapies, most of them quickly die, whereas a subset of parental cells switch to a long-lasting surviving persister state at a rate $\lambda$ and in a drug-induced fashion. Previous and current findings indicate that persister cells, under constant exposure to lethal doses of drugs, initiate a stress response that affects DNA replication fidelity[12,13], thus leading to a measurable increase of their mutation rate ($\mu_p$), therefore raising the probability that alterations conferring drug resistance could occur.

**Inhibition of mutagenic REV1 extends the efficacy of targeted therapy.** We previously reported that, in response to drug treatment, cancer cells switch from high- to low-fidelity DNA replication through downregulation of DNA repair genes and upregulation of specialized error-prone DNA polymerases[13]. This, in turn, could foster the temporary increase of the mutation rate observed in persister cells as quantitatively measured in the present study. Among the DNA polymerases that are upregulated in cancer cells on targeted therapy[13], REV1 carries out translesion synthesis (TLS), a mutagenic process that allows cells to tolerate DNA damage by bypassing lesions that block normal DNA replication, resulting in the introduction of mutations[36,37]. Interfering with TLS using a REV1 inhibitor has been shown to enhance chemotherapy efficacy and suppress tumor growth both in vitro and in vivo[38,39]. Based on the above, we hypothesized that inhibition of mutagenic TLS would probably increase the cytotoxic effects of targeted therapy-induced DNA damage, therefore delaying the acquisition of resistance during adaptive mutability.

To assess this possibility, we performed a time-to-progression (TTP) assay, an approach that we previously established[40] to monitor the development of secondary resistance in cancer cells. DiFi and WiDr CRC cells, as well as the *BRAF* V600E-mutated cell line (JVE207), were treated with a MAPK (mitogen-activated protein kinase) pathway inhibitor, the REV1 inhibitor or their combination. Pharmacological blockade of REV1 remarkably delayed or prevented the development of secondary resistance to EGFR/BRAF inhibitors (Fig. 5b and Extended Data Fig. 10).

## Discussion

We present an experimental framework that integrates biological assays and mathematical modeling to investigate population dynamics of cell lines exposed to environmental perturbations. The MC–LD assay allows quantitative comparisons of spontaneous and drug-induced mutation rates and could, in principle, be applied to measure whether and how a wide range of environmental conditions affects persister phenotype and mammalian cells' mutation rates in a considerable number of biological systems.

An important caveat of our technique is that fluctuation tests measure 'phenotypic' mutation rates, that is, rates of conversion to a phenotype (here, resistance to targeted therapies) that could result from different mutational routes, including SNVs and CNAs, both of which we found to drive resistance in our persister-derived resistant clones. Nevertheless, our approach has the advantage of bypassing several hurdles associated with sequencing-based measurement of mutation rates.

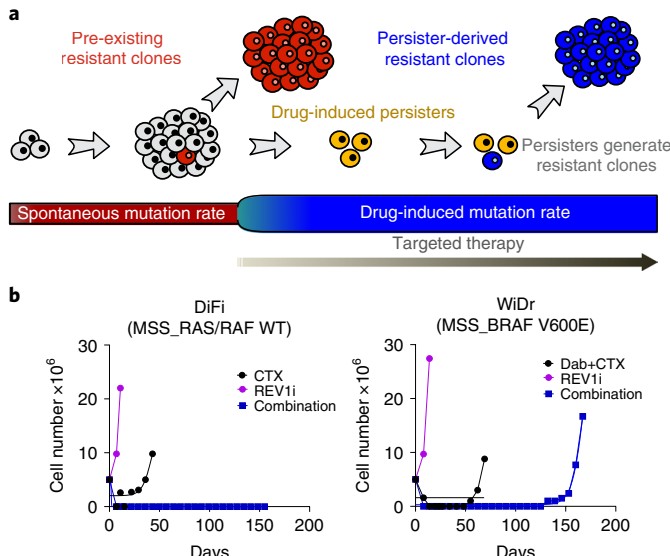

**Fig. 5 | Inhibition of error-prone DNA polymerases delays the onset of acquired resistance to targeted therapies. a**, Schematic representation of CRC cell mutational dynamics during drug treatment. Untreated cells spontaneously acquire resistant mutations at a replicative spontaneous mutation rate $\mu_s$. When cancer cells are exposed to targeted agents, a surviving persister phenotype is induced in a drug-dependent manner. Persister cells under constant drug exposure reduce DNA replication fidelity and increase their mutation rate at a rate $\mu_p$. This, in turn, boosts genetic diversity and favors the emergence of resistant clones driving tumor recurrence and treatment failure. **b**, The indicated CRC cells were treated with the anti-EGFR inhibitor cetuximab (CTX) alone or in combination with anti-BRAF inhibitor dabrafenib (Dab); the REV1 inhibitor (REV1i) was added where indicated. The number of cells was monitored during the treatment, until the emergence of resistance ($n = 1$ biological experiment for each cell line).

Although we and others have recently shown that adaptive mutability fosters the acquisition of secondary resistance by increasing genomic instability in surviving persister cells[12,13], the lack of models to quantitatively characterize the behavior of persisters under treatment has so far prevented reliable quantification of persisters' mutation rates. Although limited to cell lines, our controlled two-step fluctuation assay overcomes these issues. The results could be used to infer features of more complex systems (such as patient samples), where mathematical models are postulated and cannot be analogously validated.

Our results indicate that drug-induced sensitive-to-persister transition is a predominant path to the development of this phenotype. This is in line with recent evidence of a chemotherapy-induced persister state in CRC[41]. Although our results are coherently explained by the existence of a phenotypic switch of sensitive to persister cells, no direct observation of the switching is yet available. Alternative models whereby slower-proliferating tolerant cells generate faster-proliferating nontolerant phenotypes would also give rise to a biphasic killing curve[42,43]. In our framework, this would correspond to the case where $f_0 \gg 0$, which is ruled out by our analysis. Hence, the interpretation linking the phenotypic switch to persistence with treatment appears to be the most likely scenario.

Importantly, even a small subset of pre-existing persisters does not affect our findings that mutation rates of cancer cells remarkably increase under treatment. Persister-derived resistant clones keep emerging after several weeks of continuous drug treatment. In the absence of an increased mutation rate, it would typically take (in a conservative estimate) >100–1,000 weeks for the cells in a single

well to develop resistance based on the mutation rate of sensitive (untreated) cells. Equivalently a fluctuation assay performed on $2 \times 10^5$–$2 \times 10^6$ wells would be required to observe a few resistant clones after 10 weeks. In addition, the contribution of sensitive cells to resistance is exhausted after a few weeks of treatment, because we show that they become extinct within a few days (Fig. 1d). The evidence of active cell cycle progression, alongside an increase of mutagenic rate in persister cells, further supports previous findings of ongoing adaptive mutagenesis fostering acquisition of resistance[13].

A recent study concluded that the impact of adaptive mutability on the mutational load in the protein-coding genome is small[44]. This in line with previous findings showing that the tumor mutational burden is not strongly increased in cells that acquired therapy resistance[12,13]. There are many confounding factors in these not precisely controlled systems. Indeed, the adaptive mutability phenotype is probably restricted in time (that is, when the cells are maladapted to the new environment[13]), confined to a small subpopulation of cells and masked by the outgrowth of pre-existing resistant cells. Bulk analysis on tumor samples at relapse cannot disentangle these factors. Our characterization of persisters' dynamics and increased mutation rate have potential clinical relevance. First, the finding that higher drug concentrations induce an increased death rate of sensitive cells and an increased transition to persistence, a reservoir for the emergence of resistance, provides a rationale for therapeutic strategies to impair the emergence of persistence. Second, the involvement of error-prone DNA polymerases during adaptive mutability[13] offers opportunities for nonobvious combinatorial strategies to restrict drug resistance. Indeed, we show that inhibition of mutagenic TLS effectively delays the acquisition of secondary resistance.

Our methodology infers that clinically approved anticancer therapies can induce a temporary increase in the mutation rate of CRC cells. Our framework can be used to systematically measure mutation rates in mammalian cells exposed to a wide range of environmental stressors and to define drug combinations to restrict the emergence of therapeutic resistance.

## Online content

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

## Methods

**Experimental setup and data collection.** *Cell cultures.* Cells were routinely supplemented with 10% fetal bovine serum (FBS), 2 mM L-glutamine and antibiotics (100 U ml$^{-1}$ of penicillin and 100 mg ml$^{-1}$ of streptomycin), and grown at 37 °C on a 5% $CO_2$ air incubator. Cells were routinely screened for absence of *Mycoplasma* contamination using the Venor GeM Classic kit (Minerva Biolabs). All the cell lines used were confirmed negative for *Mycoplasma* contamination in all the tests performed. The identity of each cell line was checked no more than 3 months before performing the experiments using the PowerPlex 16 HS System (Promega), through short tandem repeat tests at 16 different loci (D5S818, D13S317, D7S820, D16S539, D21S11, vWA, TH01, TPOX, CSF1PO, D18S51, D3S1358, D8S1179, FGA, Penta D, Penta E and amelogenin). Amplicons from multiplex PCRs were separated by capillary electrophoresis (3730 DNA Analyzer, Applied Biosystems) and analyzed using GeneMapper v.3.7 software (Life Technologies). Short tandem repeat results for all the cell lines and corresponding clones matched the profiles previously published[45]. WiDr and DiFi CRC cell populations were obtained by R. Bernards and J. Baselga, respectively, as we previously reported[45]. JVE207 CRC cells were obtained by T. Van Wezel, Department of Pathology, University Medical Center, Leiden (the Netherlands).

*Isolation of CRC-derived clones.* CRC clones were obtained by seeding WiDr and DiFi CRC populations at a limiting dilution of 1 cell per well in 96-multiwell plates in complete medium. Clones were then selected for having growth kinetics and drug sensitivity comparable with those of the parental counterparts. For growth testing, WiDr and DiFi populations and derived clones were seeded in 96-multiwell plates ($2 \times 10^3$ and $3 \times 10^3$ cells per well for WiDr and DiFi, respectively) in complete medium. Plates were incubated at 37 °C in 5% $CO_2$. Cell viability, assessed every day for 4 d by measuring ATP content through Cell Titer-Glo Luminescent Cell Viability assay (Promega) using the Tecan Spark 10M plate reader with the Tecan SparkControl Magellan software (v.2.2), was compared with cell viability assayed at day 1. For drug-sensitivity testing, cells were seeded at different densities ($2 \times 10^3$ and $3 \times 10^3$ cells per well for WiDr and DiFi, respectively) in medium containing 10% FBS in 96-multiwell plates at day 0. The next day, serial dilutions of the indicated drugs in serum-free medium were added to the cells (ratio 1:1) in technical triplicates, whereas dimethyl sulfoxide (DMSO)-only treated cells were included as controls. Plates were incubated at 37 °C in 5% $CO_2$ for the indicated time. Cell viability was assessed by measuring ATP content with the Cell Titer-Glo Luminescent Cell Viability assay (Promega), using the Tecan Spark 10M plate reader with the Tecan SparkControl Magellan software (v.2.2). Dabrafenib was obtained from Selleckchem. Cetuximab was kindly provided by MERCK.

*Growth rates of CRC cell clones before drug treatment.* Clonal spontaneous growth is defined by the following parameters: the rate at which cells are born (birth rate, $b$), the rate at which cells die (death rate, $d$) and the net growth rate $b - d$. To estimate the $b - d$ rate, CRC cell clones were seeded at $3.5–4.0 \times 10^5$ cells per well in 6-multiwell plates. Plates were incubated at 37 °C in 5% $CO_2$. Starting from the next day, the number of viable cells was assessed by manual count in Trypan Blue 0.4% (Gibco) by two operators independently at the indicated timepoints, to obtain the clones' net growth rate. To estimate $d/b$, cells were seeded at different densities ($3.5–4.0 \times 10^5$ cells per well) in multiple 6-multiwell plates. Plates were incubated at 37 °C in 5% $CO_2$. At each timepoint, cells were collected and stained with propidium iodide (Sigma-Aldrich) following the manufacturer's instructions. Cells were then analyzed by flow cytometry. Data were acquired with the Beckman Coulter CyAn ADP instrument using the Summit v.4.3 software and analyzed with the FlowJo software (v.7.6). The cells in the sub-G1 phase were considered dead and used to estimate $d/b$. The values of birth and death rates were then obtained by combining $b - d$ and $d/b$ estimates (Supplementary Note).

*Dose–response growth curve assay.* CRC cell clones were seeded at different densities ($2 \times 10^3$ and $3 \times 10^3$ cells per well for WiDr and DiFi, respectively) in medium containing 10% FBS in multiple 96-multiwell plates at day 0. The next day, serial dilutions of the indicated drugs in serum-free medium were added to the cells (ratio 1:1) in technical triplicates, whereas DMSO-only treated cells were included as controls. Cell viability of WiDr and DiFi clones was assessed at indicated timepoints over 5 and 19 d of constant treatment, respectively, by measuring ATP content through Cell Titer-Glo Luminescent Cell Viability assay (Promega), using the Tecan Spark 10M plate reader with the Tecan SparkControl Magellan software (v.2.2).

*Single-dose growth curve assay.* DiFi and WiDr CRC cell clones were seeded in multiple 96-multiwell plates at 1,000 and 500 cells per well, respectively. Cells were allowed to expand for a fixed number of generations until a population size of 10,000–20,000 cells per well was reached. At that point, treatment was added (100 μg ml$^{-1}$ of cetuximab for DiFi and 1 μM dabrafenib + 50 μg ml$^{-1}$ of cetuximab for WiDr). Plates were then incubated at 37 °C in 5% $CO_2$ and cell viability was assessed at the indicated timepoints by measuring ATP content with the Cell Titer-Glo Luminescent Cell Viability assay (Promega), using the Tecan Spark 10M plate reader and the Tecan SparkControl Magellan software (v.2.2), over 22 and 32 d of constant treatment (for WiDr and DiFi, respectively). Medium

and treatment were renewed once a week. To test the effect of different seeding densities on the residual viability assayed, each clone was seeded at different densities (($3–20$) $\times 10^3$ cells per well) in complete medium. The next day, treatment was added (100 μg ml$^{-1}$ of cetuximab for DiFi and 1 μM dabrafenib + 50 μg ml$^{-1}$ of cetuximab for WiDr) and viability was assessed at the indicated timepoints by measuring ATP content.

*Staining with CFSE.* CRC clones were seeded at $2.5 \times 10^5$ (WiDr) and $6.5 \times 10^5$ (DiFi) cells in multiple 10-cm dishes. The next day, untreated cells were stained with CellTrace CFSE Cell Proliferation Kit (Invitrogen) according to the manufacturer's instructions. At the indicated timepoints, starting from the day after staining (T0), cells were collected and resuspended in 1 ml of phosphate-buffered saline (PBS) with Zombie Violet dye 1,000× (BioLegend) to exclude dead cells. Cells were then analyzed by flow cytometry. For persister proliferation analysis, CRC clones were seeded at $2 \times 10^4$ cells per well in several 24-multiwell plates. The next day, cells were treated with 100 μg ml$^{-1}$ of cetuximab (for DiFi) or 1 μM dabrafenib + 50 μg ml$^{-1}$ of cetuximab (for WiDr) and incubated at 37 °C in 5% $CO_2$ for 14 d (renewing treatment after 1 week), until a population of persister cells emerged in each well. Then, cells were stained with CellTrace CFSE Cell Proliferation Kit (Invitrogen) according to the manufacturer's instructions. At the indicated timepoints, starting from the day after staining (T0), plates were checked to exclude from the analysis wells containing resistant clones, whereas cells from the remaining wells were collected, resuspended in 1 ml$^{-1}$ of PBS with Zombie Violet dye 1,000× (BioLegend) to exclude dead cells and analyzed by flow cytometry. Medium and treatment were renewed once every week throughout the experiment. Flow cytometry was performed using the Beckman Coulter CyAn ADP analyzer with the Summit v.4.3 software and analyzed with a Python 3 script based on standard libraries (FlowCal, FlowKit). The following gating strategy was used: first, cells were selected with a light-scattering gate (FSLin versus SSLin), excluding cell doublets with a single-cell gate (FSArea versus SSArea). The following cutoffs were used (Supplementary Fig. 9): (1) FSLin: lower 5,000 and upper 60,000; (2) SSLin: lower 3,000 and upper 63,000; (3) FSArea: lower 3,000 and upper 60,000; SSArea: lower 2,000 and upper 63,000. We then evaluated the bi-dimensional distribution of the remaining datapoints in the space of the coordinates FSArea and SSArea, and retained all the datapoints that were included in the 99th percentile of the distribution. Viable cells were selected by excluding Zombie Violet dye-positive cells and CFSE signal was detected by measuring the FITC signal.

*Staining with EdU.* DiFi and WiDr clones were plated on several glass coverslips at $5 \times 10^4$ and $4 \times 10^4$ cells per coverslip, respectively. The next day, cells were treated with 100 μg ml$^{-1}$ of cetuximab (for DiFi) or 1 μM dabrafenib + 50 μg ml$^{-1}$ of cetuximab (for WiDr) and incubated at 37 °C in 5% $CO_2$ for 14 d (renewing treatment after 1 week), until a population of persister cells emerged on each coverslip. Then, at indicated timepoints (renewing medium and treatment once every week throughout the experiment), residual cells were stained with the Click-iT EdU Cell Proliferation Kit for Imaging (Invitrogen), according to the manufacturer's instruction. Briefly, cells were incubated with 10 μM EdU for 4 h. After that, cells were fixed in 4% paraformaldehyde for 20 min at room temperature and permeabilized with 0.5% Triton X-100 in PBS for 20 min at room temperature. Coverslips were then incubated in Click-iT reaction cocktail for 30 min, followed by nuclei staining with DAPI and F-actin staining with Alexa Fluor-555 phalloidin (50 μg ml$^{-1}$). Slides were then mounted using the fluorescence mounting medium (Dako). For quantification of EdU-positive persister cells, DAPI- and EdU-stained nuclei were detected with a Leica DMI6000B fluorescence microscope (Leica Microsystems) under a 40× dry objective using the Leica Application Suite Advanced Fluorescence software (v.2.6.3.8173). Images were analyzed with 'Analyze particles' function in ImageJ (v.1.53a) to calculate the percentage of EdU-positive cells out of the total number of cells in each slide (based on DAPI staining). Two separate technical replicates, with a minimum of 200 cells each, were analyzed for each timepoint for each biological replicate. Resistant colonies that had grown on each slide were manually identified in each image and excluded from the analysis. Representative images shown for each cell clone were acquired with a confocal laser scanning microscope (TCS SPE II, Leica), using the Leica Application Suite Advanced Fluorescence software (v.2.6.3.8173), and processed with Adobe Photoshop CS5.

*Timelapse microscopy assay.* For live cell-imaging experiments, DiFi and WiDr clones were seeded in 24-multiwell plates suitable for microscopy (μ-Plate 24 Well Black, ibidi) at $5 \times 10^4$ and $4 \times 10^4$ cells per well, respectively. The next day, cells were treated with 100 μg ml$^{-1}$ of cetuximab (for DiFi) or 1 μM dabrafenib + 50 μg ml$^{-1}$ of cetuximab (for WiDr) and incubated at 37 °C in 5% $CO_2$ for 14 d (renewing treatment after 1 week), until a population of persister cells emerged. Then, surviving persister cells were labeled with a fluorescent stain for nuclei (Nucblue, Invitrogen, at 0.5 drop per well) to track cell numbers and detect cell divisions, and a live fluorescent marker for the activation of caspase-3/7 (CellEvent Caspase-3/7 Green Detection Reagent, Invitrogen, at 1 drop per well) to detect cell death. Both dyes were used as recommended by the manufacturer. After labeling, cells were monitored for 5 d under an inverted widefield microscope

(Nikon Lipsi, ×20 Plan Apo objective with 0.75 numerical aperture), acquiring images every 45 min with the Nis-Element AR software (v.5.21.03, 64 bit; Nikon). For each clone, two separate wells with 16 fields of view each were monitored, for a total of more than 1,300 cells for each cell model. By digital image segmentation carried out with Ilastik (v.1.3.3 opensource)[46,47], the number of cells as a function of time for each frame was quantified and then fitted to an exponential function to extract the net growth rate ($b - d$). Data analysis and manipulations were performed by means of customized Matlab R2121a (Mathworks) scripts. Representative snapshots of cell division events and apoptotic events (obtained with Fiji v.1.53 opensource) are reported in Extended Data Fig. 3, whereas Supplementary Videos report whole timelapse experiments for selected fields of view for each clone. Scale bar in Supplementary Videos, 200 µm.

**Characterization of distribution of persister cells.** DiFi and WiDr cell clones were seeded in multiple 96-multiwell plates at 1,000 or 500 cells per well, respectively. Subsequently, cells were allowed to expand until they reached 10,000–20,000 cells per well. Cell viability was then assessed by measuring ATP content to normalize for cell number before treatment initiation. The remaining plates were treated with targeted therapies (100 µg ml⁻¹ of cetuximab for DiFi and 1 µM dabrafenib + 50 µg ml⁻¹ of cetuximab for WiDr). Medium and treatment were renewed once a week. After 3 weeks of constant drug treatment, residual viability was assessed by measuring ATP content with the Cell Titer-Glo Luminescent Cell Viability assay (Promega), using the Tecan Spark 10M plate reader with the Tecan SparkControl Magellan software (v.2.2).

*Two-step fluctuation assay.* DiFi and WiDr clones were seeded at 1,000 or 500 cells per well, respectively, in 20 96-multiwell plates each, for a total of 1,920 independent replicates. Cells were allowed to expand for a fixed number of generations until they reached 10,000–20,000 cells per well. Next, treatment was administered (100 µg ml⁻¹ of cetuximab for DiFi and 1 µM dabrafenib + 50 µg ml⁻¹ of cetuximab for WiDr). Plates were incubated at 37 °C in 5% CO₂ for the indicated time. Medium and drug treatment were renewed once a week. After 3–4 weeks of treatment, pre-existing resistant colonies were clearly distinguishable at the microscope and counted by two independent observers. The number of pre-existing resistant clones was used to estimate the spontaneous mutation rate of CRC clones (Supplementary Note). After 10–11 weeks, resistant colonies started to emerge in wells where only persisters were previously present. The number of persister-derived resistant clones was used to estimate the mutation rate of persister cells under constant treatment (Supplementary Note). Pictures of the resistant colonies were acquired using a ZEISS Axio Vert. A1 microscope equipped with a True Chrome HD II camera.

*Droplet digital PCR analysis.* Genomic DNA (gDNA) was extracted using ReliaPrep gDNA Tissue Miniprep system System (Promega). Purified gDNA was amplified using ddPCR Supermix for Probes (BioRad) using *RAS* (PrimePCR ddPCR Mutation Assay, BioRad or customized) droplet digital (dd)PCR assay for point mutation detection. The ddPCR was performed according to the manufacturer's protocol. Briefly, 5 µl of DNA template was added to 10 µl of ddPCR Supermix for Probes (BioRad), 1 µl of the primer and probe mixture. Droplets were generated using the Automated Droplet Generator (Auto-DG, BioRad) and transferred to a 96-well plate and then thermally cycled with the following conditions: 10 min at 95 °C, 40 cycles of 94 °C for 30 s, 55 °C for 1 min followed by 98 °C for 10 min (ramp rate = 2.5 °C s⁻¹). Droplets were analyzed with the QX200 Droplet Reader (BioRad) and the QuantaSoft software (v.1.7.4.0917; BioRad) for fluorescent measurement of FAM and HEX probes. Gating was performed based on positive and negative controls and mutant populations were identified. The ddPCR data were analyzed with QuantaSoft analysis software (v.1.7.4.0917; BioRad) and results were reported as the percentage (fractional abundance (FA)) of mutant DNA alleles to total (mutant plus wild-type) DNA alleles. FA is calculated as follows: percentage $FA = (N_{mut}/(N_{mut} + N_{WT}) \times 100)$, where $N_{mut}$ is the number of mutant events and $N_{WT}$ the number of wild-type events per reaction. The ddPCR analysis of normal control DNA (from cell lines) and no DNA template controls were always included.

*Library preparation and genetic analysis of whole-genome sequencing.* The gDNA was extracted using ReliaGen gDNA Tissue Miniprep system System (Promega) and sent to IntegraGen SA (Evry, France) which performed library preparation. DNA libraries were paired-end sequenced on Illumina HiSeq4000 and FASTQ files produced by IntegraGen were analyzed at Candiolo Cancer Institute. A BWA-mem algorithm was performed to align sequences on the reference human genome v.19. The resulting files were cleaned of PCR duplicates using the 'rmdup' sam tools command. For each cell line, somatic mutation analysis was performed subtracting variations found in parental (sensitive) samples to resistant counterparts according to what has been previously published[48]. For each cell line pre- and post-treatment, gene copy number (GCN) was computed as follows: first the median read depth of all genomic regions was calculated; next, for each gene the median read depth was obtained and then divided by the former value. For each gene, its GCN in the pre- and post-treatment samples and the corresponding CNV (ratio between matched GCNs) were reported. DNAcopy R module was performed to cluster CNV using a circular binary segmentation algorithm.

*TTP assay.* TTP assays were conducted as previously described[40]. Briefly, 5 million cells (for WiDr and DiFi cells) and 4.5 million cells (for JVE207 cells) were plated for each treatment condition. Then, cells were treated with MAPK pathway inhibitors (dabrafenib 1 µM + cetuximab 30 µg ml⁻¹ for WiDr and JVE207, cetuximab 50 µg ml⁻¹ for DiFi), REV1 inhibitor (2 µM) or their combination, in parallel. Medium and treatment(s) were renewed weekly. Cells were counted each week; counts as 0 represent timepoints in which cells were too few and only medium and drug refreshments were done.

**Materials availability.** The CRC cell clones generated in the present study are available through A. Bardelli (Department of Oncology, University of Torino) under a material transfer agreement.

*Theoretical modeling and computational analyses.* In the present study, we developed and used two distinct mathematical models to investigate the dynamics of cell populations. The first model describes the transition-to-persister state (TP model), and is a birth–death model with phenotypic switching, which we explored in the deterministic limit. We considered four different model variants and compared them with experimental data to infer the most likely scenario for the sensitive-to-persister transition. The second model, which we named the 'MC–LD' model, is a fully stochastic birth–death branching process that includes the mutational processes of sensitive (untreated) and persister cells (under treatment). To measure the mutation rate, stochastic fluctuations cannot be neglected. We simulated individual trajectories of the Markov process underlying the evolution of the MC–LD model by a coarse-grained version of the Gillespie algorithm[49], which groups together all stochastic events happening in discrete time intervals of fixed duration $\Delta t$.

For the inference of the birth–death rates $b$ and $d$, we used the data on growth rates of CRC clones before drug treatment. Our inference scheme is summarized in Supplementary Fig. 2a. The parameters of the TP model were inferred using a Bayesian framework and data from the single-dose and dose–response assays. Posterior distributions of the model parameters were sampled using a Hamiltonian Monte Carlo algorithm (Python 3, package pymc3, NUTS sampler)[50]. TP model variants were compared by means of the standard BIC and the AIC. To infer mutation rates for the MC–LD model, we computed an approximate analytical expression for the probability of the emergence of one mutant in an expanding population of cells in a given time interval $[0, T]$, and used it to derive estimators for the emergence of mutations before and during treatment administration (from persisters). The mutation rate of persister cells was inferred with a Bayesian framework, to account for the uncertainty of the value of the initial fraction of persister cells, $f_0$.

All the details on the theoretical/computational protocols are provided in Supplementary Note.

**Reporting summary.** Further information on research design is available in the Nature Research Reporting Summary linked to this article.

## Data availability

Data used for the analysis, source data images of EdU staining and live microscopy assay are available as a repository on Mendeley Data (https://doi.org/10.17632/mvfm7hs9kw.2)[51]. Sequencing data are available at European Nucleotide Archive accession no. PRJEB49483 (https://www.ebi.ac.uk/ena/browser/home). The CRC cell clones generated in the present study are available through A. Bardelli under a material transfer agreement.

## Code availability

All the customized code used in our analyses is available as a repository on Mendeley Data (https://doi.org/10.17632/mvfm7hs9kw.2)[51]. Bioinformatics code for sequencing data analysis are available at https://bitbucket.org/irccit/idea/src/master. The code used for the simulations of the model has been written in: C++ (C++ 14 and g++ 10.3.0), Mathematica (v.10), Python (v.3.9.7), Matlab (v.R2121a, Mathworks) and Microsoft Excel (v.16.48).

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

## Acknowledgements

We thank A. Ciliberto, M. Osella, L. Trusolino and A. Amir for useful discussions and critical reading of the manuscript. This work was supported by the following: FONDAZIONE AIRC under 5 per Mille 2018 (ID. 21091 program) to A. Bardelli, group leaders F.D.N. and A. Bertotti; Associazione Italiana per la Ricerca sul Cancro AIRC IG (grant no. 23258) to M.C.L.; AIRC IG 2017 no. 20697 to A. Bertotti; European Research Council (ERC) Consolidator (grant no. 724748, BEAT) to A. Bertotti; AIRC IG 2018 (ID. 21923 project) to A. Bardelli; AIRC under MFAG 2021-ID 26439 project to M.R.; AIRC under MFAG 2020-ID 25040 project to A.P.; International Accelerator Award, ACRCelerate, jointly funded by Cancer Research UK (nos. A26825 and A28223), FC AECC (no. GEACC18004TAB) and AIRC (no. 22795) to A. Bardelli; Ministero Salute (RC 2020) to A. Bardelli; European Research Council (ERC) under the European Union's Horizon 2020 research and innovation program (grant no. 101020342) to A. Bardelli; IMI contract no. 101007937 PERSIST-SEQ to A. Bardelli and M.R.; BiLiGeCT (Progetto PON ARS01_00492) to A. Bardelli. S.P. was supported by Fondazione Umberto Veronesi.

## Author contributions

M.R., A. Bardelli and M.C.L. conceived the study and contributed with key ideas at different stages. M.R. and A.S. performed the biological experiments. S.P., M.C., M.G. and M.C.L. conceived the modeling framework. S.P., M.C. and M.G. performed data analysis, model simulations and analytical calculations. J.E. performed timelapse live microscopy experiments. A.P. performed timelapse live microscopy data analysis. G.C. performed whole-genome sequencing data analysis. S.L. performed ddPCR analysis. A. Bertotti and F.D.N. contributed to data discussion. M.R., S.P., A. Bardelli and M.C.L. wrote the paper. All authors read and approved the final version of the paper.

## Competing interests

A. Bardelli reports receiving commercial research grants from Neophore, AstraZeneca and Boehringer; he is an advisory board member/unpaid consultant for Inivata and Neophore, holds ownership interest in Neophore, and is an advisory board member/consultant for Illumina, Guardant Health, Inivata and Roche/Genentech Global CRC. All the other authors declare no competing interests.

## Additional information

**Extended data** are available for this paper at https://doi.org/10.1038/s41588-022-01105-z.

**Correspondence and requests for materials** should be addressed to Alberto Bardelli or Marco Cosentino Lagomarsino.

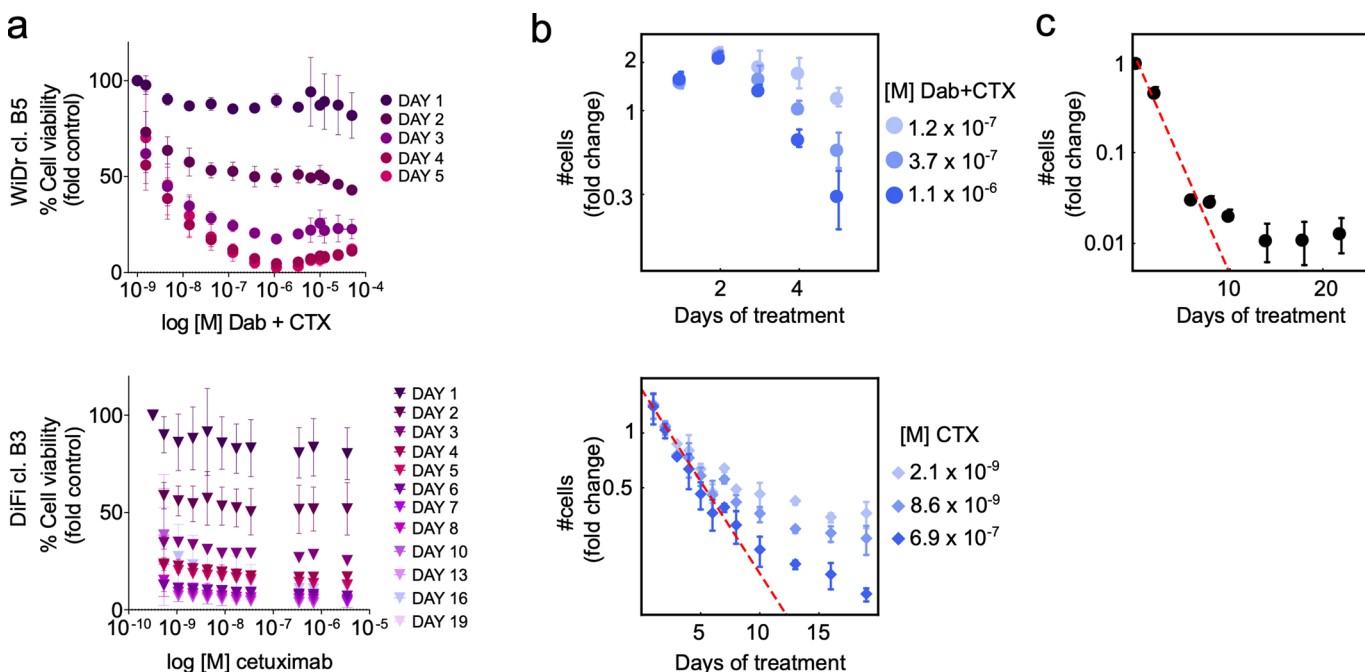

**Extended Data Fig. 1 | Population dynamics of additional CRC clones WiDr cl. B5 and DiFi cl. B3 in response to targeted therapies. a**, In the doses-response assay, WiDr cells were treated with increasing concentrations of dabrafenib (Dab) + 50 μg/ml cetuximab (CTX), while DiFi received increasing concentrations of cetuximab. Cell viability was measured by the ATP assay at the indicated time points. Results represent the average ± SD (n = 3 biologically independent experiments for WiDr; n = 4 biologically independent experiments for DiFi). **b**, Growth curves of the indicated cells under treatment, reported as fold-change of viable cells (log scale) vs time of drug exposure, were calculated from doses-response assay data by normalizing cell viability at the indicated time points by the viability measured at day 0. Growth curves for three different drug concentrations for each clone are shown as average ± SD (n = 3 biologically independent experiments for WiDr; n = 4 biologically independent experiments for DiFi). Lower panel: the total number of viable cells is compatible with an exponential decay with two-time scales, supporting the outgrowth of persisters (the dashed line indicates the initial slope). **c**, Fold-change of viable cells (log scale, assessed by ATP assay) vs time of drug exposure for WiDr cl. B5 in the single-dose assay. Symbols and error bars indicate means and standard deviations (n = 2 biologically independent experiments). The total number of viable cells is compatible with an exponential decay with two-time scales, supporting the outgrowth of persisters (the dashed line indicates the initial slope).

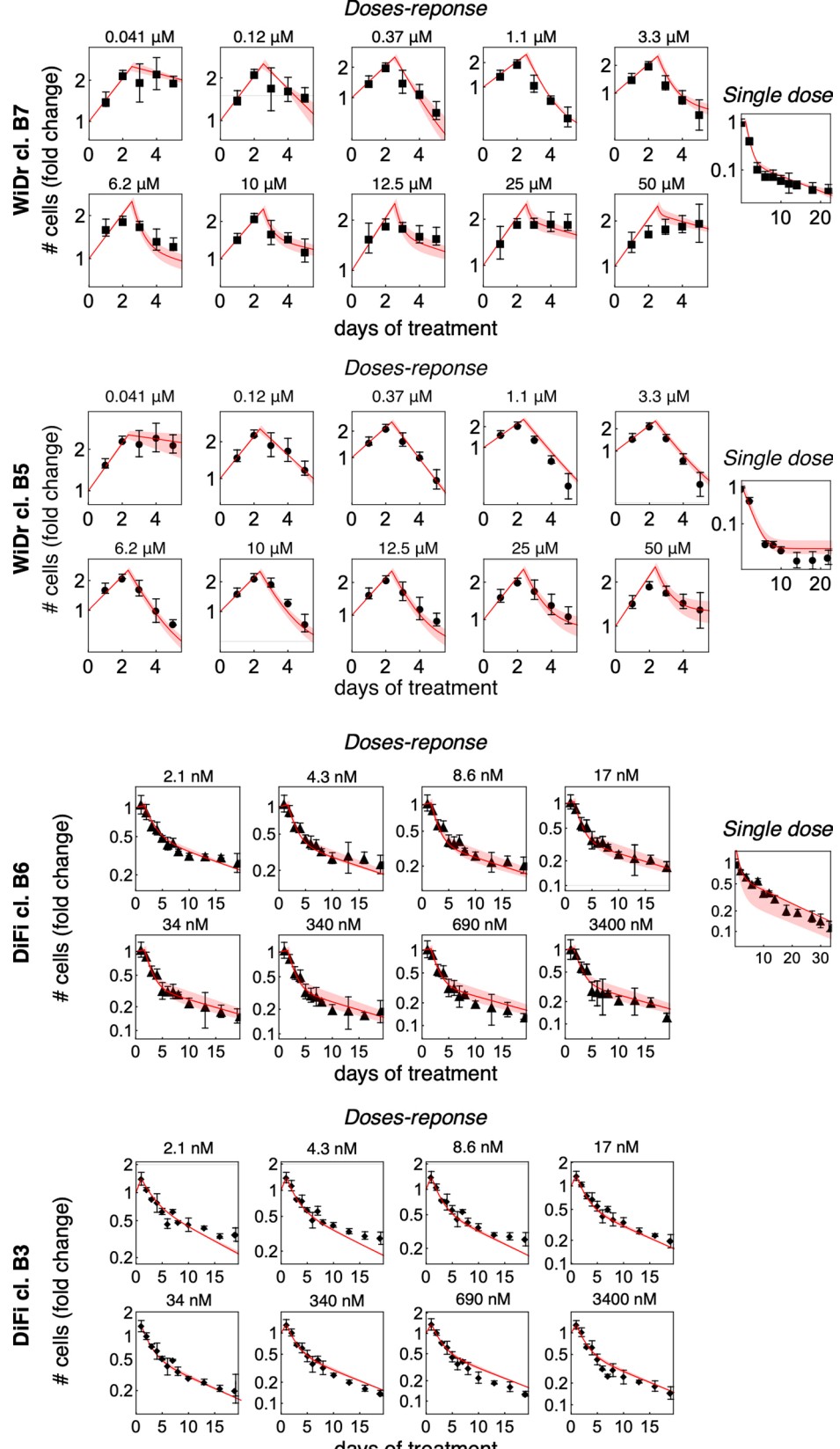

**Extended Data Fig. 2 | Transition to persisters (TP) model of CRC cells.** Fitted experimental data of the doses-response (n = 3 biologically independent experiments for WiDr, n = 5 biologically independent experiments for DiFi cl. B6 and n = 4 biologically independent experiments for DiFi cl. B3, left side) and single-dose (n = 2 biologically independent experiments, right side) datasets are indicated with black dots and error bars (mean values ± standard deviation); red lines and shadowed areas represents the model fit (Credible Interval [2.5, 97.5]%). The drug concentration is specified above each plot.

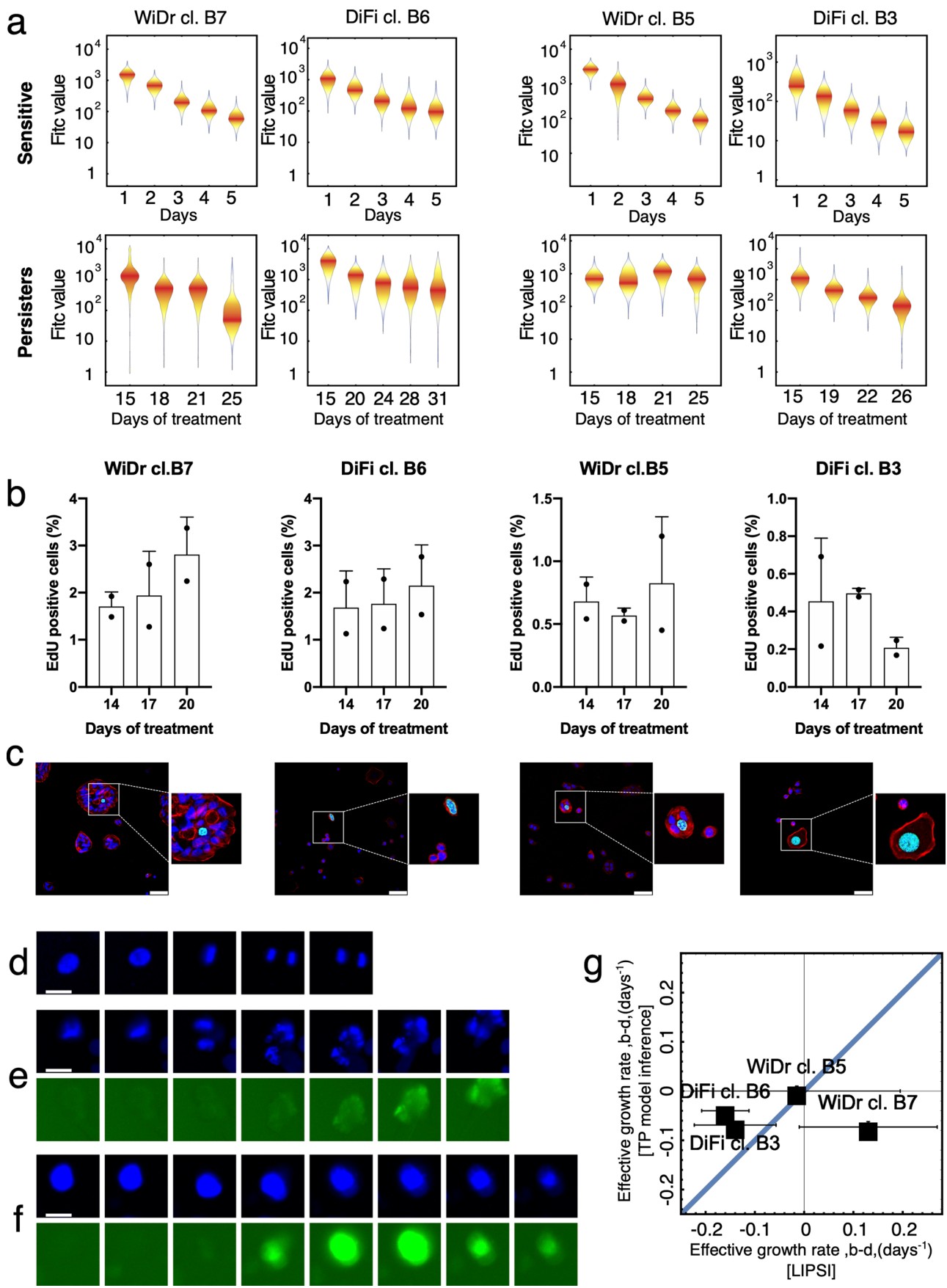

**Extended Data Fig. 3 | See next page for caption.**

**Extended Data Fig. 3 | A fraction of persister cells slowly replicate during drug treatment. a**, Distribution of the CFSE signal (Fitc) measured by flow cytometry is reported for the indicated timepoints. In each distribution the number of cells retained after the gating process was ≥ 3000. WiDr and DiFi cells were grown in standard conditions (untreated) or treated with 1 μM dabrafenib + 50 μg/ml cetuximab or 100 μg/ml cetuximab, respectively, for 2 weeks until the emergence of surviving persister cells. Both untreated (sensitive) and persister cells were stained with CFSE to quantify cell division and fluorescent signal (Fitc) was analyzed by flow cytometry at indicated time points. One representative experiment (n = 2 biologically independent experiments for untreated sensitive cells; n = 3 biologically independent experiments for persisters) is reported. **b**, Quantification of EdU positive cells at indicated time points. Surviving persisters emerged after 2 weeks of treatment of DiFi and WiDr clones with targeted therapies were labeled with EdU for 4 hours, then fixed and analyzed by fluorescence microscopy. Results represents mean ± SD of two independent experiments, each represented by two technical replicates. **c**, Representative images of EdU positive persister cells for each CRC clone analyzed of 2 biologically independent experiments. Green: EdU; Blue: DAPI; Red: Phalloidin. Scale bar 50μm. **d-f**, Representative snapshots of cell division events and apoptotic events observed in the time-lapse microscopy assay of 3 biologically independent experiments. Surviving persisters emerged after 2 weeks of treatment of DiFi and WiDr clones were stained with Nucblue® (a dye for nuclei, in blue) and Caspase Cell Event™ (a live marker for the activation of Caspase3/7, in green) while maintaining drug pressure, and monitored for 5 days under an inverted widefield microscope. **d**, cell division; **e**, cell death through apoptosis after a cell division; **f**, cell death through apoptosis in non-dividing cells. Scale bar in panels d-f: 20 μm. Mitotic events are reported with consecutive frames that are 45 minutes apart, while apoptotic events are reported 3 frames apart. **g**, Numerical values of the effective growth rates of persister cells evaluated by means of imaging-based assays and by TP model inference are compatible. The scatter plot compares the values of the effective growth rate (b-d) obtained with image segmentation (x axis, n = 1 representative biological replicate, consisting of 32 independent fields of view) and inferred with the TP model from dose-response and single dose assays (y axis, n = 3 biologically independent experiments for WiDr cl. B5 and cl. B7, n = 5 for DiFi cl. B6 and n = 4 for Difi cl. B3). Squares correspond to mean values and bars to standard deviations. The blue line marks the region of a perfect match between the two values.

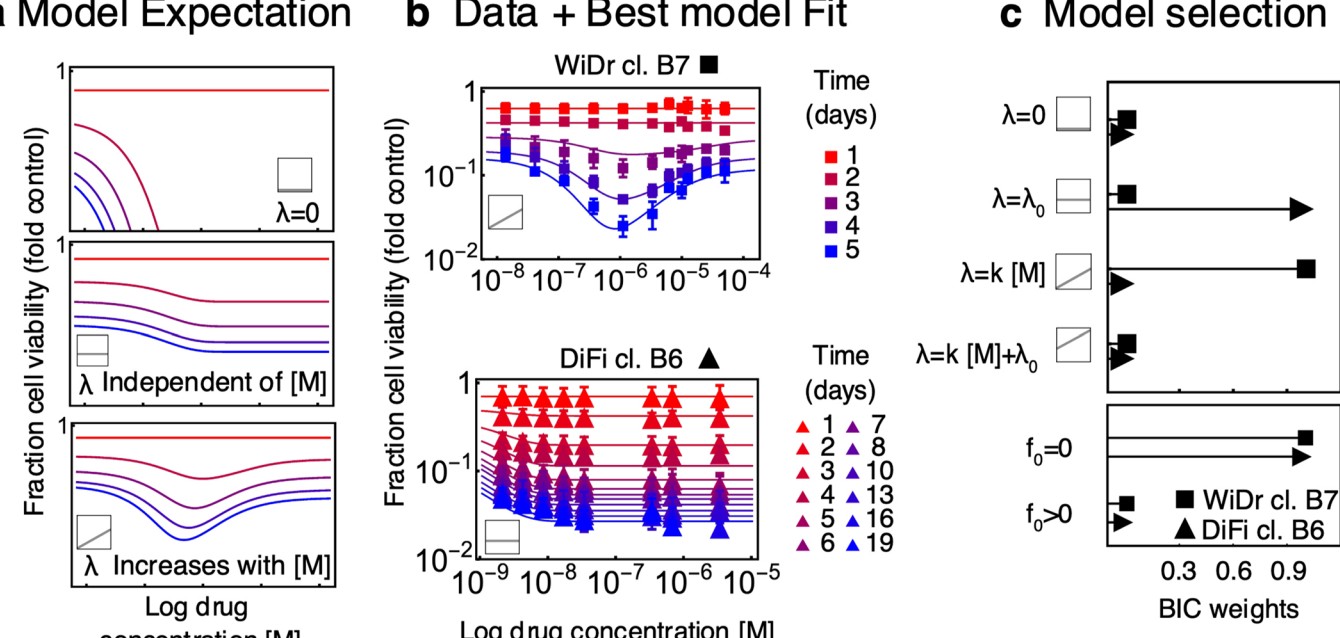

**Extended Data Fig. 4 | Bayesian inference of experimental data defines a model for the transition to persistence. a**, Schematic representation of the expected pattern of the doses-response curves for the three different model configurations: (i) null model with no transition to persistence ($\lambda = 0$, top), (ii) model with constant transition rate (mid), (iii) model with transition rate proportional to the drug concentration (bottom). Square plots indicate the dependence of $\lambda$ vs [M]. **b**, Experimentally measured dose-responses growth curves and best model fit. The doses-response datasets were normalized to the growth of the untreated cells using the best model parameters (Supplementary Table 4; n = 3 biologically independent experiments for WiDr and n = 5 biologically independent experiments for DiFi). Data are presented as mean values ± standard deviation. **c**, In the top panel, we show the Bayesian weights to compare the 4 $\lambda$ models. Bayes weights are defined as $\frac{1}{Z} \exp\left(-\text{BIC}\left(\lambda = 0\right)/2\right)$ for the model configuration with $f_0 = 0$, and as $\frac{1}{Z} \exp\left(-\text{BIC}\left(\lambda, f_0 = 0\right)/2\right) + \exp\left(-\text{BIC}\left(\lambda, f_0 > 0\right)/2\right)$ for the other models. The partition function Z ensures the global normalization $Z = \sum_{\lambda, f_0} \exp\left(-\text{BIC}\left(\lambda, f_0\right)/2\right)$. Similarly, in the bottom panel we show the two Bayes factors to compare model configuration with $f_0 = 0 \rightarrow \frac{1}{Z} \sum_{\lambda > 0} \exp\left(-\text{BIC}\left(\lambda, f_0 = 0\right)/2\right)$ (drug induced scenario) and for $f_0 > 0 \rightarrow \frac{1}{Z} \sum_{\lambda > 0} \exp\left(-\text{BIC}\left(\lambda, f_0 > 0\right)/2\right)$. Values of the BIC values are reported in Supplementary Table 3.

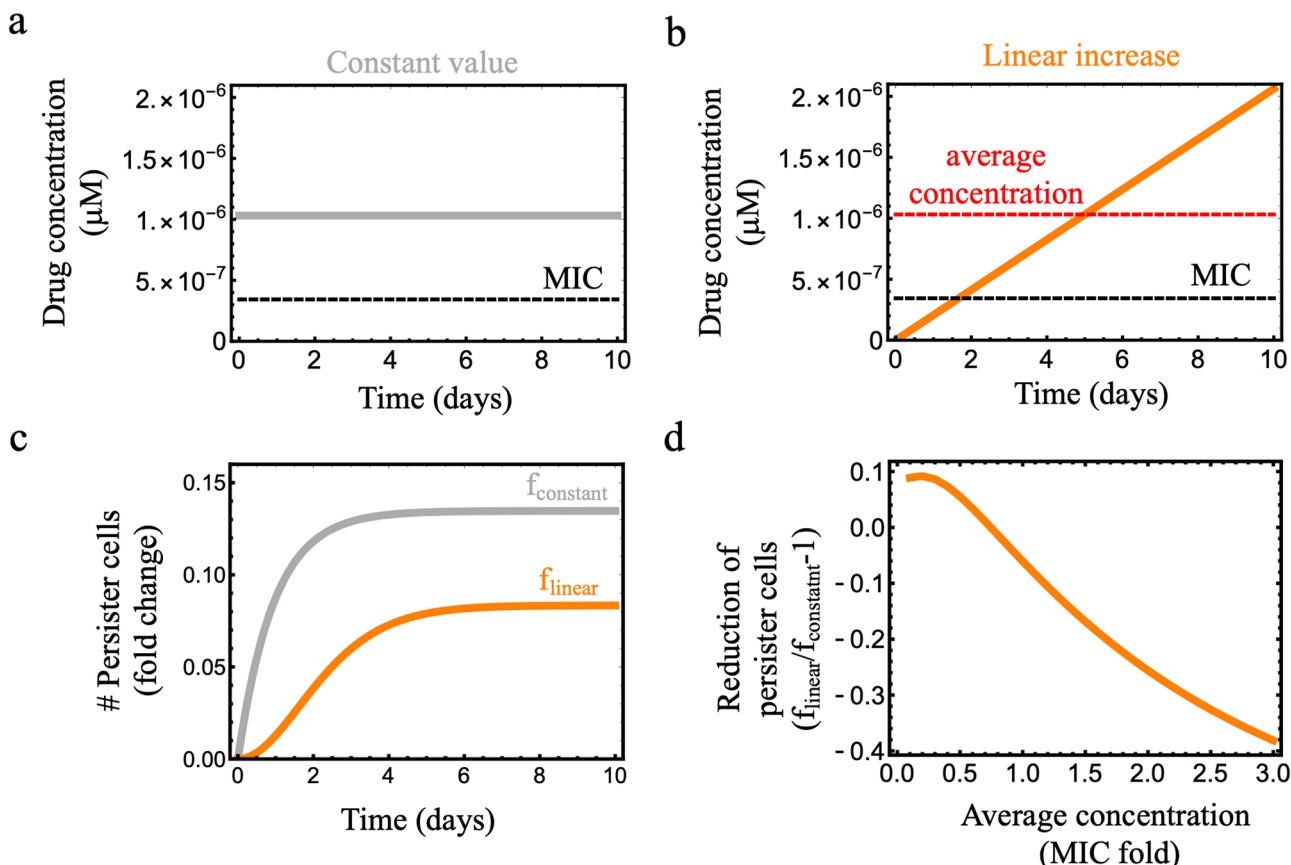

**Extended Data Fig. 5 | The dependence on the drug concentration of the transition rate to persistence can be exploited to reduce the rate of emergence of persisters. a**, The plot illustrates a standard drug delivery strategy, which consists in a constant drug concentration delivered over time. The values for the concentration of dabrafenib and cetuximab are realistic for colorectal cancer cells similar to WiDr cl. B7. The drug concentration of cetuximab is also assumed constant over time. The value of the Minimum Inhibitory Concentration (dashed black line) is the minimal value of the drug concentration for which a population of sensitive cancer cells would not increase over time. For the case of WiDr cl. B7, the value of the MIC can be obtained from our estimated model parameters, MIC $\simeq a^{-1} = 3.5 \times 10^{-7}$ μM (see Supplementary Table 4). **b**, In a modified drug delivery strategy, the drug concentration increases linearly over time. This strategy is constrained to have the same average drug concentration as in panel a, meaning that the total amount of drug delivered in 10 days is the same in the two strategies. In this case, at the beginning of the in-silico treatment, the delivered drug is below the MIC value. **c**, Expected number of persister cells predicted by the TP model (equation 1) for WiDr cl. B7 cells, corresponding to the two alternative drug delivery strategies: constant value (gray line) and linear increase (orange line). **d**, Relative fold change of the number of persister cells emerged with the drug strategy b vs drug strategy a, evaluated as a function of the average drug concentration.

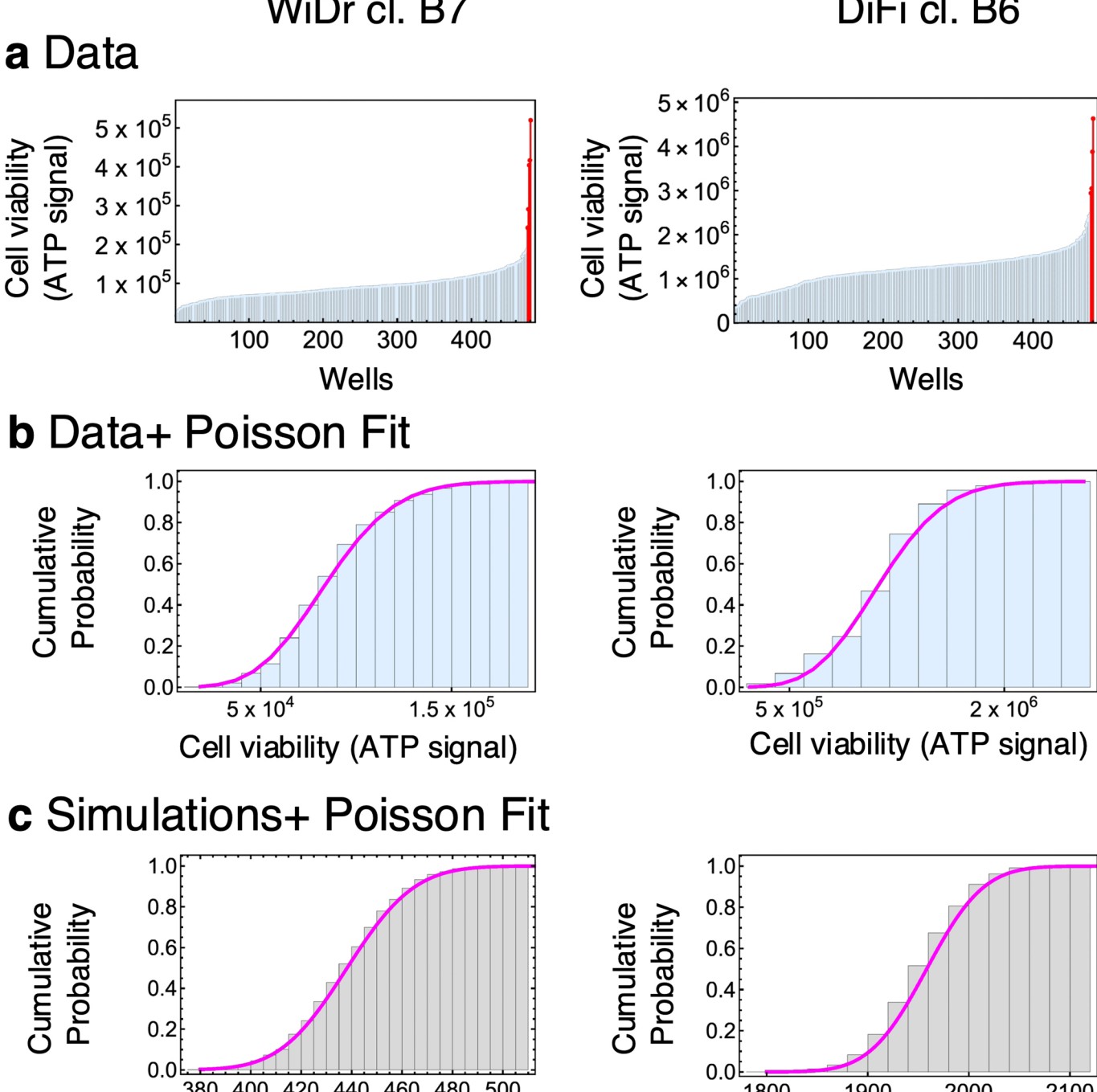

**Extended Data Fig. 6 | Distribution of persisters abundance across wells is compatible with the TP model expectation of f0=0. a**, CRC clones were seeded in multiple 96-multiwell plates and allowed to expand for about 8 cell divisions in the absence of drug, and then treated with 1 μM dabrafenib + 50 μg/ml cetuximab (WiDr) or 100 μg/ml cetuximab (DiFi) for 3 weeks. Cell viability was determined by ATP assay. Wells containing rapidly proliferating pre-existing resistant colonies are marked in red, while the remaining wells contained a small number of viable cells, which we identified as drug-tolerant persisters (indicated in light-blue). Each bar represents one well. One representative experiment of two biologically independent replicates is reported. **b**, Cumulative distribution of persisters cell viability across wells (light-blue bars) is compatible with a Poisson cumulative distribution (magenta solid line, see Supplementary methods for details on the fitting procedure). One representative experiment of two biologically independent replicates is reported. **c**, The simulation of a stochastic model for the transition to persistence shows that in the case of $f_0 = 0$ (that is, in the drug-induced scenario) the number of persister cells per well is expected to have a Poisson distribution. Simulations were performed using the TP model fit values as input parameters, and setting the initial population size to 15000 cells/well. Simulations were stopped at 21 days; the distribution was evaluated using n = 1000 independent simulations.

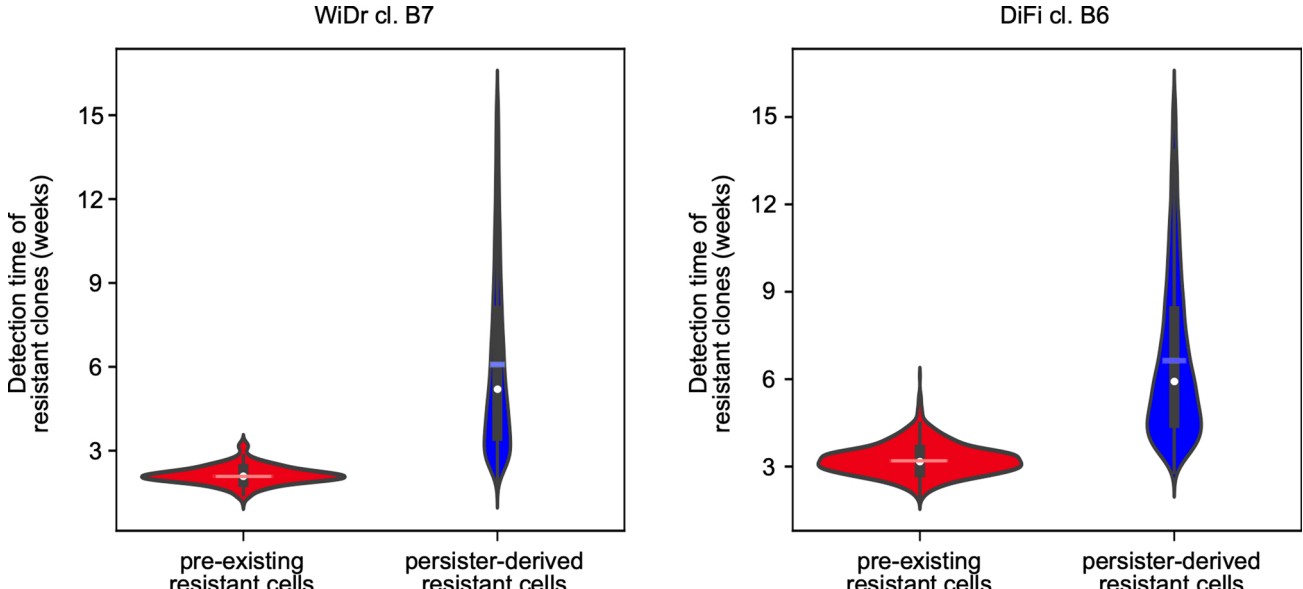

**Extended Data Fig. 7 | Pre-existing and persister-derived resistant cells emerge at different time points according to the MC-LD model.** For each clone we simulated the MC-LD fluctuation test using experimentally determined cellular parameters. The violin plot showing the time of emergence distribution is reported for pre-existing resistant cells (indicated in red) and persisters-derived resistant cells (depicted in blue). Median and mean of the distribution are represented as a white dot and a nuanced horizontal line, respectively. The thick black bar in the center indicates the interquartile range (25% and 75% percentiles) of the distribution, while the thin black line represents all the distribution except for outliers. The estimated distribution (red and blue areas) extends from the minimum to the maximum value of the data on each side of the central region, with wider and skinner sections in correspondence to regions of higher and lower probability, respectively. As previously reported, after about 3-4 weeks the vast majority of early-emerging resistant clones originate from pre-existing resistant cells, while persisters-derived resistant cells slowly accumulate subsequently over time. We run 50 simulations using the best model fit as input parameters. In each simulation, we simulated 1920 in silico independent wells. A well was considered to harbor a resistant clone when the number of resistant cells was above 20,000 units.

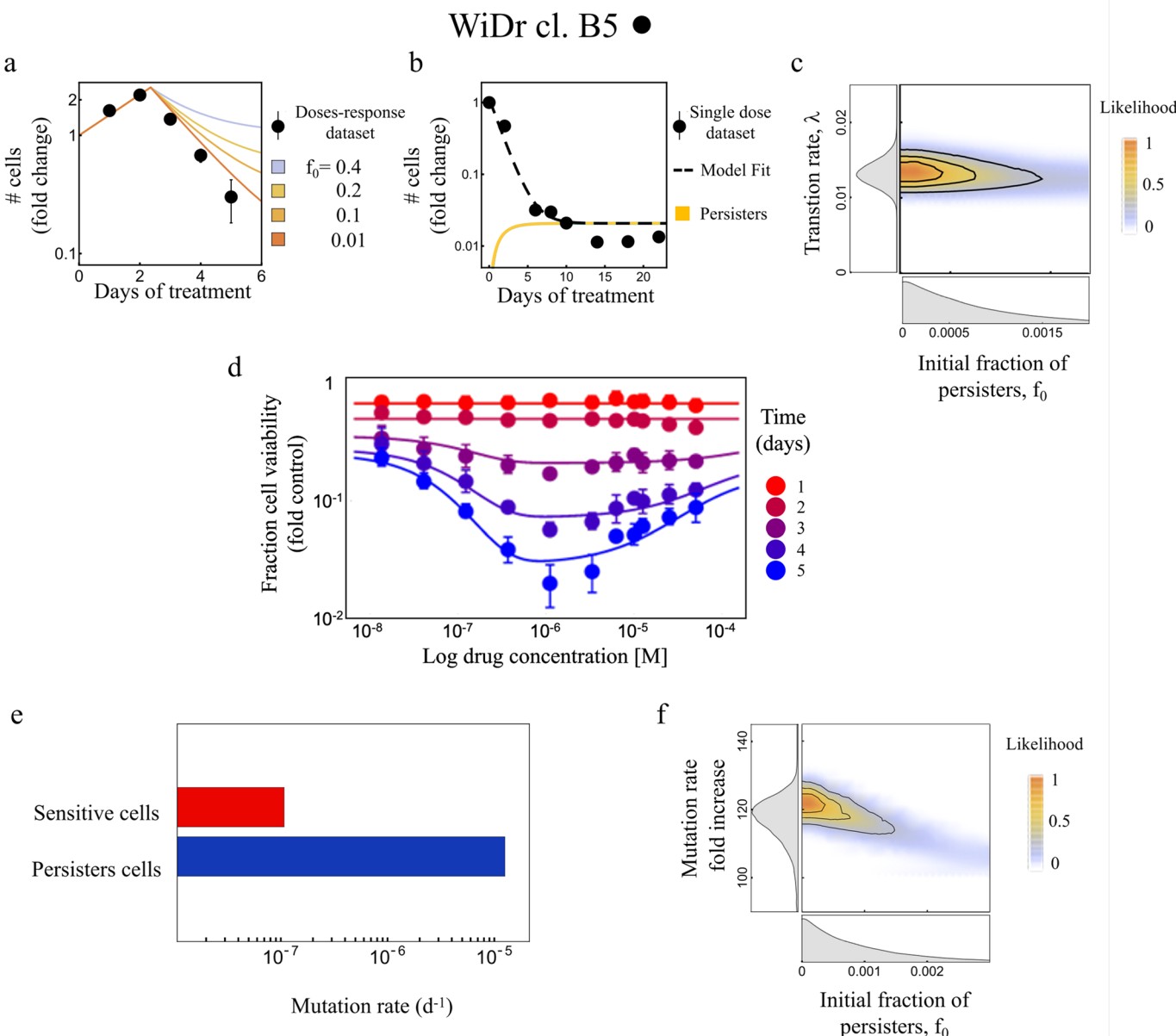

**Extended Data Fig. 8 | The analysis of an additional WiDr clone yields consistent results.** The plots show the results of the TP model and MC-LD model inference for populations derived from an additional WiDr clone, namely WiDr cl. B5. **a**, Doses-response dataset and best TP model fit as a function of the initial fraction of persister cells (n = 3 biologically independent experiments). Data are presented as mean values ± standard deviation. **b**, Single-dose dataset, best TP model fit and expected fraction of persister cells (n = 2 biologically independent experiments). **c**, Joint posterior distribution of the TP model for the initial fraction of persisters and the transition rate. **d**, Experimentally measured dose-response curves data (symbols) and best TP model (continuous lines) (n = 3 biologically independent experiments). Data are presented as mean values ± standard deviation. **e**, Quantification of mutation rates for sensitive (red) and persister (blue) cells in the MC-LD experiment (n = 2 biologically independent experiments). Here, the bar chart is used as graphical representation of inferred mutation rates (see Supplementary Table 5 for the corresponding numerical values) **f**, Joint posterior distribution of the MC-LD model for initial fraction of persisters and the fold increase of the mutation rate of persister cells.

# DiFi cl. B3 ◆

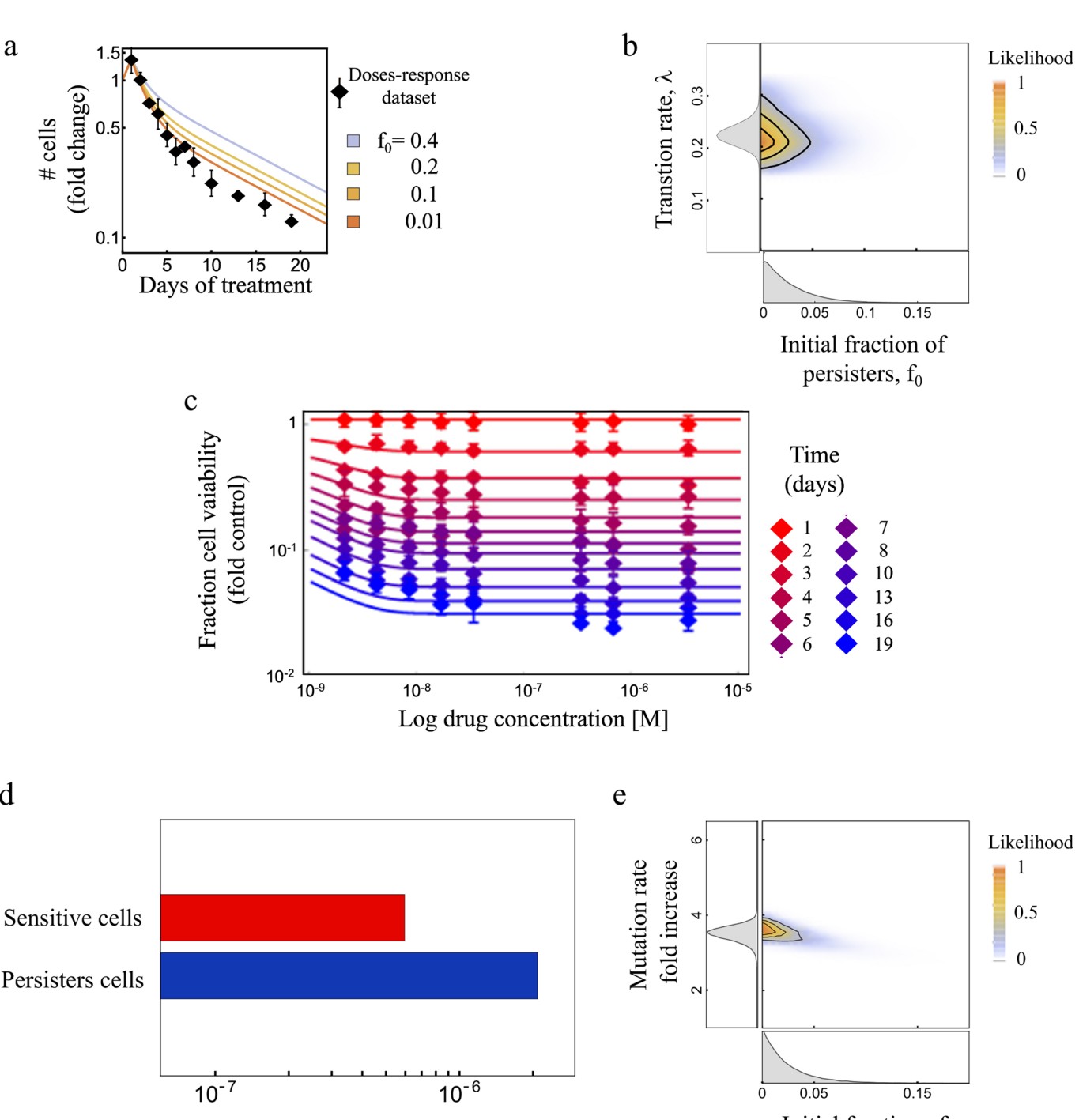

**Extended Data Fig. 9 | The analysis of an additional DiFi clone yields consistent results.** The plots show the results of the TP model and MC-LD model inference for populations derived from an additional DiFi clones, namely DiFi cl. B3. **a**, Doses-response dataset and best TP model fit as a function of the initial fraction of persister cells (n = 4 biologically independent experiments). Data are presented as mean values ± standard deviation. **b**, Joint posterior distribution of the TP model for the initial fraction of persisters and the transition rate. **c**, Experimentally measured dose-response curves data (symbols) and best TP model (continuous lines) (n = 4 biologically independent experiments). Data are presented as mean values ± standard deviation. **d**, Quantification of mutation rates for sensitive (red) and persister (blue) cells in the MC-LD experiment (n = 2 biologically independent experiments). Here, the bar chart is used as graphical representation of inferred mutation rates (see Supplementary Table 5 for the corresponding numerical values). **e**, Joint posterior distribution of the MC-LD model for initial fraction of persisters and the fold increase of the mutation rate of persister cells.

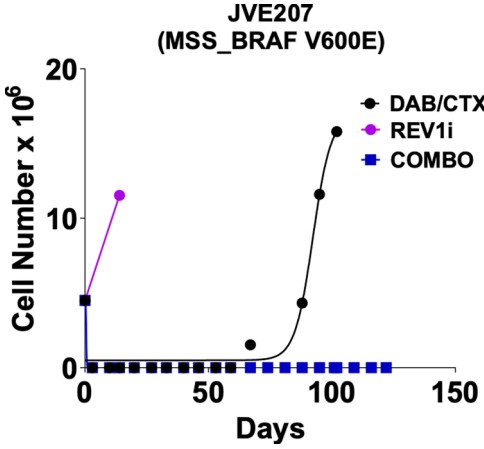

**Extended Data Fig. 10 | REV1 inhibition restricts the development of resistance to targeted therapy in CRC.** JVE207, an MSS *BRAF* V600E CRC cell line, were treated in parallel with MAPK pathway inhibitors anti-EGFR inhibitor cetuximab (CTX) in combination with anti-BRAF inhibitor dabrafenib (DAB), REV1 inhibitor or their combination, and the number of cells was monitored during the treatment, until the emergence of resistance (n = 1 biological experiment).

| | |
|---|---|

# Reporting Summary

## Statistics

For all statistical analyses, confirm that the following items are present in the figure legend, table legend, main text, or Methods section.

| n/a | Confirmed | |
|---|---|---|
| ☐ | ☒ | The exact sample size (*n*) for each experimental group/condition, given as a discrete number and unit of measurement |
| ☐ | ☒ | A statement on whether measurements were taken from distinct samples or whether the same sample was measured repeatedly |
| ☒ | ☐ | The statistical test(s) used AND whether they are one- or two-sided<br>*Only common tests should be described solely by name; describe more complex techniques in the Methods section.* |
| ☒ | ☐ | A description of all covariates tested |
| ☒ | ☐ | A description of any assumptions or corrections, such as tests of normality and adjustment for multiple comparisons |
| ☐ | ☒ | A full description of the statistical parameters including central tendency (e.g. means) or other basic estimates (e.g. regression coefficient) AND variation (e.g. standard deviation) or associated estimates of uncertainty (e.g. confidence intervals) |
| ☒ | ☐ | For null hypothesis testing, the test statistic (e.g. *F*, *t*, *r*) with confidence intervals, effect sizes, degrees of freedom and *P* value noted<br>*Give P values as exact values whenever suitable.* |
| ☐ | ☒ | For Bayesian analysis, information on the choice of priors and Markov chain Monte Carlo settings |
| ☒ | ☐ | For hierarchical and complex designs, identification of the appropriate level for tests and full reporting of outcomes |
| ☒ | ☐ | Estimates of effect sizes (e.g. Cohen's *d*, Pearson's *r*), indicating how they were calculated |

*Our web collection on statistics for biologists contains articles on many of the points above.*

## Software and code

Policy information about availability of computer code

| Data collection | Tecan SparkControl Magellan (v. 2.2); Summit 4.3.; Leica Application Suite Advanced Fluorescence (v. 2.6.3.8173); Nis-Element AR (v. 5.21.03 64 bit); QuantaSoft (v. 1.7.4.0917). |
|---|---|
| Data analysis | GraphPad prism (v.8); FlowJo (v.7.6); ImageJ (v. 1.53a); Adobe Photoshop CS5; Ilastik (v. 1.3.3 opensource); Matlab R2121a (The mathworks); Fiji 1.53 opensource; QuantaSoft (v. 1.7.4.0917); Microsoft Excel 2010; Python 3.9.7; Mathematica 10; C++14; g++ 10.3.0. Bioinformatic code for sequencing data are available at https://bitbucket.org/irccit/idea/src/master/; R (v. 3.4.4). Custom code used for the analysis are available as a repository on Mendeley Data (doi:10.17632/mvfm7hs9kw.1) |

For manuscripts utilizing custom algorithms or software that are central to the research but not yet described in published literature, software must be made available to editors and reviewers. We strongly encourage code deposition in a community repository (e.g. GitHub). See the Nature Portfolio guidelines for submitting code & software for further information.

## Data

Policy information about availability of data

All manuscripts must include a data availability statement. This statement should provide the following information, where applicable:
- Accession codes, unique identifiers, or web links for publicly available datasets
- A description of any restrictions on data availability
- For clinical datasets or third party data, please ensure that the statement adheres to our policy

Data used for the analysis, source data images of Edu staining and live microscopy assay are available as a repository on Mendeley Data (doi:10.17632/mvfm7hs9kw.1). Sequencing data are available at PRJEB49483 (ENA; https://www.ebi.ac.uk/ena/browser/home). The CRC cell clones generated in this study are available through Alberto Bardelli (Department of Oncology, University of Torino) under a Material Transfer Agreement.

# Field-specific reporting

Please select the one below that is the best fit for your research. If you are not sure, read the appropriate sections before making your selection.

☒ Life sciences          ☐ Behavioural & social sciences          ☐ Ecological, evolutionary & environmental sciences

For a reference copy of the document with all sections, see nature.com/documents/nr-reporting-summary-flat.pdf

# Life sciences study design

All studies must disclose on these points even when the disclosure is negative.

| | |
|---|---|
| Sample size | We have used a model-guided approach combined to Bayesian statistics. Within this framework, we did not perform frequentist statistical tests; hence, no considerations on minimal sample size to ensure statistical significance were needed.<br>Nonetheless, the MC-LD model was used to estimate the sample size to be used during the fluctuation test assay, in order to assure that growing colonies of pre-existing resistant clones were likely to be found in the wells after four weeks. This prior knowledge was necessary to measure mutation rates of sensitive cells. |
| Data exclusions | In the following experiments:<br>- Single-dose growth curve assay (Fig. 1d)<br>- Characterization of distribution of persister cells (Extended data fig. 6)<br>- Staining with Carboxy fluorescein succinimidyl ester (CFSE) (Extended data fig. 3)<br>- EdU staining (Extended data fig. 3)<br>plates were checked to exclude from the analysis wells containing resistant clones, with the aim to characterize cancer persister cells. |
| Replication | The number of times each experiment has been repeated with similar results is stated in each figure legend or in the methods section. |
| Randomization | No randomization, e.g. into positive vs control samples, is applicable to our experimental designs, which are based on a model-guided approach and Bayesian statistics. |
| Blinding | Manual count of viable cells was assessed by two operators in blinded fashion. |

# Reporting for specific materials, systems and methods

We require information from authors about some types of materials, experimental systems and methods used in many studies. Here, indicate whether each material, system or method listed is relevant to your study. If you are not sure if a list item applies to your research, read the appropriate section before selecting a response.

## Materials & experimental systems

| n/a | Involved in the study |
|---|---|
| ☒ | ☐ Antibodies |
| ☐ | ☒ Eukaryotic cell lines |
| ☒ | ☐ Palaeontology and archaeology |
| ☒ | ☐ Animals and other organisms |
| ☒ | ☐ Human research participants |
| ☒ | ☐ Clinical data |
| ☒ | ☐ Dual use research of concern |

## Methods

| n/a | Involved in the study |
|---|---|
| ☒ | ☐ ChIP-seq |
| ☐ | ☒ Flow cytometry |
| ☒ | ☐ MRI-based neuroimaging |

# Eukaryotic cell lines

Policy information about cell lines

| | |
|---|---|
| Cell line source(s) | WiDr and DiFi CRC cell populations were obtained by Prof. Bernards and Prof. Baselga, respectively, as previously reported in our work "Medico et al., Nature Communications 2015". JVE207 CRC cells were obtained by Dr. Wezel, Department of Pathology, Leiden, University Medical Center. |
| Authentication | The identity of each cell line was checked no more than three months before performing the experiments using the PowerPlex® 16 HS System (Promega), through Short Tandem Repeats (STR) tests at 16 different loci (D5S818, D13S317, D7S820, D16S539, D21S11, vWA, TH01, TPOX, CSF1PO, D18S51, D3S1358, D8S1179, FGA, Penta D, Penta E, and amelogenin). Amplicons from multiplex PCRs were separated by capillary electrophoresis (3730 DNA Analyzer, Applied Biosystems) and analyzed using GeneMapper v.3.7 software (Life Technologies). STR results for all the cell lines and corresponding clones matched the profiles. |
| Mycoplasma contamination | Cells were routinely screened for absence of Mycoplasma contamination using the Venor® GeM Classic kit (Minerva biolabs) and tested negative. |

# Flow Cytometry

## Plots

Confirm that:

☒ The axis labels state the marker and fluorochrome used (e.g. CD4-FITC).

☒ The axis scales are clearly visible. Include numbers along axes only for bottom left plot of group (a 'group' is an analysis of identical markers).

☐ All plots are contour plots with outliers or pseudocolor plots.

☐ A numerical value for number of cells or percentage (with statistics) is provided.

## Methodology

| | |
|---|---|
| Sample preparation | - Growth rates: CRC cells were fixed in EtOH 70% and stained with Propidium Iodide following manifacturer's instructions.<br>- CFSE staining: CRC cells were stained with CellTrace™ CFSE Cell Proliferation Kit (Invitrogen™) according to manufacturer's instructions. At the indicated timepoints, cells were collected and resuspended in 1mL PBS with Zombie Violet™ 1000x (BioLegend®) to exclude dead cells. |
| Instrument | Beckman Coulter CyAnTM ADP |
| Software | - Growth rates: Data were collected using Summit (v. 4.3) and analyzed using FlowJo (v. 7.6)<br>- CFSE staining: Data were collected using Summit (v. 4.3) and analyzed with a Python script based on standard libraries (FlowCal, FlowKit). |
| Cell population abundance | Sorting experiments were not performed |
| Gating strategy | - Growth rates: First gate: FSLin vs SSLin; Second gate: Pulse Width vs FL3Area<br><br>- CFSE staining: cells were selected with a light scattering gate (FSLin vs SS), excluding cell doublets with a single cell gate (FSArea vs SSArea). The following cutoffs were used: (i) FS Lin: lower 5000 and upper 60000; (ii) SS Lin: lower 3000 and upper 63000; (iii) FS Area: lower 3000 and upper 60000; SS Area: lower 2000 and upper 63000. We then evaluated the bi-dimensional distribution of the remaining data points in the space of the coordinates FS Area and SS Area, and retained all the data-points that were included in the 99nth percentile of the distribution. Viable cells were selected by excluding Zombie VioletTM-positive cells and CFSE signal was detected by measuring Fitc signal. |

☒ Tick this box to confirm that a figure exemplifying the gating strategy is provided in the Supplementary Information.