## [Peer Review File · Nature Genetics]

Peer Review Information

Manuscript Title: A modified fluctuation-test framework characterizes population dynamics and mutation rate of colorectal cancer persister cells

Corresponding author name(s): Professor Alberto Bardelli

Reviewer Comments & Decisions:

Decision Letter, initial version:
--

14th Jun 2021

Dear Professor Bardelli,

First of all, please allow me to apologise for the delay in returning this decision to you. Thank you so much for your patience.

Your Article entitled "Drug-induced colorectal cancer persister cells show increased mutation rate" has now been seen by 2 referees, whose comments are attached. While they find your work of potential interest, they have raised serious concerns which in our view are sufficiently important that they preclude publication of the work in Nature Genetics, at least in its present form.

While the referees find your work of some interest, their reports concerns about the overall advance provided by the work, and its clinical relevance. Taken together, it is our editorial opinion that the study has not garnered the level of reviewer support we'd expect at this stage. However, should further experimental data allow you to fully address these criticisms we would be willing to consider an appeal of our decision (unless, of course, something similar has by then been accepted at Nature Genetics or appeared elsewhere). This includes submission or publication of a portion of this work someplace else.

The required new experiments and data include, but are not limited to those detailed here. We hope you understand that until we have read the revised manuscript in its entirety we cannot promise that it will be sent back for peer review.

If you are interested in attempting to revise this manuscript for submission to Nature Genetics in the future, please contact me to discuss a potential appeal. Otherwise, we hope that you find our referees'

comments helpful when preparing your manuscript for resubmission elsewhere.

Sincerely,

Safia Danovi
Editor
Nature Genetics

Referee expertise:

Referee #1: drug resistance

Referee #2: cancer evolution and mathematical modelling

Reviewers' Comments:

Reviewer #1:

Remarks to the Author:

In this manuscript, the authors use a combination of mathematical modeling and quantitative cell culture experiments to study the basis of the emergence of drug tolerant persister cells in colorectal cancer (CRC) patient-derived cell lines under therapy with an EGFR antibody and/or a BRAF inhibitor. The main claims are that the persister cells emerging later during therapy show an increased mutation rate that contributes to therapy resistance. The findings and the approaches used are interesting. The experiments are well conducted and described appropriately. There are issues that diminish enthusiasm however.

(1) The authors seem to indicate that the persister cells are dying off slowly yet at the same proliferating slowly. This seems to lack clarity of logic. Which is it and via what mechanisms?

(2) The authors show a potentially increased mutation rate in persister cells. However, there is no causative link shown between this process and the actual phenotype of persister cell insensitivity to therapy or the emergence of full drug resistance.

(3) What are the discrete mutations causing the emergence of late/full resistance?

(4) Are there chromosomal copy number changes that are also heightened in the persister cells in addition to coding mutations?

(5) Data shown include only 2 cell line models and are all conducted in vitro. The authors should expand the number of patient-derived models and also include more clinically relevant models such as patient-derived organoids.

(6) There are no data from clinical specimens to corroborate the main findings of increased mutation rate in therapy exposed tumors. This is a major gap that limits the potential clinical relevance.

(7) Are there vulnerabilities that could be therapeutically exploited as a result of the increased mutation rate in the persister cells? If so, which are they and can the authors test at least one as proof-of-principle?

(8) Is there a causal link between inhibition of EGFR or RAF signaling and the increased mutation rate in persister cells? This should be experimentally tested here. Does the signaling pathway remain silent throughout persister development and is this required for the increased mutation rate?

(9) The authors should test another EGFR and BRAF inhibitor to exclude drug-specific effects.

(10) Is it not clear that the advance presented in this manuscript is significantly further along or novel beyond the authors previous related study so as to warrant publication in Nature Genetics.

Reviewer #2:

Remarks to the Author:

This manuscript builds on the authors' previous work showing that, in human colorectal cancer cell lines, EGFR/BRAF inhibition substantially increases the mutation rate of drug-tolerant persister cells. The main novelty of the new study is the application of mathematical modelling to infer the rates at which persisters arise and acquire drug resistance. I have focussed my review on the mathematical methods as this is my area of expertise.

The mathematical methods – and to the best of my knowledge the experimental methods – are well chosen and have been carefully applied. The results are convincing and clearly presented. The conclusions are important inasmuch as they challenge prevailing models of how cancer cells acquire resistance to targeted therapies. The big caveat is that strong evidence for this phenomenon has so far been found only in cell lines and not in patients.

Regarding this caveat, a new study in cetuximab-treated colorectal cancers casts doubt on the clinical relevance of the current manuscript's results by finding that, "despite the functional evidence for cetuximab-induced mutagenesis in CRC cell lines, our analysis in patients shows that its contribution to cetuximab resistance evolution is probably small." [Woolston et al. Nat. Ecol. Evol., <https://www.nature.com/articles/s41559-021-01470-8>]. These new clinical findings should be discussed. In particular, the current manuscript's conclusion that "The combined experimental and modeling framework presented here may have broad and far-reaching clinical implications" and the claim that the manuscript "pinpoints new strategies to restrict tumor recurrence" should be reassessed in light of the work by Woolston et al.

Besides the above, I have only minor suggestions for improvement.

Results page 13 (no line numbering): "we assumed that resistant cells divide with rate b and die with rate d , just like untreated cells." Unless I've misunderstood, this assumption is at odds with the conventional assumption that resistance typically imposes a cellular fitness cost (i.e. lower b and/or higher d) relative to sensitive cells growing in the absence of treatment. Can the authors provide evidence to support their assumption? If not, I suggest they examine whether their results are robust to adding a fitness cost (e.g. 10% lower b for resistant cells, relative to sensitive cells).

Methods page 26 (no line numbering): "Persisters cells display a moderate division rate under drug treatment ($b \approx 0.3$ days⁻¹, Extended Data Fig. 8)". It's unclear to me how this b value was calculated, nor how it relates to Extended Data Fig. 8. This evaluation of parameter b for persister cells seems to contradict a claim in lines 294-295: "accurate quantification of slow cell division in persister cells was unfeasible". Please clarify.

Methods page 29 (no line numbering): Here I find the explanation of the normalization process difficult to follow. I suggest adding an extended data figure showing some typical data before and after normalization, both to illustrate the method and to enable readers to assess how much normalization changes the data.

Methods pages 30-31 (no line numbering): "We found that the best λ model the for WiDr is the one where the transition rate to persistence is linearly proportional to the drug concentration $\lambda([M]) = k[M]$, while for the DiFi the data is best described by a TP model variant with a constant transition rate $\lambda([M]) = \lambda_0$." The finding that qualitatively different models fit data for the two different cell lines is interesting and deserves further scrutiny. Eyeballing Extended Data Figure 6b, it's tempting to hypothesize that if experiments were conducted with drug concentrations substantially above 10^{-5} then cell viability might be seen to increase in DiFi at later time points, as in WiDr. If so then, across the wider range of drug concentrations, a linear model (or a model with some other monotonically increasing function) would fit DiFi data better than a model with constant λ . Looking at it another way: had the WiDr experiments been restricted to concentrations below 10^{-5} then a model with constant λ might have been mistakenly preferred. If the authors agree that this hypothesis is plausible then they should test it with new experiments, unless it would be infeasible to do so. In any case, I suggest adding to the manuscript some discussion of whether, notwithstanding the BIC and AIC values, it is plausible that λ increases monotonically (not necessarily linearly) with $[M]$ in DiFi. This discussion could be supported by a plot illustrating the best-fitting linear model (for the current DiFi data set or, if warranted and feasible, including new results at higher drug concentrations).

Related to the above: I recommend adding some discussion of how the drug concentrations used in these experiments compare to physiological concentrations during therapy. This is important for assessing how much treatment might affect mutation rates in the clinic.

Although the manuscript is very well written, it is peppered with minor spelling and grammar mistakes. I recommend a thorough proof-reading. For example (by line number where available):

- 229: alike -> like
- 242: models -> model
- 280: other -> others
- Bottom of page 13: "the the mutation rates" -> "the mutation rate"
- 391: "alongside to increase" -> "alongside an increase"
- Page 29: concentration -> concentration
- Page 29: Averages -> Average
- Page 29: cuvers -> curves
- Page 30: indipendent -> independent
- Page 30: keepeing -> keeping
- Page 30: "the for" -> for
- Page 35: analitcal -> analytical
- Numerous instances: "persisters cells" -> "persister cells"

Author Rebuttal to Initial comments

We thank both referees for their work on our manuscript. We report below our point-by-point replies to their comments.

Referee expertise:

Referee #1: drug resistance

Referee #2: cancer evolution and mathematical modelling

Reviewers' Comments:

Reviewer #1:

Remarks to the Author:

In this manuscript, the authors use a combination of mathematical modeling and quantitative cell culture experiments to study the basis of the emergence of drug tolerant persister cells in colorectal cancer (CRC) patient-derived cell lines under therapy with an EGFR antibody and/or a BRAF inhibitor. The main claims are that the persister cells emerging later during therapy show an increased mutation rate that contributes to therapy resistance. The findings and the approaches used are interesting. The experiments are well conducted and described appropriately. There are issues that diminish enthusiasm however.

We thank the reviewer for showing interest in our work and for appreciating our experimental design.

(1) The authors seem to indicate that the persister cells are dying off slowly yet at the same proliferating slowly. This seems to lack clarity of logic. Which is it and via what mechanisms?

We thank the referee for raising this point. In the previous version of our manuscript, experimental data based upon CFSE measurements and the single-dose assay already supported the notion that persisters undergo both slow proliferation (shown by the dilution of the CFSE dye) and cell death (given the net decrease that we observe in cell number during prolonged drug treatment in single-dose drug assay).

To better elucidate persisters dynamics and cell death mechanisms during drug treatment we performed additional experiments.

We initially quantified the fraction of proliferating persisters by fluorescently labeling surviving cells with EdU (5-ethynyl-2'-deoxyuridine), a modified thymidine analogue that is efficiently incorporated into DNA during active DNA replication. Quantification of EdU positive cancer persister cells after 14, 17 and 20 days of treatment unveiled a fraction of replicating persisters ranging between 0,2 and 2,5 % (Rebuttal Figure 1, Extended Data Figure 6 in the revised manuscript).

Rebuttal Figure 1. A fraction of cancer persister cells slowly replicate under constant drug treatment. (a) DiFi and WiDr CRC clones were treated with anti-EGFR inhibitor cetuximab alone (DiFi) or in combination with dabrafenib (WiDr) for 2 weeks. After that, surviving residual cells were labelled with EdU at indicated time points for 4 hours, then fixed and analyzed by fluorescent microscopy. (b) Quantification of EdU positive cells at indicated time points. EdU-positive and total number of cells (based on nuclei staining with DAPI) were quantified using the “Analyze particles” function in ImageJ. A minimum of 200 cells for each timepoint were analyzed. Results represent average of two independent experiments, each represented by two technical replicates. (c) Representative images of EdU-positive persister cells for each CRC clone analyzed.

We then used the Nikon LIPSI live imaging system to set up a time-lapse microscopy assay. In detail, we treated both DiFi and WiDr clones for 2 weeks with anti-EGFR inhibitor cetuximab alone or in combination with BRAF inhibitor dabrafenib, respectively, until the emergence of persister cells. Labeling surviving persister cells with fluorescent dyes allows an accurate and sensitive detection of cell division (by staining of nuclei with NucBlue®) and cell death (with CellEvent™, a live Caspase3/7 activation fluorescent dye) within persisters subpopulations across several days of drug treatment.

Although the majority of CRC persister cells did not replicate, sparse cell division events were evident for persisters from all the CRC clones analyzed (Rebuttal Figure 2, Extended Data Figure 6 in the revised manuscript, and Extended data Movies in the revised manuscript). In some instances, cell division was successful (Rebuttal Figure 2a and b), while in other cases cells died concomitantly (Rebuttal Figure 2c). In parallel, we detected cell death, as highlighted by cells positive for the CellEvent signal either after a cell division event, or in non-dividing cells (Rebuttal Figure 2c and d, respectively). The imaging-based data was also used to estimate (roughly due to the small statistics) the effective growth rate of persister cells, which is compatible with the values obtained by the TP model inference (Rebuttal Figure 2e).

In summary, live monitoring of CRC persister cells confirms that cancer persister populations are slowly dying, while a fraction of cells is slowly replicating, therefore validating our previous findings.

Rebuttal Figure 2. Cell division and death of CRC persister cells can be directly visualized by microscopy. CRC cells were seeded in 24-multwell plates and treated for 2 weeks with anti-EGFR inhibitor cetuximab alone (DiFi) or in combination with BRAF inhibitor dabrafenib (WiDr). After 2 weeks of constant drug treatment, surviving persister cells were stained with Nucblue® (a dye for nuclei, in blue) and Caspase Cell EventTM (a live marker for the activation of Caspase3/7, in green) while maintaining drug pressure, and monitored for 5 days under an inverted widefield microscope (Nikon Lipsi, 20X Plan Apo objective with 0.75 NA) acquiring images every 45 minutes. For each clone we were able to follow 2 technical replicates, with 16 fields of view each, for a total of more than 1300 cells for each condition. Representative snapshots of cell division events and apoptotic events are reported, while supplementary movies report whole time-lapse experiments for representative fields of view for each clone. (a-b) cell division events; (c) cell death through apoptosis after a cell division; (d) cell death through apoptosis in non-dividing cells. (e) Numerical values of the effective growth rates of persister cells evaluated by means of imaging-based assays and by TP model inference are compatible. The scatter plot compares the values of the effective growth rate (b-d) obtained with image segmentation (x axis) and inferred with the TP model from dose-response and single dose assays (y axis). Squares correspond to center values and bars to standard deviations. The blue line marks the region of a perfect match between the two values.

(2) The authors show a potentially increased mutation rate in persister cells. However, there is no causative link shown between this process and the actual phenotype of persister cell insensitivity to therapy or the emergence of full drug resistance.

We thank the reviewer for the constructive criticism. We speculate that the adaptive mutability process observed in response to targeted therapy is not the origin of the persisters drug-tolerance phenotype, but a conserved mechanism of adaptation to treatment-induced stress (previously described in bacteria) that becomes relevant in surviving persister cells, which have lost multicellularity homeostatic controls as it happens for tumor cells (Russo et al., *Cancer Discovery* 2021).

Crucially, a causative link between mutation rate in persisters and emergence of drug resistance is established in this study: persisters must show accelerated mutation rates, or the timeframe at which persisters-derived resistant clones emerge in our experimental settings would not be justified.

This is supported by the following calculation based on our results. Persister-derived resistant clones are observed after >10 weeks of drug exposure. Since their mutation rate is 10-100x larger than (proliferating) untreated cells, in the absence of an increased mutation rate it would typically take (in a conservative estimate) $10 \times (10-100) \sim 100-1000$ weeks for a well to develop resistance based on the mutation rate of sensitive (untreated) cells that we experimentally calculated, or equivalently it would take $2 \times 10^5 - 2 \times 10^6$ wells to observe a few resistant clones after 10 weeks. We have clarified this point in the revised manuscript.

(3) What are the discrete mutations causing the emergence of late/full resistance?

Following the referee's question, we performed whole genome sequencing (WGS) and droplet digital PCR (ddPCR) on persister-derived resistant cells isolated from the MC-LD fluctuation experiments in order to characterize the molecular drivers of drug resistance.

As reported in the tables below the new experiments unveiled both single nucleotide variants (SNV) and copy number variation (CNV) as mechanisms of resistance in persisters-derived resistant clones. These drivers were among the known mechanisms of resistance to MAPK pathway inhibition in CRC previously described (Misale et al., *Nature* 2012; Misale et al, *STM* 2014; Arena et al., *Clin Cancer Res* 2015).

DiFi clones acquired either KRAS gene amplification (Rebuttal Figure 3, Extended Data Figure 16 in the revised manuscript), or different KRAS and NRAS mutations (Rebuttal Table 1, Extended Data Table 6 in the revised manuscript), while WiDr cells unveiled heterogeneous mutations in RAS genes (Rebuttal Table 1). We now discuss the meaning of these findings in the main text.

Rebuttal Figure 3. GCN analysis of DiFi persisters-derived resistant cells. Copy number variation (CNV) analysis of DiFi resistant cells derived from persisters within the MC-LD experimental setting. Upper panel shows the CNV differential analysis of two different parental (sensitive, untreated) clones. CNVs of three independent persister-derived resistant clones (namely clones 44, 46, 48) vs the parental counterpart were reported in the lower panels. Arrows indicate the KRAS amplification acquired at resistance to targeted therapy in all the 3 clones analyzed.

CRC cell line	Sample	Target	Fractional Abundance (%)
WiDr	Persister-derived resistant cells_1	KRAS_G12D	0.27
		KRAS_G12V	1.58
		KRAS_G13D	0.2
WiDr	Persister-derived resistant cells_2	KRAS_G12D	0.31
		KRAS_G12V	1.72
		KRAS_G13D	0.72
WiDr	Persister-derived resistant cells_3	KRAS_G12V	3.6
		KRAS_G13D	0.9
WiDr	Persister-derived resistant cells_4	KRAS_G12D	2.4
		KRAS_G12V	7
		KRAS_G13D	1.8
WiDr	Persister-derived resistant cells_5	KRAS_G12D	0.55
		KRAS_G12V	0.65
WiDr	Persister-derived resistant cells_6	KRAS G12C	1.23
		KRAS G12D	5.6
		KRAS G12V	7.6
		KRAS G13D	5.1
WiDr	Persister-derived resistant cells_7	KRAS G12C	0.47
		KRAS G12D	0.73
		KRAS G12V	4.4
WiDr	Persister-derived resistant cells_9	KRAS_G12D	0.28
		KRAS_G12V	1.13
		KRAS_G13D	0.29
WiDr	Persister-derived resistant cells_8	KRAS_G12V	0.73
WiDr	Persister-derived resistant cells_11	KRAS_G12V	3
DiFi	Persister-derived resistant cells_1	KRAS G12D	0.24
		KRAS Q61m	0.25
DiFi	Persister-derived resistant cells_2	KRAS G12D	4.9
		KRAS G13D	0.94
		KRAS Q61m	0.19
DiFi	Persister-derived resistant cells_3	KRAS G12D	1.3
		KRAS G12V	0.41
DiFi	Persister-derived resistant cells_5	KRAS G12D	0.44
		KRAS G12V	1.58

Rebuttal Table 1. Mutational analysis of persisters-derived resistant cells isolated from MC-LD experiment. Table lists the relative fractional abundance of mutated alleles detected by droplet digital PCR (ddPCR) analysis of indicated resistant cells derived from WiDr or DiFi persisters within the MC-LD experimental setting.

(4) Are there chromosomal copy number changes that are also heightened in the persister cells in addition to coding mutations?

Please refer to our answer to the previous point above.

(5) Data shown in include on only 2 cell line models and are all conducted in vitro. The authors should expand the number of patient-derived models and also include more clinically relevant models such as patient-derived organoids.

Following the suggestion, we replicated the full set of experiments and ran the analysis pipeline for two additional clones, one for each CRC cell line, called WiDr cl. B5 and DiFi cl. B3 (Rebuttal Figures 4 and 5, respectively; Extended Data Figures 14 and 15 in the revised manuscript, respectively), to exclude any clonal bias.

In both cases we found agreement with a drug-induced scenario for the persister transition (Rebuttal Figures 4a-b and 5a).

Considering WiDr cl. B5 we did not find evidence for a decline of persister cells over time ($D_p \approx 0$, Rebuttal Figure 4b), while we observed a decline of DiFi cl. B3 persister cells over time ($D_p > 0$, Rebuttal Figure 5a).

We found that WiDr cl. B5 is best described by a TP model with a transition rate proportional to the drug concentration (Rebuttal Figure 4c-d), consistently with what we observed for WiDr cl. B7. Similarly, for DiFi cl. B3 the best TP model variant is the one with a constant transition rate (Rebuttal Figure 5b-c), consistently with what we observed for DiFi cl. B6.

Notably, both clones display an increased mutation rate of persister cells (Rebuttal Figures 4e and 5d). Both transition rate to persisters and increased mutation rate did not vary when considering different values of the initial fraction of persisters f_0 within the range of values compatible with the dose-response and single dose assay (Rebuttal Figures 4 c,f and 5 b,e), thereby confirming our previous findings and excluding a clonal bias.

Rebuttal Figure 4. The analysis of an additional WiDr clone yields consistent results. The plots show the results of the TP model and MC-LD model inference for populations derived from an additional WiDr clone, namely WiDr cl. B5. (a) Doses-response dataset and best TP model fit as a function of the initial fraction of persister cells f_0 . (b) Single-dose dataset, best TP model fit and expected fraction of persister cells. (c) Joint posterior distribution of the TP model for the initial fraction of persisters and the transition rate. (d) Experimentally measured dose-responses growth curves (symbols) and best TP model fit (continuous lines). The doses-response datasets were normalized to the growth of the untreated cells using the best model parameters. (e) Quantification of mutation rates for sensitive (red) and persister (blue) cells in the MC-LD experiment. (f) Joint posterior distribution of the MC-LD model for initial fraction of persisters and the fold increase of the mutation rate of persister cells.

DiFi cl. B3 ◆

Rebuttal Figure 5. The analysis of an additional DiFi clone yields consistent results. The plots show the results of the TP model and MC-LD model inference for populations derived from an additional DiFi clones, namely DiFi cl B3. (a) Doses-response dataset and best TP model fit as a function of the initial fraction of persister cells. (b) Joint posterior distribution of the TP model for the initial fraction of persisters and the transition rate. (c) Experimentally measured dose-responses growth curves (symbols) and best TP model fit (continuous lines). The single-dose datasets were normalized to the growth of the untreated cells using the best model parameters. (d) Quantification of mutation rates for sensitive (red) and persister (blue) cells in the MC-LD experiment. (e) Joint posterior distribution of the MC-LD model for initial fraction of persisters and the fold increase of the mutation rate of persister cells.

We have considered the reviewer's suggestion to expand the analysis to patient-derived organoids. Unfortunately deploying the MC-LD fluctuation assay on patient-derived organoids (PDOs) is currently not conducive for the following reasons. Within the MC-LD setup, proliferation parameters and mutation rate are quantified precisely, and to this aim cancer cells must be expanded in twenty 96-well plates and kept under constant treatment for more than 12 weeks. Differently from 2D cell lines, the expansion of 3D growing PDOs in Matrigel does not allow precise quantification of population-dynamics parameters and reaching the sufficient number of cells to seed twenty 96-well plates could be prohibitive. Additionally, culturing of PDOs requires continuous (weekly) refresh of media and Matrigel (through centrifugation of PDOs and Matrigel, removal of Matrigel and seeding in fresh Matrigel) and this is not currently compatible with the MC-LD protocol.

We underline that the level of quantitative control we developed to perform and interpret the MC-LD experiment is unprecedented in mammalian cells, and that controlling all the parameters requires a suitable model system.

We understand that this point is at the same time an important limitation of our approach. In the revised manuscript, we have tuned all our claims, in order to make this limitation explicit, and at the same time we have clarified its broad potential for applicability.

While PDOs are (currently) not conducive for our purpose for the reasons above, it is potentially feasible to replicate the MC-LD fluctuation test in cell lines directly derived from CRC patients.

To this end, we attempted to replicate the TP pipeline using our molecularly annotated collection of 2D primary cell models (Russo et al., *Canc Discov* 2016; Siravegna et al., *Cancer Cell* 2018; Lazzari et al., *Clin Canc Res* 2019). We found that these CRC lines are still challenging to handle, but could in the future be exploited to set up MC-LD experiments.

These experiments still require several months to complete the initial set up. The initial results obtained with a RAS/BRAF wildtype CRC patient-derived primary cell line, sensitive to anti-EGFR cetuximab, are reported in Rebuttal Figure 6. We observed a bi-phasic killing curve at high drug concentration, hence supporting the presence of a transition to persister cells, similar to the one observed in WiDr and DiFi clones. The full results will be the subject of a follow up story.

Rebuttal Figure 6. CRC patient-derived primary cell line shows a bi-phasic killing curve in response to cetuximab treatment. We show here data collected in the dose-response assay (blue dots) of a RAS/BRAF wildtype CRC patient-derived primary cell line, namely CRC0078, for three different cetuximab concentrations (values indicated in red above each plot). The blue lines show the model expectation if no transition to persistence would occur. The lines do not capture the trend of the collected data for long times of treatment (>12 days). Hence, the collected data suggest the presence of a transition to persister cells.

(6) There are no data from clinical specimens to corroborate the main findings of increased mutation rate in therapy exposed tumors. This is a major gap that limits the potential clinical relevance.

We conceived this study to develop a comprehensively controlled assay to monitor multiple parameters and measure mutation rate changes during therapeutic treatment of tumor (and other) mammalian cells. The current manuscript reflects this and provides an experimental strategy that quantifies all the relevant parameters and provides a fully controlled framework to study persister population dynamics in human tumor cells. The approach we developed can be exploited in future studies, extend beyond cancer and used broadly to measure mutation rates in mammalian cells undergoing therapeutic stress.

We appreciate the suggestions to extend the analysis to clinical specimens. However, the tumor genetic heterogeneity in patients or in deriving in vitro models makes the increased mutations rate very difficult to detect, especially when resistance already occurred. Indeed, by using standard NGS-based approaches, we previously showed that tumor mutational burden is not significantly increased in cells that acquired resistance to targeted therapy after undergoing the persister phase (Russo et al., *Science* 2019).

Notably, the adaptive mutability phenotype is likely restricted in time (i.e., when the cells adapt to the new environment, through the acquisition of permanent mechanisms of resistance, the phenotype reverts) and confined to a subfraction of cells that would be difficult to capture with bulk analysis, especially once resistance has fully blown.

Based also on this, while tissue biopsy is usually collected from tumor lesions that relapsed upon therapy, ideally, one should be able to collect longitudinally minimal residual disease (MRD), resembling surviving persisters in vitro, at different time points to be able to measure the mutation rate in patients during therapeutic treatment, but this is clearly unfeasible.

Therefore, characterizing an increased mutation rate in clinical specimens collected from patients that progressed upon treatment with targeted therapies after an initial clinical response might be challenging, or even unfeasible.

Notably, Cipponi and colleagues recently reported WGS analysis on single clones-derived populations obtained from cancer cell lines during the early phase of the resistance development (i.e., when cells are not yet adapted to drug-induced stress), showed increased intra-clonal diversity (measured as the number of de novo SNVs) in resistant cells (Cipponi et al., Science 2020). We exploited the whole-genome sequencing performed by Cipponi and coworkers on clone-derived populations obtained from melanoma and liposarcoma cell lines untreated and in the early phase of the resistance development, to quantify mutation rates using our modified Luria-Delbruck (MC-LD) model. The analysis is reported below in the reply to point 2 of reviewer #2 below.

What is important to stress here, is that the estimated mutation rates of treated cells in their experiments are only mildly higher for treated cells, because of the less controlled setup as well as the fact that they do not keep into account the role of persisters (which give rise to many late-emerging resistance mutations from a very small residual population, a fact that can be explained only with a strongly increased mutation rate). This is a further proof of how the detection of an increased mutation rate may be difficult even in highly controlled systems.

(7) Are there vulnerabilities that could be therapeutically exploited as a result of the increased mutation rate in the persister cells? If so, which are they and can the authors test at least one as proof-of-principle?

We thank the referee for raising the point. We agree this is potentially an exciting implication of our results and we are indeed pursuing different approaches aiming at targeting the players of adaptive mutability in cancer cells to possibly delay, or even prevent, the occurrence of secondary resistance. Our previous findings unveiled a switch from high to low fidelity DNA replication process through downregulation of MMR and HR DNA repair genes and upregulation of specialized error-prone DNA polymerases in presence of increased DNA damage in response to treatment with targeted therapies (Russo et al., Science 2019).

Accordingly, inhibition of DNA polymerases might be exploited as novel combinatorial treatment strategy in tumor cells.

Among the DNA polymerases that we found upregulated in cancer cells upon targeted therapy-induced stress response, REV1 carries out the mutagenic translesion synthesis (TLS), a process that allows cells to tolerate DNA damage by bypassing lesions that block normal DNA replication, resulting in the introduction of mutations. Indeed, the inhibition of TLS would likely increase the cytotoxic effects of DNA damaging agents and potentiate their effectiveness, especially in cancer cells that are defective in DNA repair. Importantly, inhibition of TLS activity might prevent drug resistance caused by the mutagenic replication of DNA. Interfering with TLS with a REV1 small molecule inhibitor has been shown to enhance chemotherapy efficacy by suppressing tumor growth both in vitro and in vivo (Wojtaszek et al., Cell. 2019; Chatterjee et al., PNAS 2020).

Based on this, we performed initial tests of the REV1 inhibitor (REV1i) in combination with MAPK pathway in a 5-days drug response assay. We found that REV1i does not enhance the acute effect of MAPK pathway inhibition in reducing cell viability (Rebuttal Figure 7a).

Next, we performed a Time-To-Progression (TTP) assay we previously established (Russo et al., Nat Comm 2018) to monitor the development of secondary resistance in cancer cells. In detail, we treated MSS DiFi and WiDr CRC cells with either MAPK pathway inhibitor, or REV1 inhibitor or their combination. As shown in Rebuttal Figure 7 below, REV1 inhibitor significantly delayed (in case of WiDr) or even prevented (in case of DiFi) the development of secondary resistance to EGFR/BRAF inhibitors (Rebuttal Figure 7b), therefore confirming our hypothesis.

Although we find these results exciting and promising, further characterization of the mechanism(s) of action of this and other inhibitors are needed, and this is part of our future experiments. We believe this kind of studies are beyond the purpose of the present work, whose scope is to introduce and deploy the fluctuation assay setup to quantify the persister transition and measure for the first time mutation rate under therapy in cancer cells. Accordingly, we have not included the combinatorial results in this manuscript, but wanted to present them as preview to the reviewers since these new data suggest that the MC-LD assay can lead to new discoveries with translational relevance.

Rebuttal Figure 7. Inhibition of error-prone DNA polymerase REV1 delays the onset of secondary resistance to MAPK pathway inhibition in CRC cells. (a) WiDr and DiFi CRC cells were treated with indicated drugs for 5 days. After that, cell viability was measured by the ATP assay. (b) WiDr and DiFi CRC cells were treated in parallel with MAPK pathway inhibitors (anti-EGFR inhibitor cetuximab alone in the case of DiFi, or in combination with anti-BRAF inhibitor dabrafenib in the case of WiDr), REV1 inhibitor or their combination and number of cells was monitored during the treatment, until the emergence of resistance.

In parallel, in the revised manuscript we are now providing proof-of-principle computational evidence that the drug-dependent transition to persisters can be therapeutically exploited.

The best TP model variant that describes the observed dynamics for WiDr cells is the one with a transition rate to persistence that is linearly dependent on drug concentration $\lambda([M]) = k [M]$ (see below). This particular functional dependence corresponds to a scenario where higher drug concentrations induce both an increased death rate of sensitive cells, which is desirable, but, at the same time, an increased transition to the persister state, which act as a reservoir for the emergence of resistance. We show here that, by modulating the drug concentration over time, this trade off can be exploited to reduce the number of persisters cells emerging in response to the therapeutic drug.

To illustrate this point, we considered two alternative drug delivery strategies: (i) the concentration of the drug is maintained constant over time; (ii) the concentration of the drug is linearly increased over time. In order to define a fair comparison between the two strategies, we constrained this second strategy to deliver the same average drug concentration as the first one, meaning that the total amount of drug delivered is the same for the two approaches. The choice of the two strategies is shown in Rebuttal Figure 8 a,b below (Extended Data Figure 17 in the revised manuscript).

We solved the TP model (eq.1) numerically, using these two alternative choices for the dependence of the drug concentration over time as model inputs, and compared the two outcomes (Rebuttal Figure 8 a-b). The other model parameters were set to the experimentally derived values for WiDr cl. B7 (see Extended Data Table 4 in the revised manuscript). In order to investigate the worst-case scenario, i.e., the one with the maximal number of emergent persister cells, we neglected their death ($D_p=0$). We found that a drug delivery strategy with a linearly increasing drug concentration can reduce the total number of emergent persister cells (Rebuttal Figure 8d). Interestingly, with this strategy the drug concentration is also kept under the minimum inhibitory concentration (MIC) value at the beginning of the treatment, suggesting that one could design therapies that are less aggressive and also more effective.

Finally, we note that this setup would produce resistance much slower in a MC-LD setup, as it greatly reduces the number of persisters (up to 40% less) (Rebuttal Figure 8d).

Rebuttal Figure 8. The dependence on the drug concentration of the transition rate to persistence can be exploited to reduce the rate of emergence of persisters. (a) The plot illustrates a standard drug delivery strategy, which consists of a constant drug concentration delivered over time. The values for the concentration of dabrafenib and cetuximab are realistic for colorectal cancer cells similar to WiDr cl. B7. The drug concentration of cetuximab is also assumed constant over time. The value of the Minimum Inhibitory Concentration (MIC, dashed black line) is the minimal value of the drug concentration for which a population of sensitive cancer cells would not increase over time. For the case of WiDr cl B7, the value of the MIC can be obtained from our estimated model parameters, $\text{MIC} = a-1 = 3.5 \times 10^{-7} \mu\text{M}$ (see Extended Data Table 4 in the revised manuscript). (b) In a modified drug delivery strategy, the drug concentration of dabrafenib increases linearly over time, and the drug concentration of cetuximab is also assumed constant over time. This strategy is constrained to have the same average drug concentration as in panel (a), meaning that the total amount of drug delivered in 10 days is the same in the two strategies. In this case, at the beginning of the in-silico treatment, the delivered drug is below the MIC value. (c) Expected number of persister cells predicted by the TP model (eq. 1) for WiDr cl. B7 cells, corresponding to the two alternative drug delivery strategies: constant value (gray line) and linear increase (orange line). (d) Relative fold change of the number of persister cells emerged with the drug strategy B vs drug strategy A, evaluated as a function of the average drug concentration.

(8) Is there a causal link between inhibition of EGFR or RAF signaling and the increased mutation rate in persister cells? This should be experimentally tested here. Does the signaling pathway remain silent throughout persister development and is this required for the increased mutation rate?

We thank the referee for this comment. In our previous study (Russo et al, Science 2019) we showed that the adaptive mutability phenotype is not a drug-dependent side-effect, but rather it is connected to the perturbation of oncogene addiction in cancer cells.

Indeed, downregulation of EGFR and KRAS in EGFR-amplified DiFi cells, or of EGFR and BRAF in BRAF-mutated WiDr cells through siRNA recapitulates the same phenotype of DNA repair impairment observed upon treatment with targeted therapies (Rebuttal Figure 9, left part; reproduced from Russo et al., Science 2019, Fig. 3e). Differently, other cellular stresses such as cell cycle arrest or direct DNA damaging agents did not recapitulate the phenotype, therefore suggesting that the interruption of oncogene addiction initiate adaptive mutability-stress response, leading in turn to genetic instability (Russo et al., Science 2019).

Additionally, after more than two weeks of constant drug pressure, DNA repair downmodulation is maintained in persisters together with inhibition of the MAPK pathway, thus supporting the concept that the signaling pathway

remains silent throughout the persister phase (Rebuttal Figure 9, right part; reproduced from Russo et al., Science 2019, Fig. S4B).

Rebuttal Figure 9 (modified from Russo et al., Science 2019, Fig.3E and S4B). Perturbation of oncogene addiction initiates adaptive mutability stress response in CRC cells. (e) Wt DiFi (left panel) and BRAF mutated WiDr (right panel) cells were transfected with indicated siRNA or combinations of them. After 72 hours protein lysates were analyzed by western blot. All star, non-targeting siRNA. (b) Expression levels of MMR and HR DNA repair proteins were evaluated in 2 independent persister populations (P1 and P2) for each cell line model. NT stands for untreated cells.

(9) The authors should test another EGFR and BRAF inhibitor to exclude drug-specific effects.

In this study, we used two CRC cell lines that are dependent upon distinct oncogenic events (EGFR amplification and BRAF mutation) and for which we have already shown that adaptive mutability is not dependent on the use of a specific drug, as it can be replicated by siRNA-mediated downregulation of the oncogenic pathways the cell line is addicted to, based on the genetic milieu (please see Rebuttal Figure 9 above), therefore excluding a drug-specific effect.

(10) Is it not clear that the advance presented in this manuscript is significantly further along or novel beyond the authors previous related study so as to warrant publication in Nature Genetics.

We thank the referee for this comment, and we apologize if these aspects did not emerge clearly from the manuscript. The scope of this work is to provide for the first time, to the best of our knowledge, a fluctuation assay for a mammalian cell system under treatment with targeted therapies, where quantitative precision can be achieved, and where mathematical models for the persisters transition and the mutation rate can be validated. The two main findings that were not demonstrated in the Russo et al. 2019 study are: (i) the transition to persister is mainly drug-induced and (ii) persisters quantitatively increase their mutation rate.

Here, for the first time we quantitatively measured the mutation rate of cancer cells during drug treatment, formally demonstrating that the previously unveiled stress response induced by targeted therapy (Russo et al., Science 2019) actually translates into 7-50 fold increase of mutation rate in surviving persister cells. Importantly, this was possible only by coupling experimental characterization of cancer cells, evaluation of multiple parameters of cancer cell population dynamics during drug exposure and mathematical modelling.

Notably, the mathematical framework coupled to the experimental assays presented here could be applied to unveil and quantify the impact of a wide range of environmental conditions on the persister phenotype and mutation rates in mammalian cells.

Moreover, to the best of our knowledge a quantitative definition of cancer persisters was lacking together with several aspects of their population dynamics, including whether they are pre-existing in a cancer population prior to treatment and whether they display proliferation and cell death. As highlighted by referee #2, for the first time

in human cancer cells we deployed rigorous and controlled experimental designs and mathematical modeling to track persisters dynamics over time.

We have modified the manuscript to present our achievements in light of our previously published results, and to highlight their relevance and the advances they provide.

Reviewer #2:

(1)

Remarks to the Author: This manuscript builds on the authors' previous work showing that, in human colorectal cancer cell lines, EGFR/BRAF inhibition substantially increases the mutation rate of drug-tolerant persister cells. The main novelty of the new study is the application of mathematical modelling to infer the rates at which persisters arise and acquire drug resistance. I have focussed my review on the mathematical methods as this is my area of expertise.

The mathematical methods – and to the best of my knowledge the experimental methods – are well chosen and have been carefully applied. The results are convincing and clearly presented. The conclusions are important inasmuch as they challenge prevailing models of how cancer cells acquire resistance to targeted therapies. The big caveat is that strong evidence for this phenomenon has so far been found only in cell lines and not in patients.

We thank the reviewer for highlighting the novelty of our approach and the implications of our work.

As explained in response to Reviewer #1 above, detection of increased mutation rate in clinical specimens collected from patients that progressed upon treatment with targeted therapies after an initial clinical response might be challenging. While tissue biopsy is usually collected from tumor lesions that relapsed upon therapy, we should instead have access to minimal residual disease (MRD) (most probably due to surviving persister cells) at different time points during therapy. As the reviewer certainly appreciate this unfortunately remains technically very challenging. We are considering developing a clinical protocol, to intercept these types samples but so far, we have not been successful, also in light of the clinical and ethical feasibility issues.

Notably, we underline again that the level of quantitative control of the MC-LD experiment is unprecedented in mammalian cells, and achieving this control requires a simple system. We have clarified these points in the revised manuscript.

(2)

Regarding this caveat, a new study in cetuximab-treated colorectal cancers casts doubt on the clinical relevance of the current manuscript's results by finding that, "despite the functional evidence for cetuximab-induced mutagenesis in CRC cell lines, our analysis in patients shows that its contribution to cetuximab resistance evolution is probably small." [Woolston et al. Nat. Ecol. Evol., <https://www.nature.com/articles/s41559-021-01470-8>]. These new clinical findings should be discussed. In particular, the current manuscript's conclusion that "The combined experimental and modeling framework presented here may have broad and far-reaching clinical implications" and the claim that the manuscript "pinpoints new strategies to restrict tumor recurrence" should be reassessed in light of the work by Woolston et al.

We thank the reviewer for raising this important point. We have reformulated and clarified our claims. A crucial point is that the tumor genetic heterogeneity in patients or in deriving in vitro models (such as PDOs, for which the fluctuation assay setup cannot be applied) makes the increased mutations rate very difficult to detect, especially when resistance already occurred.

Tackling this issue is part of our long-term goals and we have launched an extensive study to monitor sequencing mutation rates in patient-derived xenografts (PDXs) by sequencing; however, this approach, which is briefly highlighted below, will require many months to complete.

We have read with great interest the work of Woolston and colleagues, and have discussed the findings in the revised manuscript. Indeed, the sequencing analyses performed by Woolston and colleagues are in line with previous reports showing that tumor mutational burden is not significantly increased in cells that acquired resistance to targeted therapy after undergoing persister phase, including our own analysis in Russo et al., Science 2019.

The crucial point is that without a controlled setup these changes are very difficult to detect. To explain this problem, we can use a recent analysis by Cipponi and coworkers showing an increased intra-clonal diversity (number of sub-clonal SNVs) in melanoma and liposarcoma cell lines that developed resistance in vitro, as

compared to their untreated counterpart consistent with increase of mutation rate as the authors suggested (Cipponi et al., Science 2020).

We have taken advantage of the whole-genome sequencing data from this work, performed on single cell clones-derived populations obtained from melanoma and liposarcoma cell lines untreated or in the early phase of the resistance development, to quantify mutation rates applying our modified Luria-Delbruck (MC-LD) model.

The generalized estimator we presented in our work in Eq.s (2-4) can be modified to describe the emergence of sub-clonal mutations, by associating:

- (i) the number of wells developing resistance  number of base sites with a sub-clonal mutation
- (ii) total number of wells  total genome length ($L = 3.2 * [10]^9$ sites)

The mathematical description as well as further details about the extension of our modelling to the analysis of sequencing data are active subjects of research in our group. While the derivation and the validation of this model are beyond the scope of this study, here, we report the mathematical definition of the estimator which we have applied to the data we retrieved from Cipponi et al., Science 2020.

The estimator of the mutation rate to be applied to sequencing data takes the form

$$\mu = -\text{Log}(1 - f) / \mathcal{M}$$

Where $f = N_{\text{sub}}/L$ is the fraction of genome base sites with an observed sub-clonal mutation, and \mathcal{M} are the total number of cell divisions in which the N_{sub} mutations have emerged. Of note, this mutation rate is per generation, and not chronological.

We have retrieved the numerical values of N_{sub} corresponding to the clones investigated in Cipponi et al., Science 2020 (Table S3, entry "Bozic Adjusted"). The numerical value of \mathcal{M} is set by the depth of the sequencing used. For the genomic data considered we have $\mathcal{M} \approx 15$, which is based on the sequencing features to detect mutation with an intra-population frequency $>1/16$. Hence, all the observed sub-clonal mutations must have emerged in the first 4 generations.

We find that (Rebuttal Figure 10),

- (i) estimated mutation rates are in general agreement with other estimates (point mutation rate around 10^{-9} per generation);
- (ii) resistant clones show a systematic increase of the mean value of the mutation rate, although this enhancement is not as strong as in our data;
- (iii) silencing of MTOR induced an enhancement of the mutation rate with respect to control as expected (Cipponi et al., Science 2020,).

Rebuttal Figure 10. Values of the mutation rate estimated from genetic data reported in Cipponi et al., Science 2020. We show here box-whisker plots (mean: red line, box: 1st and 3rd quartile) representing the distribution of mutation rates inferred from Cipponi et al., Science 2020 and for indicated clones (SKMEL28: human melanoma cell line; 94T778: human liposarcoma cell line). NS, shRNAs: 94T778 cells engineered with a nonsilencing, control short hairpin RNA; MTOR, ShRNAs; 94T778 cells engineered with mTOR-silencing short hairpin RNA).

It is important to stress that the estimated mutation rates of treated cells from the Cipponi et al. data are only mildly higher as compared to their untreated counterpart. The reason of this are the limitations of the experimental setup employed in this study, compared to ours. Specifically, in Cipponi et al., resistance developed from a cancer

cell population of decreasing and not-well-defined size. Moreover, in their setup they do not separate pre-existing and newly-developed resistant cells as we do: the outgrowth of pre-existing resistant cells represents a strong confounding effect on the slower development of resistance from persister cells, and therefore could mask an increase in mutation rate in residual persister cells. Moreover, the authors do not keep into account the role of persisters in their experimental setup.

In brief, in our experimental setup persisters give rise to unexpectedly many late-emerging resistance mutations from a very small residual population, a fact that can be explained only with a strongly increased mutation rate, but the detection of an increased mutation rate would be very difficult in any “normal” situation where these mutations are masked by pre-existing resistant clones.

Besides the above, I have only minor suggestions for improvement.

(3)

Results page 13 (no line numbering): “we assumed that resistant cells divide with rate b and die with rate d , just like untreated cells.” Unless I’ve misunderstood, this assumption is at odds with the conventional assumption that resistance typically imposes a cellular fitness cost (i.e. lower b and/or higher d) relative to sensitive cells growing in the absence of treatment. Can the authors provide evidence to support their assumption? If not, I suggest they examine whether their results are robust to adding a fitness cost (e.g. 10% lower b for resistant cells, relative to sensitive cells).

We thank the referee for her/his suggestion. The estimators developed for the fluctuation assay only depend on b - d in the case of persisters, so that our findings are robust in terms of the detailed values of these parameters.

Following the reviewer’s suggestion, we have investigated the stability of the inferred values of the mutation rates against variation of the division rate of resistant cells. To do so, we considered mutation rate estimators computed assuming a value of the division rate of resistant cells of the form $b_{res} = [(1 - f)]_{cost} * b$, where b is the division rate of the sensitive cells, and f_{cost} is the fitness cost of resistance, which we have varied in the region $[0, 10\%]$. The result of this analysis is shown in Rebuttal Figure 11 below (Extended Data Figure 13 in the revised manuscript).

We find that both the mutation rate of the sensitive cells and the fold increase of the mutation rate of persisters cells display a very mild variation on the fitness cost.

Rebuttal Figure 11. The inferred values of the mutation rate are stable against the variation of the value of the fitness cost of resistant cells. (a-b) Inferred values of the mutation rate of sensitive cells as a function of the fitness cost of resistant cells for WiDr cl. B7 (a) DiFi cl. B6 (b). The variation of the estimated mutation rates is very small. (C-D) Inferred values of the fold increase of the mutation rate of persister cells compared to that of sensitive cells as a function of the fitness cost of resistant cells for WiDr cl. B7 (c) DiFi cl. B6 (d).

(4)

Methods page 26 (no line numbering): “Persisters cells display a moderate division rate under drug treatment ($b \approx 0.3$ days⁻¹, Extended Data Fig. 8)”. It’s unclear to me how this b value was calculated, nor how it relates to Extended Data Fig. 8. This evaluation of parameter b for persister cells seems to contradict a claim in lines 294-295: “accurate quantification of slow cell division in persister cells was unfeasible”. Please clarify.

We apologize for the lack of clarity of this point. Related to this, please refer to the reply to point 1 raised by reviewer 1 above, and to Rebuttal Figures 1 and 2. We have modified the manuscript for clarification, as well as providing additional experimental evidence on persister cells’ proliferation and death rate.

Methods page 29 (no line numbering): Here I find the explanation of the normalization process difficult to follow. I suggest adding an extended data figure showing some typical data before and after normalization, both to illustrate the method and to enable readers to assess how much normalization changes the data.

We thank the referee for this comment. Following the suggestion, the effect of this normalization process is now illustrated in Rebuttal Figure 12 (Extended Data Figure 4 in the revised manuscript).

Rebuttal Figure 12. Illustration of the normalization process of the doses-response assay data used to obtain growth curves. (a) Originally measured values of the cell viability, quantified by the ATP signal for DiFi cl. B6 cells. The plots report ATP signal vs days of treatment of 5 independent replicates (marked with different colors) and for the indicated drug concentrations. (b) Values of the ATP signal shown in (a) are scaled by a constant value which is replicate-specific, i.e., is the same for each data point collected in the same replicate, as described in the Methods (see section “Calculation of growth curves from drug response assays”). (c) Values of the scaled ATP signal averaged across the 5 replicates.

(5)

Methods pages 30-31 (no line numbering): “We found that the best λ model the for WiDr is the one where the transition rate to persistence is linearly proportional to the drug concentration $\lambda([M]) = k[M]$, while for the DiFi the data is best described by a TP model variant with a constant transition rate $\lambda([M]) = \lambda_0$.” The finding that qualitatively different models fit data for the two different cell lines is interesting and deserves further scrutiny. Eyeballing Extended Data Figure 6b, it’s tempting to hypothesize that if experiments were conducted with drug concentrations substantially above 10^{-5} then cell viability might be seen to increase in DiFi at later time points, as in WiDr. If so then, across the wider range of drug concentrations, a linear model (or a model with some other monotonically increasing function) would fit DiFi data better than a model with constant λ . Looking at it another way: had the WiDr experiments been restricted to concentrations below 10^{-5} then a model with constant λ might have been mistakenly preferred. If the authors agree that this hypothesis is plausible then they should test it with new experiments, unless it would be infeasible to do so. In any case, I suggest adding to the manuscript some discussion of whether, notwithstanding the BIC and AIC values, it is plausible that λ increases monotonically (not necessarily linearly) with $[M]$ in DiFi. This discussion could be supported by a plot illustrating the best-fitting linear model (for the current DiFi data set or, if warranted and feasible, including new results at higher drug concentrations).

We agree with the referee that this point is interesting. Moreover, we used a wide range of drug concentrations in the doses-response assay with the highest doses of both dabrafenib and cetuximab being 10 or 50 times higher than the clinically relevant dose. We remark that experiments presented in the current version of the manuscript adequately reflect the clinically achievable plasma concentrations of each drug. From a pragmatic perspective, considering the high cost of cetuximab, performing experiments with even higher drug concentrations would also require a large amount of funding.

We find that it is plausible that the persister transition increases monotonically with [M] also in DiFi. In particular, we considered a functional dependence of the transition rate on the drug dependence of the form

$$\lambda([M]) = \lambda_0 (1 - e^{-\beta[M]})$$

This particular form is compatible with both a constant value, which is reached for high values of the concentrations

$$[M] \gg \beta^{-1} \rightarrow \lambda([M]) = \lambda_0$$

and with a linear increase for values

$$\beta * [M] \ll 1 \rightarrow \lambda([M]) \simeq \lambda_0 * \beta * [M] \equiv k * [M]$$

Single dose data of WiDr cl. B7 are therefore compatible with this functional form for values of $\beta^{-1} > 10^4$ Mol, although it is not possible to identify the exact value of this model parameter because it is probably bigger than the highest concentration we consider (we do not see any signal for the saturation of the transition rate).

Conversely, the single dose data of DiFi cl. B6 is compatible with this functional dependence for values of $[\beta^{-1} > 5 * 10^8$ Mol, and of $\lambda_0 = 0.234 \text{ days}^{-1}$ (Rebuttal Figure 13; Extended Data Figure 10 in the revised manuscript).

Hence, the two clones have a dependence on the drug concentration that can be explained by the same functional dependence, but with different values of the typical inverse concentration β .

We speculate that this difference is related to the fact that the two clones harbor a different genetic background (therefore representing two different types of CRC patients) which make them sensitive to different therapeutic regimen, which differently impact intracellular signaling, specifically, anti-EGFR cetuximab + anti-BRAF dabrafenib for BRAF mutated WiDr cells; anti-EGFR cetuximab alone for RAS/RAF wildtype DiFi cells.

Rebuttal Figure 13. DiFi cl. B6 single-dose data are compatible with a TP model with drug-dependent transition rate. The figure shows the TP model fit (right-hand plots) associated to four different scenarios (panels a, b, c and d) where the drug dependence on the drug concentration (black continuous lines in the left-hand side plots) differs. The first scenario (a) shows the preferred TP model variant: a constant transition rate. In the other three scenarios (B-D), the functional dependence of the transition rate on the drug concentration was assumed to be $\lambda([M]) = \lambda_0 (1 - e^{-\beta[M]})$, which is such that the transition rate λ reaches a plateau for drug concentration $M \gg \beta^{-1}$. Second Column: best model fit associated to the specified drug dependence on the drug concentration (lines) shown together with the single dose data. Values of the parameters: a: $\lambda_0=0.234 \text{ days}^{-1}$, b: $\lambda_0=0.234 \text{ days}^{-1}$, $\beta=5 \times 10^9 \text{ Mol}^{-1}$. c: $\lambda_0=0.234 \text{ days}^{-1}$, $\beta=5 \times 10^8 \text{ Mol}^{-1}$. d: $\lambda_0=0.234 \text{ days}^{-1}$, $\beta=5 \times 10^7 \text{ Mol}^{-1}$. The fits for the scenarios illustrated in panels B and C display a good agreement with the data, similar to a, suggesting that DiFi cl. B6 can also be described by a drug-dependent transition rate.

Related to the above: I recommend adding some discussion of how the drug concentrations used in these experiments compare to physiological concentrations during therapy. This is important for assessing how much treatment might affect mutation rates in the clinic.

We thank the referee for raising the point. The experiments presented in the current version of the manuscript adequately reflect the clinically achievable plasma concentrations of each drug. We clarified this point in the discussion as suggested by the reviewer.

Although the manuscript is very well written, it is peppered with minor spelling and grammar mistakes. I recommend a thorough proof-reading. For example (by line number where available):

- 229: alike -> like
- 242: models -> model
- 280: other -> others
- Bottom of page 13: "the the mutation rates" -> "the mutation rate"
- 391: "alongside to increase" -> "alongside an increase"
- Page 29: concentration -> concentration
- Page 29: Averages -> Average
- Page 29: cuvers -> curves
- Page 30: indipendent -> independent
- Page 30: keepeing -> keeping
- Page 30: "the for" -> for
- Page 35: analitcal -> analytical
- Numerous instances: "persisters cells" -> "persister cells"

We apologize for these problems - we corrected the typos in the revised version of the manuscript

Decision Letter, first revision:

IMPORTANT: Please note the reference number: NG-A57462R-Z Bardelli. This number must be quoted whenever you communicate with us regarding this paper.

16th Dec 2021

Dear Dr. Bardelli,

Thank you for your message of 16th Dec 2021, asking us to reconsider our decision on your manuscript "Drug-induced colorectal cancer persister cells show increased mutation rate". I have now discussed the points of your letter with my colleagues, and we think that you have some valid points. We therefore invite you to revise your manuscript along the lines that you propose.

When preparing a revision, please ensure that it fully complies with our editorial requirements for format and style; details can be found in the Guide to Authors on our website (<http://www.nature.com/ng/>).

Please be sure that your manuscript is accompanied by a separate letter detailing the changes you have made and your response to the points raised. At this stage we will need you to upload:

1) a copy of the manuscript in MS Word .docx format.

2) The Editorial Policy Checklist:

<https://www.nature.com/documents/nr-editorial-policy-checklist.pdf>

3) The Reporting Summary:

(Here you can read about the role of the Reporting Summary in reproducible science:

<https://www.nature.com/news/announcement-towards-greater-reproducibility-for-life-sciences-research-in-nature-1.22062>)

Please use the link below to be taken directly to the site and view and revise your manuscript:

[REDACTED]

With kind wishes,

Safia Danovi

Editor

Nature Genetics

Author Rebuttal, first revision:

We thank both referees for their work on our manuscript. We report below our point-by-point replies to their comments.

Referee expertise:

Referee #1: drug resistance

Referee #2: cancer evolution and mathematical modelling

Reviewers' Comments:

Reviewer #1:

Remarks to the Author:

In this manuscript, the authors use a combination of mathematical modeling and quantitative cell culture experiments to study the basis of the emergence of drug tolerant persister cells in colorectal cancer (CRC) patient-derived cell lines under therapy with an EGFR antibody and/or a BRAF inhibitor. The main claims are that the persister cells emerging later during therapy show an increased mutation rate that contributes to therapy resistance. The findings and the approaches used are interesting. The experiments are well conducted and described appropriately. There are issues that diminish enthusiasm however.

We thank the reviewer for showing interest in our work and for appreciating our experimental design.

(1) The authors seem to indicate that the persister cells are dying off slowly yet at the same proliferating slowly. This seems to lack clarity of logic. Which is it and via what mechanisms?

We thank the referee for raising this point. In the previous version of our manuscript, experimental data based upon CFSE measurements and the single-dose assay already supported the notion that persisters undergo both slow proliferation (shown by the dilution of the CFSE dye) and cell death (given the net decrease that we observe in cell number during prolonged drug treatment in single-dose drug assay).

To better elucidate persisters dynamics and cell death mechanisms during drug treatment we performed additional experiments.

We initially quantified the fraction of proliferating persisters by fluorescently labeling surviving cells with EdU (5-ethynyl-2'-deoxyuridine), a modified thymidine analogue that is efficiently incorporated into DNA during active DNA replication. Quantification of EdU positive cancer persister cells after 14, 17 and 20 days of treatment unveiled a fraction of replicating persisters ranging between 0,2 and 2,5 % (Rebuttal Figure 1, Extended Data Figure 6 in the revised manuscript).

Rebuttal Figure 1. A fraction of cancer persister cells slowly replicate under constant drug treatment. (a) DiFi and WiDr CRC clones were treated with anti-EGFR inhibitor cetuximab alone (DiFi) or in combination with dabrafenib (WiDr) for 2 weeks. After that, surviving residual cells were labelled with EdU at indicated time points for 4 hours, then fixed and analyzed by fluorescent microscopy. (b) Quantification of EdU positive cells at indicated time points. EdU-positive and total number of cells (based on nuclei staining with DAPI) were quantified using the “Analyze particles” function in ImageJ. A minimum of 200 cells for each timepoint were analyzed. Results represent average of two independent experiments, each represented by two technical replicates. (c) Representative images of EdU-positive persister cells for each CRC clone analyzed.

We then used the Nikon LIPSI live imaging system to set up a time-lapse microscopy assay. In detail, we treated both DiFi and WiDr clones for 2 weeks with anti-EGFR inhibitor cetuximab alone or in combination with BRAF inhibitor dabrafenib, respectively, until the emergence of persister cells. Labeling surviving persister cells with fluorescent dyes allows an accurate and sensitive detection of cell division (by staining of nuclei with NucBlue®) and cell death (with CellEvent™, a live Caspase3/7 activation fluorescent dye) within persisters subpopulations across several days of drug treatment.

Although the majority of CRC persister cells did not replicate, sparse cell division events were evident for persisters from all the CRC clones analyzed (Rebuttal Figure 2, Extended Data Figure 6 in the revised manuscript, and Extended data Movies in the revised manuscript). In some instances, cell division was successful (Rebuttal Figure 2a and b), while in other cases cells died concomitantly (Rebuttal Figure 2c). In parallel, we detected cell death, as highlighted by cells positive for the CellEvent signal either after a cell division event, or in non-dividing cells (Rebuttal Figure 2c and d, respectively). The imaging-based data was also used to estimate (roughly due to the small statistics) the effective growth rate of persister cells, which is compatible with the values obtained by the TP model inference (Rebuttal Figure 2e).

In summary, live monitoring of CRC persister cells confirms that cancer persister populations are slowly dying, while a fraction of cells is slowly replicating, therefore validating our previous findings.

Rebuttal Figure 2. Cell division and death of CRC persister cells can be directly visualized by microscopy. CRC cells were seeded in 24-multwell plates and treated for 2 weeks with anti-EGFR inhibitor cetuximab alone (DiFi) or in combination with BRAF inhibitor dabrafenib (WiDr). After 2 weeks of constant drug treatment, surviving persister cells were stained with Nucblue® (a dye for nuclei, in blue) and Caspase Cell EventTM (a live marker for the activation of Caspase3/7, in green) while maintaining drug pressure, and monitored for 5 days under an inverted widefield microscope (Nikon Lipsi, 20X Plan Apo objective with 0.75 NA) acquiring images every 45 minutes. For each clone we were able to follow 2 technical replicates, with 16 fields of view each, for a total of more than 1300 cells for each condition. Representative snapshots of cell division events and apoptotic events are reported, while supplementary movies report whole time-lapse experiments for representative fields of view for each clone. (a-b) cell division events; (c) cell death through apoptosis after a cell division; (d) cell death through apoptosis in non-dividing cells. (e) Numerical values of the effective growth rates of persister cells evaluated by means of imaging-based assays and by TP model inference are compatible. The scatter plot compares the values of the effective growth rate (b-d) obtained with image segmentation (x axis) and inferred with the TP model from dose-response and single dose assays (y axis). Squares correspond to center values and bars to standard deviations. The blue line marks the region of a perfect match between the two values.

(2) The authors show a potentially increased mutation rate in persister cells. However, there is no causative link shown between this process and the actual phenotype of persister cell insensitivity to therapy or the emergence of full drug resistance.

We thank the reviewer for the constructive criticism. We speculate that the adaptive mutability process observed in response to targeted therapy is not the origin of the persisters drug-tolerance phenotype, but a conserved mechanism of adaptation to treatment-induced stress (previously described in bacteria) that becomes relevant in surviving persister cells, which have lost multicellularity homeostatic controls as it happens for tumor cells (Russo et al., *Cancer Discovery* 2021).

Crucially, a causative link between mutation rate in persisters and emergence of drug resistance is established in this study: persisters must show accelerated mutation rates, or the timeframe at which persisters-derived resistant clones emerge in our experimental settings would not be justified.

This is supported by the following calculation based on our results. Persister-derived resistant clones are observed after >10 weeks of drug exposure. Since their mutation rate is 10-100x larger than (proliferating) untreated cells, in the absence of an increased mutation rate it would typically take (in a conservative estimate) $10 \times (10-100) \sim 100-1000$ weeks for a well to develop resistance based on the mutation rate of sensitive (untreated) cells that we experimentally calculated, or equivalently it would take $2 \times 10^5 - 2 \times 10^6$ wells to observe a few resistant clones after 10 weeks. We have clarified this point in the revised manuscript.

(3) What are the discrete mutations causing the emergence of late/full resistance?

Following the referee's question, we performed whole genome sequencing (WGS) and droplet digital PCR (ddPCR) on persister-derived resistant cells isolated from the MC-LD fluctuation experiments in order to characterize the molecular drivers of drug resistance.

As reported in the tables below the new experiments unveiled both single nucleotide variants (SNV) and copy number variation (CNV) as mechanisms of resistance in persisters-derived resistant clones. These drivers were among the known mechanisms of resistance to MAPK pathway inhibition in CRC previously described (Misale et al., *Nature* 2012; Misale et al, *STM* 2014; Arena et al., *Clin Cancer Res* 2015).

DiFi clones acquired either KRAS gene amplification (Rebuttal Figure 3, Extended Data Figure 16 in the revised manuscript), or different KRAS and NRAS mutations (Rebuttal Table 1, Extended Data Table 6 in the revised manuscript), while WiDr cells unveiled heterogeneous mutations in RAS genes (Rebuttal Table 1). We now discuss the meaning of these findings in the main text.

Rebuttal Figure 3. GCN analysis of DiFi persisters-derived resistant cells. Copy number variation (CNV) analysis of DiFi resistant cells derived from persisters within the MC-LD experimental setting. Upper panel shows the CNV differential analysis of two different parental (sensitive, untreated) clones. CNVs of three independent persister-derived resistant clones (namely clones 44, 46, 48) vs the parental counterpart were reported in the lower panels. Arrows indicate the KRAS amplification acquired at resistance to targeted therapy in all the 3 clones analyzed.

CRC cell line	Sample	Target	Fractional Abundance (%)
WiDr	Persister-derived resistant cells_1	KRAS_G12D	0.27
		KRAS_G12V	1.58
		KRAS_G13D	0.2
WiDr	Persister-derived resistant cells_2	KRAS_G12D	0.31
		KRAS_G12V	1.72
		KRAS_G13D	0.72
WiDr	Persister-derived resistant cells_3	KRAS_G12V	3.6
		KRAS_G13D	0.9
WiDr	Persister-derived resistant cells_4	KRAS_G12D	2.4
		KRAS_G12V	7
		KRAS_G13D	1.8
WiDr	Persister-derived resistant cells_5	KRAS_G12D	0.55
		KRAS_G12V	0.65
WiDr	Persister-derived resistant cells_6	KRAS G12C	1.23
		KRAS G12D	5.6
		KRAS G12V	7.6
		KRAS G13D	5.1
WiDr	Persister-derived resistant cells_7	KRAS G12C	0.47
		KRAS G12D	0.73
		KRAS G12V	4.4
WiDr	Persister-derived resistant cells_9	KRAS_G12D	0.28
		KRAS_G12V	1.13
		KRAS_G13D	0.29
WiDr	Persister-derived resistant cells_8	KRAS_G12V	0.73
WiDr	Persister-derived resistant cells_11	KRAS_G12V	3
DiFi	Persister-derived resistant cells_1	KRAS G12D	0.24
		KRAS Q61m	0.25
DiFi	Persister-derived resistant cells_2	KRAS G12D	4.9
		KRAS G13D	0.94
		KRAS Q61m	0.19
DiFi	Persister-derived resistant cells_3	KRAS G12D	1.3
		KRAS G12V	0.41
DiFi	Persister-derived resistant cells_5	KRAS G12D	0.44
		KRAS G12V	1.58

Rebuttal Table 1. Mutational analysis of persisters-derived resistant cells isolated from MC-LD experiment. Table lists the relative fractional abundance of mutated alleles detected by droplet digital PCR (ddPCR) analysis of indicated resistant cells derived from WiDr or DiFi persisters within the MC-LD experimental setting.

(4) Are there chromosomal copy number changes that are also heightened in the persister cells in addition to coding mutations?

Please refer to our answer to the previous point above.

(5) Data shown in include on only 2 cell line models and are all conducted in vitro. The authors should expand the number of patient-derived models and also include more clinically relevant models such as patient-derived organoids.

Following the suggestion, we replicated the full set of experiments and ran the analysis pipeline for two additional clones, one for each CRC cell line, called WiDr cl. B5 and DiFi cl. B3 (Rebuttal Figures 4 and 5, respectively; Extended Data Figures 14 and 15 in the revised manuscript, respectively), to exclude any clonal bias.

In both cases we found agreement with a drug-induced scenario for the persister transition (Rebuttal Figures 4a-b and 5a).

Considering WiDr cl. B5 we did not find evidence for a decline of persister cells over time ($D_p \approx 0$, Rebuttal Figure 4b), while we observed a decline of DiFi cl. B3 persister cells over time ($D_p > 0$, Rebuttal Figure 5a).

We found that WiDr cl. B5 is best described by a TP model with a transition rate proportional to the drug concentration (Rebuttal Figure 4c-d), consistently with what we observed for WiDr cl. B7. Similarly, for DiFi cl. B3 the best TP model variant is the one with a constant transition rate (Rebuttal Figure 5b-c), consistently with what we observed for DiFi cl. B6.

Notably, both clones display an increased mutation rate of persister cells (Rebuttal Figures 4e and 5d). Both transition rate to persisters and increased mutation rate did not vary when considering different values of the initial fraction of persisters f_0 within the range of values compatible with the dose-response and single dose assay (Rebuttal Figures 4 c,f and 5 b,e), thereby confirming our previous findings and excluding a clonal bias.

Rebuttal Figure 4. The analysis of an additional WiDr clone yields consistent results. The plots show the results of the TP model and MC-LD model inference for populations derived from an additional WiDr clone, namely WiDr cl. B5. (a) Doses-response dataset and best TP model fit as a function of the initial fraction of persister cells f_0 . (b) Single-dose dataset, best TP model fit and expected fraction of persister cells. (c) Joint posterior distribution of the TP model for the initial fraction of persisters and the transition rate. (d) Experimentally measured dose-responses growth curves (symbols) and best TP model fit (continuous lines). The doses-response datasets were normalized to the growth of the untreated cells using the best model parameters. (e) Quantification of mutation rates for sensitive (red) and persister (blue) cells in the MC-LD experiment. (f) Joint posterior distribution of the MC-LD model for initial fraction of persisters and the fold increase of the mutation rate of persister cells.

DiFi cl. B3 ◆

Rebuttal Figure 5. The analysis of an additional DiFi clone yields consistent results. The plots show the results of the TP model and MC-LD model inference for populations derived from an additional DiFi clones, namely DiFi cl B3. (a) Doses-response dataset and best TP model fit as a function of the initial fraction of persister cells. (b) Joint posterior distribution of the TP model for the initial fraction of persisters and the transition rate. (c) Experimentally measured dose-responses growth curves (symbols) and best TP model fit (continuous lines). The single-dose datasets were normalized to the growth of the untreated cells using the best model parameters. (d) Quantification of mutation rates for sensitive (red) and persister (blue) cells in the MC-LD experiment. (e) Joint posterior distribution of the MC-LD model for initial fraction of persisters and the fold increase of the mutation rate of persister cells.

We have considered the reviewer's suggestion to expand the analysis to patient-derived organoids. Unfortunately deploying the MC-LD fluctuation assay on patient-derived organoids (PDOs) is currently not conducive for the following reasons. Within the MC-LD setup, proliferation parameters and mutation rate are quantified precisely, and to this aim cancer cells must be expanded in twenty 96-well plates and kept under constant treatment for more than 12 weeks. Differently from 2D cell lines, the expansion of 3D growing PDOs in Matrigel does not allow precise quantification of population-dynamics parameters and reaching the sufficient number of cells to seed twenty 96-well plates could be prohibitive. Additionally, culturing of PDOs requires continuous (weekly) refresh of media and Matrigel (through centrifugation of PDOs and Matrigel, removal of Matrigel and seeding in fresh Matrigel) and this is not currently compatible with the MC-LD protocol.

We underline that the level of quantitative control we developed to perform and interpret the MC-LD experiment is unprecedented in mammalian cells, and that controlling all the parameters requires a suitable model system.

We understand that this point is at the same time an important limitation of our approach. In the revised manuscript, we have tuned all our claims, in order to make this limitation explicit, and at the same time we have clarified its broad potential for applicability.

While PDOs are (currently) not conducive for our purpose for the reasons above, it is potentially feasible to replicate the MC-LD fluctuation test in cell lines directly derived from CRC patients.

To this end, we attempted to replicate the TP pipeline using our molecularly annotated collection of 2D primary cell models (Russo et al., *Canc Discov* 2016; Siravegna et al., *Cancer Cell* 2018; Lazzari et al., *Clin Canc Res* 2019). We found that these CRC lines are still challenging to handle, but could in the future be exploited to set up MC-LD experiments.

These experiments still require several months to complete the initial set up. The initial results obtained with a RAS/BRAF wildtype CRC patient-derived primary cell line, sensitive to anti-EGFR cetuximab, are reported in Rebuttal Figure 6. We observed a bi-phasic killing curve at high drug concentration, hence supporting the presence of a transition to persister cells, similar to the one observed in WiDr and DiFi clones. The full results will be the subject of a follow up story.

Rebuttal Figure 6. CRC patient-derived primary cell line shows a bi-phasic killing curve in response to cetuximab treatment. We show here data collected in the dose-response assay (blue dots) of a RAS/BRAF wildtype CRC patient-derived primary cell line, namely CRC0078, for three different cetuximab concentrations (values indicated in red above each plot). The blue lines show the model expectation if no transition to persistence would occur. The lines do not capture the trend of the collected data for long times of treatment (>12 days). Hence, the collected data suggest the presence of a transition to persister cells.

(6) There are no data from clinical specimens to corroborate the main findings of increased mutation rate in therapy exposed tumors. This is a major gap that limits the potential clinical relevance.

We conceived this study to develop a comprehensively controlled assay to monitor multiple parameters and measure mutation rate changes during therapeutic treatment of tumor (and other) mammalian cells. The current manuscript reflects this and provides an experimental strategy that quantifies all the relevant parameters and provides a fully controlled framework to study persister population dynamics in human tumor cells. The approach we developed can be exploited in future studies, extend beyond cancer and used broadly to measure mutation rates in mammalian cells undergoing therapeutic stress.

We appreciate the suggestions to extend the analysis to clinical specimens. However, the tumor genetic heterogeneity in patients or in deriving in vitro models makes the increased mutations rate very difficult to detect, especially when resistance already occurred. Indeed, by using standard NGS-based approaches, we previously showed that tumor mutational burden is not significantly increased in cells that acquired resistance to targeted therapy after undergoing the persister phase (Russo et al., *Science* 2019).

Notably, the adaptive mutability phenotype is likely restricted in time (i.e., when the cells adapt to the new environment, through the acquisition of permanent mechanisms of resistance, the phenotype reverts) and confined to a subfraction of cells that would be difficult to capture with bulk analysis, especially once resistance has fully blown.

Based also on this, while tissue biopsy is usually collected from tumor lesions that relapsed upon therapy, ideally, one should be able to collect longitudinally minimal residual disease (MRD), resembling surviving persisters in vitro, at different time points to be able to measure the mutation rate in patients during therapeutic treatment, but this is clearly unfeasible.

Therefore, characterizing an increased mutation rate in clinical specimens collected from patients that progressed upon treatment with targeted therapies after an initial clinical response might be challenging, or even unfeasible.

Notably, Cipponi and colleagues recently reported WGS analysis on single clones-derived populations obtained from cancer cell lines during the early phase of the resistance development (i.e., when cells are not yet adapted to drug-induced stress), showed increased intra-clonal diversity (measured as the number of de novo SNVs) in resistant cells (Cipponi et al., Science 2020). We exploited the whole-genome sequencing performed by Cipponi and coworkers on clone-derived populations obtained from melanoma and liposarcoma cell lines untreated and in the early phase of the resistance development, to quantify mutation rates using our modified Luria-Delbruck (MC-LD) model. The analysis is reported below in the reply to point 2 of reviewer #2 below.

What is important to stress here, is that the estimated mutation rates of treated cells in their experiments are only mildly higher for treated cells, because of the less controlled setup as well as the fact that they do not keep into account the role of persisters (which give rise to many late-emerging resistance mutations from a very small residual population, a fact that can be explained only with a strongly increased mutation rate). This is a further proof of how the detection of an increased mutation rate may be difficult even in highly controlled systems.

(7) Are there vulnerabilities that could be therapeutically exploited as a result of the increased mutation rate in the persister cells? If so, which are they and can the authors test at least one as proof-of-principle?

We thank the referee for raising the point. We agree this is potentially an exciting implication of our results and we are indeed pursuing different approaches aiming at targeting the players of adaptive mutability in cancer cells to possibly delay, or even prevent, the occurrence of secondary resistance. Our previous findings unveiled a switch from high to low fidelity DNA replication process through downregulation of MMR and HR DNA repair genes and upregulation of specialized error-prone DNA polymerases in presence of increased DNA damage in response to treatment with targeted therapies (Russo et al., Science 2019).

Accordingly, inhibition of DNA polymerases might be exploited as novel combinatorial treatment strategy in tumor cells.

Among the DNA polymerases that we found upregulated in cancer cells upon targeted therapy-induced stress response, REV1 carries out the mutagenic translesion synthesis (TLS), a process that allows cells to tolerate DNA damage by bypassing lesions that block normal DNA replication, resulting in the introduction of mutations. Indeed, the inhibition of TLS would likely increase the cytotoxic effects of DNA damaging agents and potentiate their effectiveness, especially in cancer cells that are defective in DNA repair. Importantly, inhibition of TLS activity might prevent drug resistance caused by the mutagenic replication of DNA. Interfering with TLS with a REV1 small molecule inhibitor has been shown to enhance chemotherapy efficacy by suppressing tumor growth both in vitro and in vivo (Wojtaszek et al., Cell. 2019; Chatterjee et al., PNAS 2020).

Based on this, we performed initial tests of the REV1 inhibitor (REV1i) in combination with MAPK pathway in a 5-days drug response assay. We found that REV1i does not enhance the acute effect of MAPK pathway inhibition in reducing cell viability (Rebuttal Figure 7a).

Next, we performed a Time-To-Progression (TTP) assay we previously established (Russo et al., Nat Comm 2018) to monitor the development of secondary resistance in cancer cells. In detail, we treated MSS DiFi and WiDr CRC cells with either MAPK pathway inhibitor, or REV1 inhibitor or their combination. As shown in Rebuttal Figure 7 below, REV1 inhibitor significantly delayed (in case of WiDr) or even prevented (in case of DiFi) the development of secondary resistance to EGFR/BRAF inhibitors (Rebuttal Figure 7b), therefore confirming our hypothesis.

Although we find these results exciting and promising, further characterization of the mechanism(s) of action of this and other inhibitors are needed, and this is part of our future experiments. We believe this kind of studies are beyond the purpose of the present work, whose scope is to introduce and deploy the fluctuation assay setup to quantify the persister transition and measure for the first time mutation rate under therapy in cancer cells. Accordingly, we have not included the combinatorial results in this manuscript, but wanted to present them as preview to the reviewers since these new data suggest that the MC-LD assay can lead to new discoveries with translational relevance.

Rebuttal Figure 7. Inhibition of error-prone DNA polymerase REV1 delays the onset of secondary resistance to MAPK pathway inhibition in CRC cells. (a) WiDr and DiFi CRC cells were treated with indicated drugs for 5 days. After that, cell viability was measured by the ATP assay. (b) WiDr and DiFi CRC cells were treated in parallel with MAPK pathway inhibitors (anti-EGFR inhibitor cetuximab alone in the case of DiFi, or in combination with anti-BRAF inhibitor dabrafenib in the case of WiDr), REV1 inhibitor or their combination and number of cells was monitored during the treatment, until the emergence of resistance.

In parallel, in the revised manuscript we are now providing proof-of-principle computational evidence that the drug-dependent transition to persisters can be therapeutically exploited.

The best TP model variant that describes the observed dynamics for WiDr cells is the one with a transition rate to persistence that is linearly dependent on drug concentration $\lambda([M]) = k [M]$ (see below). This particular functional dependence corresponds to a scenario where higher drug concentrations induce both an increased death rate of sensitive cells, which is desirable, but, at the same time, an increased transition to the persister state, which act as a reservoir for the emergence of resistance. We show here that, by modulating the drug concentration over time, this trade off can be exploited to reduce the number of persisters cells emerging in response to the therapeutic drug.

To illustrate this point, we considered two alternative drug delivery strategies: (i) the concentration of the drug is maintained constant over time; (ii) the concentration of the drug is linearly increased over time. In order to define a fair comparison between the two strategies, we constrained this second strategy to deliver the same average drug concentration as the first one, meaning that the total amount of drug delivered is the same for the two approaches. The choice of the two strategies is shown in Rebuttal Figure 8 a,b below (Extended Data Figure 17 in the revised manuscript).

We solved the TP model (eq.1) numerically, using these two alternative choices for the dependence of the drug concentration over time as model inputs, and compared the two outcomes (Rebuttal Figure 8 a-b). The other model parameters were set to the experimentally derived values for WiDr cl. B7 (see Extended Data Table 4 in the revised manuscript). In order to investigate the worst-case scenario, i.e., the one with the maximal number of emergent persister cells, we neglected their death ($D_p=0$). We found that a drug delivery strategy with a linearly increasing drug concentration can reduce the total number of emergent persister cells (Rebuttal Figure 8d). Interestingly, with this strategy the drug concentration is also kept under the minimum inhibitory concentration (MIC) value at the beginning of the treatment, suggesting that one could design therapies that are less aggressive and also more effective.

Finally, we note that this setup would produce resistance much slower in a MC-LD setup, as it greatly reduces the number of persisters (up to 40% less) (Rebuttal Figure 8d).

Rebuttal Figure 8. The dependence on the drug concentration of the transition rate to persistence can be exploited to reduce the rate of emergence of persisters. (a) The plot illustrates a standard drug delivery strategy, which consists of a constant drug concentration delivered over time. The values for the concentration of dabrafenib and cetuximab are realistic for colorectal cancer cells similar to WiDr cl. B7. The drug concentration of cetuximab is also assumed constant over time. The value of the Minimum Inhibitory Concentration (MIC, dashed black line) is the minimal value of the drug concentration for which a population of sensitive cancer cells would not increase over time. For the case of WiDr cl B7, the value of the MIC can be obtained from our estimated model parameters, $\text{MIC} = a-1 = 3.5 \times 10^{-7} \mu\text{M}$ (see Extended Data Table 4 in the revised manuscript). (b) In a modified drug delivery strategy, the drug concentration of dabrafenib increases linearly over time, and the drug concentration of cetuximab is also assumed constant over time. This strategy is constrained to have the same average drug concentration as in panel (a), meaning that the total amount of drug delivered in 10 days is the same in the two strategies. In this case, at the beginning of the in-silico treatment, the delivered drug is below the MIC value. (c) Expected number of persister cells predicted by the TP model (eq. 1) for WiDr cl. B7 cells, corresponding to the two alternative drug delivery strategies: constant value (gray line) and linear increase (orange line). (d) Relative fold change of the number of persister cells emerged with the drug strategy B vs drug strategy A, evaluated as a function of the average drug concentration.

(8) Is there a causal link between inhibition of EGFR or RAF signaling and the increased mutation rate in persister cells? This should be experimentally tested here. Does the signaling pathway remain silent throughout persister development and is this required for the increased mutation rate?

We thank the referee for this comment. In our previous study (Russo et al, Science 2019) we showed that the adaptive mutability phenotype is not a drug-dependent side-effect, but rather it is connected to the perturbation of oncogene addiction in cancer cells.

Indeed, downregulation of EGFR and KRAS in EGFR-amplified DiFi cells, or of EGFR and BRAF in BRAF-mutated WiDr cells through siRNA recapitulates the same phenotype of DNA repair impairment observed upon treatment with targeted therapies (Rebuttal Figure 9, left part; reproduced from Russo et al., Science 2019, Fig. 3e). Differently, other cellular stresses such as cell cycle arrest or direct DNA damaging agents did not recapitulate the phenotype, therefore suggesting that the interruption of oncogene addiction initiate adaptive mutability-stress response, leading in turn to genetic instability (Russo et al., Science 2019).

Additionally, after more than two weeks of constant drug pressure, DNA repair downmodulation is maintained in persisters together with inhibition of the MAPK pathway, thus supporting the concept that the signaling pathway

remains silent throughout the persister phase (Rebuttal Figure 9, right part; reproduced from Russo et al., Science 2019, Fig. S4B).

Rebuttal Figure 9 (modified from Russo et al., Science 2019, Fig.3E and S4B). Perturbation of oncogene addiction initiates adaptive mutability stress response in CRC cells. (e) Wt DiFi (left panel) and BRAF mutated WiDr (right panel) cells were transfected with indicated siRNA or combinations of them. After 72 hours protein lysates were analyzed by western blot. All star, non-targeting siRNA. (b) Expression levels of MMR and HR DNA repair proteins were evaluated in 2 independent persister populations (P1 and P2) for each cell line model. NT stands for untreated cells.

(9) The authors should test another EGFR and BRAF inhibitor to exclude drug-specific effects.

In this study, we used two CRC cell lines that are dependent upon distinct oncogenic events (EGFR amplification and BRAF mutation) and for which we have already shown that adaptive mutability is not dependent on the use of a specific drug, as it can be replicated by siRNA-mediated downregulation of the oncogenic pathways the cell line is addicted to, based on the genetic milieu (please see Rebuttal Figure 9 above), therefore excluding a drug-specific effect.

(10) Is it not clear that the advance presented in this manuscript is significantly further along or novel beyond the authors previous related study so as to warrant publication in Nature Genetics.

We thank the referee for this comment, and we apologize if these aspects did not emerge clearly from the manuscript. The scope of this work is to provide for the first time, to the best of our knowledge, a fluctuation assay for a mammalian cell system under treatment with targeted therapies, where quantitative precision can be achieved, and where mathematical models for the persisters transition and the mutation rate can be validated. The two main findings that were not demonstrated in the Russo et al. 2019 study are: (i) the transition to persister is mainly drug-induced and (ii) persisters quantitatively increase their mutation rate.

Here, for the first time we quantitatively measured the mutation rate of cancer cells during drug treatment, formally demonstrating that the previously unveiled stress response induced by targeted therapy (Russo et al., Science 2019) actually translates into 7-50 fold increase of mutation rate in surviving persister cells. Importantly, this was possible only by coupling experimental characterization of cancer cells, evaluation of multiple parameters of cancer cell population dynamics during drug exposure and mathematical modelling.

Notably, the mathematical framework coupled to the experimental assays presented here could be applied to unveil and quantify the impact of a wide range of environmental conditions on the persister phenotype and mutation rates in mammalian cells.

Moreover, to the best of our knowledge a quantitative definition of cancer persisters was lacking together with several aspects of their population dynamics, including whether they are pre-existing in a cancer population prior to treatment and whether they display proliferation and cell death. As highlighted by referee #2, for the first time

in human cancer cells we deployed rigorous and controlled experimental designs and mathematical modeling to track persisters dynamics over time.

We have modified the manuscript to present our achievements in light of our previously published results, and to highlight their relevance and the advances they provide.

Reviewer #2:

(1)

Remarks to the Author: This manuscript builds on the authors' previous work showing that, in human colorectal cancer cell lines, EGFR/BRAF inhibition substantially increases the mutation rate of drug-tolerant persister cells. The main novelty of the new study is the application of mathematical modelling to infer the rates at which persisters arise and acquire drug resistance. I have focussed my review on the mathematical methods as this is my area of expertise.

The mathematical methods – and to the best of my knowledge the experimental methods – are well chosen and have been carefully applied. The results are convincing and clearly presented. The conclusions are important inasmuch as they challenge prevailing models of how cancer cells acquire resistance to targeted therapies. The big caveat is that strong evidence for this phenomenon has so far been found only in cell lines and not in patients.

We thank the reviewer for highlighting the novelty of our approach and the implications of our work.

As explained in response to Reviewer #1 above, detection of increased mutation rate in clinical specimens collected from patients that progressed upon treatment with targeted therapies after an initial clinical response might be challenging. While tissue biopsy is usually collected from tumor lesions that relapsed upon therapy, we should instead have access to minimal residual disease (MRD) (most probably due to surviving persister cells) at different time points during therapy. As the reviewer certainly appreciate this unfortunately remains technically very challenging. We are considering developing a clinical protocol, to intercept these types samples but so far, we have not been successful, also in light of the clinical and ethical feasibility issues.

Notably, we underline again that the level of quantitative control of the MC-LD experiment is unprecedented in mammalian cells, and achieving this control requires a simple system. We have clarified these points in the revised manuscript.

(2)

Regarding this caveat, a new study in cetuximab-treated colorectal cancers casts doubt on the clinical relevance of the current manuscript's results by finding that, "despite the functional evidence for cetuximab-induced mutagenesis in CRC cell lines, our analysis in patients shows that its contribution to cetuximab resistance evolution is probably small." [Woolston et al. Nat. Ecol. Evol., <https://www.nature.com/articles/s41559-021-01470-8>]. These new clinical findings should be discussed. In particular, the current manuscript's conclusion that "The combined experimental and modeling framework presented here may have broad and far-reaching clinical implications" and the claim that the manuscript "pinpoints new strategies to restrict tumor recurrence" should be reassessed in light of the work by Woolston et al.

We thank the reviewer for raising this important point. We have reformulated and clarified our claims. A crucial point is that the tumor genetic heterogeneity in patients or in deriving in vitro models (such as PDOs, for which the fluctuation assay setup cannot be applied) makes the increased mutations rate very difficult to detect, especially when resistance already occurred.

Tackling this issue is part of our long-term goals and we have launched an extensive study to monitor sequencing mutation rates in patient-derived xenografts (PDXs) by sequencing; however, this approach, which is briefly highlighted below, will require many months to complete.

We have read with great interest the work of Woolston and colleagues, and have discussed the findings in the revised manuscript. Indeed, the sequencing analyses performed by Woolston and colleagues are in line with previous reports showing that tumor mutational burden is not significantly increased in cells that acquired resistance to targeted therapy after undergoing persister phase, including our own analysis in Russo et al., Science 2019.

The crucial point is that without a controlled setup these changes are very difficult to detect. To explain this problem, we can use a recent analysis by Cipponi and coworkers showing an increased intra-clonal diversity (number of sub-clonal SNVs) in melanoma and liposarcoma cell lines that developed resistance in vitro, as

compared to their untreated counterpart consistent with increase of mutation rate as the authors suggested (Cipponi et al., Science 2020).

We have taken advantage of the whole-genome sequencing data from this work, performed on single cell clones-derived populations obtained from melanoma and liposarcoma cell lines untreated or in the early phase of the resistance development, to quantify mutation rates applying our modified Luria-Delbruck (MC-LD) model.

The generalized estimator we presented in our work in Eq.s (2-4) can be modified to describe the emergence of sub-clonal mutations, by associating:

- (i) the number of wells developing resistance  number of base sites with a sub-clonal mutation
- (ii) total number of wells  total genome length ($L = 3.2 * [10]^9$ sites)

The mathematical description as well as further details about the extension of our modelling to the analysis of sequencing data are active subjects of research in our group. While the derivation and the validation of this model are beyond the scope of this study, here, we report the mathematical definition of the estimator which we have applied to the data we retrieved from Cipponi et al., Science 2020.

The estimator of the mutation rate to be applied to sequencing data takes the form

$$\mu = -\text{Log}(1 - f) / \mathcal{M}$$

Where $f = N_{\text{sub}}/L$ is the fraction of genome base sites with an observed sub-clonal mutation, and \mathcal{M} are the total number of cell divisions in which the N_{sub} mutations have emerged. Of note, this mutation rate is per generation, and not chronological.

We have retrieved the numerical values of N_{sub} corresponding to the clones investigated in Cipponi et al., Science 2020 (Table S3, entry "Bozic Adjusted"). The numerical value of \mathcal{M} is set by the depth of the sequencing used. For the genomic data considered we have $\mathcal{M} \approx 15$, which is based on the sequencing features to detect mutation with an intra-population frequency $>1/16$. Hence, all the observed sub-clonal mutations must have emerged in the first 4 generations.

We find that (Rebuttal Figure 10),

- (i) estimated mutation rates are in general agreement with other estimates (point mutation rate around 10^{-9} per generation);
- (ii) resistant clones show a systematic increase of the mean value of the mutation rate, although this enhancement is not as strong as in our data;
- (iii) silencing of MTOR induced an enhancement of the mutation rate with respect to control as expected (Cipponi et al., Science 2020,).

Rebuttal Figure 10. Values of the mutation rate estimated from genetic data reported in Cipponi et al., Science 2020. We show here box-whisker plots (mean: red line, box: 1st and 3rd quartile) representing the distribution of mutation rates inferred from Cipponi et al., Science 2020 and for indicated clones (SKMEL28: human melanoma cell line; 94T778: human liposarcoma cell line). NS, shRNAs: 94T778 cells engineered with a nonsilencing, control short hairpin RNA; MTOR, ShRNAs; 94T778 cells engineered with mTOR-silencing short hairpin RNA).

It is important to stress that the estimated mutation rates of treated cells from the Cipponi et al. data are only mildly higher as compared to their untreated counterpart. The reason of this are the limitations of the experimental setup employed in this study, compared to ours. Specifically, in Cipponi et al., resistance developed from a cancer

cell population of decreasing and not-well-defined size. Moreover, in their setup they do not separate pre-existing and newly-developed resistant cells as we do: the outgrowth of pre-existing resistant cells represents a strong confounding effect on the slower development of resistance from persister cells, and therefore could mask an increase in mutation rate in residual persister cells. Moreover, the authors do not keep into account the role of persisters in their experimental setup.

In brief, in our experimental setup persisters give rise to unexpectedly many late-emerging resistance mutations from a very small residual population, a fact that can be explained only with a strongly increased mutation rate, but the detection of an increased mutation rate would be very difficult in any “normal” situation where these mutations are masked by pre-existing resistant clones.

Besides the above, I have only minor suggestions for improvement.

(3)

Results page 13 (no line numbering): “we assumed that resistant cells divide with rate b and die with rate d , just like untreated cells.” Unless I’ve misunderstood, this assumption is at odds with the conventional assumption that resistance typically imposes a cellular fitness cost (i.e. lower b and/or higher d) relative to sensitive cells growing in the absence of treatment. Can the authors provide evidence to support their assumption? If not, I suggest they examine whether their results are robust to adding a fitness cost (e.g. 10% lower b for resistant cells, relative to sensitive cells).

We thank the referee for her/his suggestion. The estimators developed for the fluctuation assay only depend on $b-d$ in the case of persisters, so that our findings are robust in terms of the detailed values of these parameters.

Following the reviewer’s suggestion, we have investigated the stability of the inferred values of the mutation rates against variation of the division rate of resistant cells. To do so, we considered mutation rate estimators computed assuming a value of the division rate of resistant cells of the form $b_{res} = [(1 - f)]_{cost} * b$, where b is the division rate of the sensitive cells, and f_{cost} is the fitness cost of resistance, which we have varied in the region $[0, 10\%]$. The result of this analysis is shown in Rebuttal Figure 11 below (Extended Data Figure 13 in the revised manuscript).

We find that both the mutation rate of the sensitive cells and the fold increase of the mutation rate of persisters cells display a very mild variation on the fitness cost.

Rebuttal Figure 11. The inferred values of the mutation rate are stable against the variation of the value of the fitness cost of resistant cells. (a-b) Inferred values of the mutation rate of sensitive cells as a function of the fitness cost of resistant cells for WiDr cl. B7 (a) DiFi cl. B6 (b). The variation of the estimated mutation rates is very small. (C-D) Inferred values of the fold increase of the mutation rate of persister cells compared to that of sensitive cells as a function of the fitness cost of resistant cells for WiDr cl. B7 (c) DiFi cl. B6 (d).

(4)

Methods page 26 (no line numbering): “Persisters cells display a moderate division rate under drug treatment ($b \approx 0.3$ days⁻¹, Extended Data Fig. 8)”. It’s unclear to me how this b value was calculated, nor how it relates to Extended Data Fig. 8. This evaluation of parameter b for persister cells seems to contradict a claim in lines 294-295: “accurate quantification of slow cell division in persister cells was unfeasible”. Please clarify.

We apologize for the lack of clarity of this point. Related to this, please refer to the reply to point 1 raised by reviewer 1 above, and to Rebuttal Figures 1 and 2. We have modified the manuscript for clarification, as well as providing additional experimental evidence on persister cells’ proliferation and death rate.

Methods page 29 (no line numbering): Here I find the explanation of the normalization process difficult to follow. I suggest adding an extended data figure showing some typical data before and after normalization, both to illustrate the method and to enable readers to assess how much normalization changes the data.

We thank the referee for this comment. Following the suggestion, the effect of this normalization process is now illustrated in Rebuttal Figure 12 (Extended Data Figure 4 in the revised manuscript).

Rebuttal Figure 12. Illustration of the normalization process of the doses-response assay data used to obtain growth curves. (a) Originally measured values of the cell viability, quantified by the ATP signal for DiFi cl. B6 cells. The plots report ATP signal vs days of treatment of 5 independent replicates (marked with different colors) and for the indicated drug concentrations. (b) Values of the ATP signal shown in (a) are scaled by a constant value which is replicate-specific, i.e., is the same for each data point collected in the same replicate, as described in the Methods (see section “Calculation of growth curves from drug response assays”). (c) Values of the scaled ATP signal averaged across the 5 replicates.

(5)

Methods pages 30-31 (no line numbering): “We found that the best λ model the for WiDr is the one where the transition rate to persistence is linearly proportional to the drug concentration $\lambda([M]) = k[M]$, while for the DiFi the data is best described by a TP model variant with a constant transition rate $\lambda([M]) = \lambda_0$.” The finding that qualitatively different models fit data for the two different cell lines is interesting and deserves further scrutiny. Eyeballing Extended Data Figure 6b, it’s tempting to hypothesize that if experiments were conducted with drug concentrations substantially above 10^{-5} then cell viability might be seen to increase in DiFi at later time points, as in WiDr. If so then, across the wider range of drug concentrations, a linear model (or a model with some other monotonically increasing function) would fit DiFi data better than a model with constant λ . Looking at it another way: had the WiDr experiments been restricted to concentrations below 10^{-5} then a model with constant λ might have been mistakenly preferred. If the authors agree that this hypothesis is plausible then they should test it with new experiments, unless it would be infeasible to do so. In any case, I suggest adding to the manuscript some discussion of whether, notwithstanding the BIC and AIC values, it is plausible that λ increases monotonically (not necessarily linearly) with $[M]$ in DiFi. This discussion could be supported by a plot illustrating the best-fitting linear model (for the current DiFi data set or, if warranted and feasible, including new results at higher drug concentrations).

We agree with the referee that this point is interesting. Moreover, we used a wide range of drug concentrations in the doses-response assay with the highest doses of both dabrafenib and cetuximab being 10 or 50 times higher than the clinically relevant dose. We remark that experiments presented in the current version of the manuscript adequately reflect the clinically achievable plasma concentrations of each drug. From a pragmatic perspective, considering the high cost of cetuximab, performing experiments with even higher drug concentrations would also require a large amount of funding.

We find that it is plausible that the persister transition increases monotonically with [M] also in DiFi. In particular, we considered a functional dependence of the transition rate on the drug dependence of the form

$$\lambda([M]) = \lambda_0 (1 - e^{-\beta[M]})$$

This particular form is compatible with both a constant value, which is reached for high values of the concentrations

$$[M] \gg \beta^{-1} \rightarrow \lambda([M]) = \lambda_0$$

and with a linear increase for values

$$\beta * [M] \ll 1 \rightarrow \lambda([M]) \simeq \lambda_0 * \beta * [M] \equiv k * [M]$$

Single dose data of WiDr cl. B7 are therefore compatible with this functional form for values of $\beta^{-1} > 10^4$ Mol, although it is not possible to identify the exact value of this model parameter because it is probably bigger than the highest concentration we consider (we do not see any signal for the saturation of the transition rate).

Conversely, the single dose data of DiFi cl. B6 is compatible with this functional dependence for values of $[\beta^{-1} > 5 * 10^8$ Mol, and of $\lambda_0 = 0.234 \text{ days}^{-1}$ (Rebuttal Figure 13; Extended Data Figure 10 in the revised manuscript).

Hence, the two clones have a dependence on the drug concentration that can be explained by the same functional dependence, but with different values of the typical inverse concentration β .

We speculate that this difference is related to the fact that the two clones harbor a different genetic background (therefore representing two different types of CRC patients) which make them sensitive to different therapeutic regimen, which differently impact intracellular signaling, specifically, anti-EGFR cetuximab + anti-BRAF dabrafenib for BRAF mutated WiDr cells; anti-EGFR cetuximab alone for RAS/RAF wildtype DiFi cells.

Rebuttal Figure 13. DiFi cl. B6 single-dose data are compatible with a TP model with drug-dependent transition rate. The figure shows the TP model fit (right-hand plots) associated to four different scenarios (panels a, b, c and d) where the drug dependence on the drug concentration (black continuous lines in the left-hand side plots) differs. The first scenario (a) shows the preferred TP model variant: a constant transition rate. In the other three scenarios (B-D), the functional dependence of the transition rate on the drug concentration was assumed to be $\lambda([M]) = \lambda_0 (1 - e^{-\beta[M]})$, which is such that the transition rate λ reaches a plateau for drug concentration $M \gg \beta^{-1}$. Second Column: best model fit associated to the specified drug dependence on the drug concentration (lines) shown together with the single dose data. Values of the parameters: a: $\lambda_0=0.234 \text{ days}^{-1}$, b: $\lambda_0=0.234 \text{ days}^{-1}$, $\beta=5 \times 10^9 \text{ Mol}^{-1}$. c: $\lambda_0=0.234 \text{ days}^{-1}$, $\beta=5 \times 10^8 \text{ Mol}^{-1}$. d: $\lambda_0=0.234 \text{ days}^{-1}$, $\beta=5 \times 10^7 \text{ Mol}^{-1}$. The fits for the scenarios illustrated in panels B and C display a good agreement with the data, similar to a, suggesting that DiFi cl. B6 can also be described by a drug-dependent transition rate.

Related to the above: I recommend adding some discussion of how the drug concentrations used in these experiments compare to physiological concentrations during therapy. This is important for assessing how much treatment might affect mutation rates in the clinic.

We thank the referee for raising the point. The experiments presented in the current version of the manuscript adequately reflect the clinically achievable plasma concentrations of each drug. We clarified this point in the discussion as suggested by the reviewer.

Although the manuscript is very well written, it is peppered with minor spelling and grammar mistakes. I recommend a thorough proof-reading. For example (by line number where available):

- 229: alike -> like
- 242: models -> model
- 280: other -> others
- Bottom of page 13: “the the mutation rates” -> “the mutation rate”
- 391: “alongside to increase” -> “alongside an increase”
- Page 29: concentration -> concentration
- Page 29: Averages -> Average
- Page 29: cuvers -> curves
- Page 30: indipendent -> independent
- Page 30: keepeing -> keeping
- Page 30: “the for” -> for
- Page 35: analitcal -> analytical
- Numerous instances: “persisters cells” -> “persister cells”

We apologize for these problems - we corrected the typos in the revised version of the manuscript

Decision Letter, second revision:

27th Jan 2022

Dear Professor Bardelli,

First of all, please accept my apologies for the delay in returning this decision to you. I am grateful for your patience.

Your Article, "A modified fluctuation-test framework characterizes population dynamics and mutation rate of cancer persister cells" has now been seen by 3 referees. You will see from their comments below that while they find your work of interest, some important points are raised. We are interested in the possibility of publishing your study in Nature Genetics, but would like to consider your response to these concerns in the form of a revised manuscript before we make a final decision on publication.

You'll see that Reviewer #1 notes the revisions and has indicated which to include in the next draft of the work. They remain concerned about the lack of genuine clinical validation and while we will not ask you for further experiments to allay this concern, you might like to consider bolstering your discussion of this limitation in the text.

As you know, I recruited Reviewer #3 to look over the mathematical analyses in the revision as Reviewer #2 was unable to do so in detail owing to time constraints. Reviewer #3 considers the work to be technically sound, but they do have some suggestions regarding to the interpretation of the work, and developing a more nuanced narrative regarding alternative scenarios that might equally well explain your findings. Please address these comments in full.

When the revision comes in, we will aim to assess it in-house but depending on your response, we might return to one or more reviews if we feel that we are unable to do so. Please be assured that we will only do this if absolutely necessary.

We therefore invite you to revise your manuscript taking into account all reviewer and editor comments. Please highlight all changes in the manuscript text file. At this stage we will need you to upload a copy of the manuscript in MS Word .docx or similar editable format.

*2) If you have not done so already please begin to revise your manuscript so that it conforms to our Article format instructions, available

[here](http://www.nature.com/ng/authors/article_types/index.html).

*3) Include a revised version of any required Reporting Summary:

[REDACTED]

We hope to receive your revised manuscript within four to eight weeks. If you cannot send it within this time, please let us know.

Sincerely,

Safia Danovi
Editor
Nature Genetics

Referee expertise:

Referee #1: drug resistance

Referee #2: mathematical modelling, cancer evolution

Referee #3: mathematical modelling, cancer evolution (recruited to check mathematical analyses that Reviewer #2 was unable to)

Reviewers' Comments:

Reviewer #1:

Remarks to the Author:

The authors have submitted a revised manuscript that is improved. This reviewer appreciates the thoughtful responses and additional experiments. However, there are still important limitations in that the number of cell lines is essentially 2 (albeit using different clones), as far as this reviewer can tell and in vivo models are scarce in this study so there is little mitigation of potential cell line-specific biases and clinical heterogeneity. Additionally, there is no attempt to show some evidence of relevance using human clinical samples, even if in a small number of cases (although the authors points are understood). Finally, the therapeutic experiments using the REV inhibitor should be confirmed/validated and included in this study, as this is an interesting observation that could have translational relevance.

Reviewer #2:

Remarks to the Author:

In my review of the original version of this manuscript, I was doubtful of the clinical relevance of the findings, especially in light of a recent paper by Woolston et al. The authors have addressed this concern in their revision by adding a new paragraph to the Discussion, rephrasing some of their claims, and conducting new analyses. They have also made appropriate changes in response to my other questions and minor suggestions. Although the clinical relevance of the work remains somewhat uncertain, I'm satisfied that the authors have addressed my concerns and I have no further suggestions for improvement.

Reviewer #3:

Remarks to the Author:

This is a very interesting manuscript that investigates the response of colon cancer cell lines to targeted therapy. It combines multiple experimental approaches and mathematical and computational modelling.

I will restrict my comments on the theoretical and computational modelling and data interpretation but will not comment on details of experimental design or data generation.

I have no major concerns on the formal analysis of the mathematical and computational model. The

equations and analytical approximations seem sound to me.

I have a few questions on the interpretation of the data:

The authors explain the two slopes of the treatment response by a treatment induced non (or slow) proliferating cell population with higher treatment resistance, making analogies to similar dynamic patterns in bacteria. I agree with the author's that this is one possible interpretation. However, although I am not aware of such treatment responses in colon cancer patients, there are classic examples (most prominently the treatment response of chronic myeloid leukemia to Imatinib) that do show the same very typical two slope response to targeted therapy (see for example Michor and Nowak, Nature 2005). Classically, these dynamics are explained by a proliferation hierarchy, where slower stem like cells give rise to faster proliferating progenitor or differentiated cells. However, the transition rates in these models are reversed. Slower proliferating cells at low frequency give rise to faster proliferating cells (see for example Werner, Scott et al. Cancer Research 2016 for examples of the dynamic patterns, but there are many other papers). In these models, treatment does not actively induce quiescence by a phenotypic switch, but simply selects on the pre-existing hierarchy.

In the same line of arguments, a Poisson like distribution of quiescent (slow proliferating) phenotypes in a luria-delbrück like experiment itself does not necessarily show a phenotypic switch. If quiescent cells are pre-existing at a low frequency, a luria-delbrück experiment would imply a sampling process and we would predict a simple Binomial distribution of quiescent phenotypes after sampling. The expected parameter range of this distribution would be a small p and large n , which implies that the binomial distribution would be very close to a Poisson distribution.

I agree with the authors that phenotypic switching is one possible explanation, but I do not believe the authors actually show that phenotypic switching occurs. It's a difficult task to unambiguously show phenotypic switching as driving force of resistance. However, I would suggest that the authors formulate the results on that aspect a little more balanced and state that there are uncertainties and possible other explanations (see points above).

The authors need to provide a bit more detail and explanation on the stochastic modelling approach. What proliferation parameters have been assumed for the quiescent cell population, how does mutation accumulation occur in quiescent cells and do quiescent cells immediately transition into proliferating cells after they acquired a resistant mutations?

Author Rebuttal, second revision:

Reviewer #1:

Remarks to the Author:

The authors have submitted a revised manuscript that is improved. This reviewer appreciates the thoughtful responses and additional experiments.

We are grateful to the referee for her/his appreciation of our work.

However, there are still important limitations in that the number of cell lines is essentially 2 (albeit using different clones), as far as this reviewer can tell and in vivo models are scarce in this study so there is little mitigation of potential cell line-specific biases and clinical heterogeneity.

Although we understand the reviewer's point, we would like to highlight the fact that the two CRC cell models that we have used have different genetic background, different population dynamic, and they are sensitive to different therapeutic regimen. They were chosen with the aim of reducing, to the best of our possibility, cell-line specific biases.

Additionally, there is no attempt to show some evidence of relevance using human clinical samples, even if in a small number of cases (although the authors points are understood). Finally, the therapeutic experiments using the REV inhibitor should be confirmed/validated and included in this study, as this is an interesting observation that could have translational relevance.

We agree with the referee on the potential translational relevance of the findings obtained with the REV1 inhibitor. Notably, we recently performed the TTP assay with REV1 inhibitor in combination with MAPK pathway inhibitors in a third MSS CRC model (JVE207, harboring a BRAF V600E mutation), the additional results confirm and extend our previous data (Rebuttal Figure 1). We had initially planned to present and extend the REV1 data in a separate manuscript. Following the referee's suggestion, we have instead included the exciting data using the REV1 inhibitor in WiDr, DiFi cells and the new cell line (JVE207) in the revised manuscript (Figure 5 and Extended Data Fig. 18).

Rebuttal Figure 1. Inhibition of error-prone DNA polymerase REV1 delays the onset of secondary resistance to MAPK pathway inhibition in CRC cells. MSS BRAF mutated JVE207 CRC cells were treated with the anti-EGFR inhibitor cetuximab (CTX) in combination with the anti-BRAF inhibitor dabrafenib (DAB), REV1 inhibitor or their combination and the number of cells was monitored during the treatment, until the emergence of resistance.

Reviewer #2:

Remarks to the Author:

In my review of the original version of this manuscript, I was doubtful of the clinical relevance of the findings, especially in light of a recent paper by Woolston et al. The authors have addressed this concern in their revision by adding a new paragraph to the Discussion, rephrasing some of their claims, and conducting new analyses. They have also made appropriate changes in response to my other questions and minor suggestions. Although the clinical relevance of the work remains somewhat uncertain, I'm satisfied that the authors have addressed my concerns and I have no further suggestions for improvement.

We thank the reviewer for the positive comments.

Reviewer #3:

Remarks to the Author:

This is a very interesting manuscript that investigates the response of colon cancer cell lines to targeted therapy. It combines multiple experimental approaches and mathematical and computational modelling.

I will restrict my comments on the theoretical and computational modelling and data interpretation but will not comment on details of experimental design or data generation.

I have no major concerns on the formal analysis of the mathematical and computational model. The equations and analytical approximations seem sound to me.

We thank the reviewer for the kind statements and for the positive evaluation of the mathematical and computational model.

I have a few questions on the interpretation of the data:

The authors explain the two slopes of the treatment response by a treatment induced non (or slow) proliferating cell population with higher treatment resistance, making analogies to similar dynamic patterns in bacteria. I agree with the author's that this is one possible interpretation. However, although I am not aware of such treatment responses in colon cancer patients, there are classic examples (most prominently the treatment response of chronic myeloid leukemia to Imatinib) that do show the same very typical two slope response to targeted therapy (see for example Michor and Nowak, Nature 2005). Classically, these dynamics are explained by a proliferation hierarchy, where slower stem like cells give rise to faster proliferating progenitor or differentiated cells. However, the transition rates in these

models are reversed. Slower proliferating cells at low frequency give rise to faster proliferating cells (see for example Werner, Scott et al. Cancer Research 2016 for examples of the dynamic patterns, but there are many other papers). In these models, treatment does not actively induce quiescence by a phenotypic switch, but simply selects on the pre-existing hierarchy.

We thank the reviewer for pointing our attention to this important thread of literature. These works support a scenario (in other systems) where there are phenotypic switches from slower-proliferating tolerant cells to faster-proliferating non-tolerant phenotypes, and, as the authors show, this scenario can also give rise to a biphasic killing curve. However, in our framework, this scenario would correspond to the case of $f_0 \gg 0$, which is actually ruled out by our analysis. Nevertheless, following the referee's comment, we have applied the Michor-Nowak 2005 (MN) model (presented here in rebuttal Annex 1 and Rebuttal Figure 2) to our framework. We found that, while this model can reproduce the biphasic killing curve of our data, it cannot reproduce the trend of the killing curve with the drug concentration.

Thus, while we fully agree (and now duly mention in the revised manuscript) that other interpretations of our results might be considered, we believe that our current interpretation on the nature of the phenotypic switch seems the most likely based on the data we obtained.

We have revised the text in order to properly discuss these studies.

Rebuttal Annex 1 – Analysis of the Michor-Nowak 2005 Model

The Michor-Nowak model posits the co-existence of a set of sub-populations of cancer cells: stem cells, progenitors, differentiated, and fully differentiated ones. Following the notation used in MN, we denote the number of cells in each sub-population as y_0, y_1, y_2 and y_3 , respectively. Each sub-population is characterized by death rates (denoted as d_0, d_1, d_2 and d_3 , respectively). Stem cells can generate progenitors with a rate a_y , progenitors can generate differentiated cells with rate b_y , and differentiated cells can generate fully differentiated cells with rate c_y .

During treatment, the differentiation rates a_y, b_y are modified by the action of the drug, which impedes the generation of progenitors and of differentiated cells. The reduction of the generation rates ($a'_y < a_y, b'_y < b_y$) leads the population to a new stationary state, and to a decline of the total number of cancer cells. The relaxation dynamics to the new state explain the biphasic curve. The first slope reflects the decay of differentiated cells, with an exponential decline $\sim e^{-d'_2 t}$, while the second slope reveals the decline of progenitors ($\sim e^{-d'_1 t}$). Death rates of both progenitors and differentiated cells during drug treatment (d'_1 and d'_2) can also be altered by the action of the drug ($d'_1 \geq d_1$ and $d'_2 \geq d_2$).

Although the data we collected to investigate the proliferation dynamics of untreated cancer cells (Extended data Figure 6) point to a homogeneous population, and do not seem to reveal the presence of sub-populations with differential proliferations rates as described in MN, we cannot exclude that sub-

population of slowly proliferating stem and progenitor cells could be present at very low frequencies. However, we think that the MN cannot explain the dynamics of WiDr clones under treatment in a straightforward way.

For both WiDr clones (cl. B5 and cl. B7) we experimentally observe that, during drug treatment, the number of surviving residual cells increases with increasing drug concentrations (cfr. Extended Data Figure 9 and 14). This effect is captured by the TP model with a linear dependence of the switch rate of sensitive to persister cells on the drug concentration, suggesting that higher drug doses could further impeded cell proliferation, favoring the transition to the quiescent state. Conversely, in the MN model, higher drug doses should generically lead to a reduction of generation rates (a'_y, b'_y) and to an increase of death rates (d'_1, d'_2), hence resulting in an additional decline of the number of residual cells with increasing drug concentrations. We illustrate this point in Rebuttal Figure 2.

Hence, although the MN can probably be generalized to capture this effect (e.g., adding further transitions), a natural extension of the existing model formulation, to include an explicit dependence on the drug concentration, would give a quantitative disagreement with the data we have collected.

Rebuttal Figure 2. The Michor-Nowak model can reproduce the biphasic curve of the TP model for a given drug concentration but cannot reproduce an increased number of residual cells described by the TP model for higher drug concentrations. A: Model parameters of the Michor-Nowak model were adjusted to reproduce the biphasic curve of the TP model inferred for WiDr cl. B7, with a drug concentration $[M]=1 \mu\text{M}$ (See Extended Data Table 4 for the numerical values of the TP model parameters). B: Growth curve of TP model, for WiDr cl. B7 and for a higher drug concentration $[M]=3 \mu\text{M}$, displays an increased number of residual cells compared to A, consistently with the dynamics observed in our experimental data (Extended Data Figure 5 and 9). Conversely, in the Michor-Nowak (MN) model, an increased drug concentration would imply a reduction of the rate constants a'_y and b'_y with respect to A, which results in a reduction of the number of residual cells. In A and B the numerical values of the death rates were set to match the two slopes of the corresponding biphasic curves of the TP model. MN growth curves of A and B share the initial condition of the population, i.e., the values $y_0(0), y_1(0), y_2(0), y_3(0)$ at $t=0$, since these model parameters are independent of the drug concentration. MN growth curves shows the value $(y_0(t)+y_1(t)+y_2(t)+y_3(t))/(y_0(0), y_1(0), y_2(0), y_3(0))$ vs t . Numerical values used of the model parameters of the Michor-Nowak model for A: $d_0=0, d_1=0.073 \text{ days}^{-1}, d_2=1.15 \text{ days}^{-1}, d_3=1.5 \text{ days}^{-1}, r_y=0.01, c'_y=80 \text{ days}^{-1}, y_0(0)=100, y_1(0)=10\,000, y_2(0)=80\,000, y_3(0)=29\,600\,000, a'_y=0.016 \text{ days}^{-1}, b'_y=8 \text{ days}^{-1}$, and B: $d_1=0.073 \text{ days}^{-1}, d_2=1.3 \text{ days}^{-1}, a'_y, b'_y$ as indicated in the legend, other model parameters as in A.

In the same line of arguments, a Poisson like distribution of quiescent (slow proliferating) phenotypes in a luria-delbrück like experiment itself does not necessarily show a phenotypic switch. If quiescent cells are pre-existing at a low frequency, a luria-delbrück experiment would imply a sampling process and we would predict a simple Binomial distribution of quiescent phenotypes after sampling. The expected parameter range of this distribution would be a small p and large n , which implies that the binomial distribution would be very close to a Poisson distribution.

We thank the referee for this remark, which prompted us to better clarifying this analysis. Accordingly, we have revised the methods section referring to this analysis. While we agree on the mathematical argument suggested by the referee, we believe that, because of the expansion of the cell population before the drug treatment administration, in our experimental scenario the expected distribution for the abundance of pre-existing cells across wells tested in this analysis would not be Binomial, but a Luria-Delbruck distribution.

What we have shown with numerical simulations (Extended data Figure 11) is that, provided that no persister cell pre-exists the treatment initiation, the number of persister cells emerging from sensitive cells during drug treatment would be Poisson distributed. This finding also implies that, if persister cells pre-exist the treatment and their abundances are also Poisson distributed, then the final distribution of the cell abundances (pre-existing + derived) would still be Poisson.

However, in our framework pre-existing persisters derive from a phenotypic switch occurred in sensitive cells before the treatment initiation (cell populations used in our experiments are generated from the expansion of single clones). Hence, pre-existing persisters are generated with a constant rate from an exponentially expanding population, a dynamic which would imply that the number of pre-existing cells is not Poisson distributed, but is described by a Luria-Delbruck distribution (with Variance \gg Mean). We find that the final distribution of persister cells is Poisson, suggesting that no persisters cells were present (hence generated) before the treatment.

Finally, in the Michor-Nowak scenario, persisters abundance would show the variability of the population size in a birth-death model, which is also likely wider than Poisson given the exponential nature of this process.

I agree with the authors that phenotypic switching is one possible explanation, but I do not believe the authors actually show that phenotypic switching occurs. It's a difficult task to unambiguously show phenotypic switching as driving force of resistance. However, I would suggest that the authors formulate

the results on that aspect a little more balanced and state that there are uncertainties and possible other explanations (see points above).

We agree with the reviewer, and we have included these aspects in the revised Discussion. Besides the point above on the possible alternative scenario for the biphasic curve, we have also added a comment to the Discussion where we explicitly state that we do not have a direct observation of the phenotypic switching (as a reliable marker of persisters phenotype is currently missing), and it would be useful to achieve this result in future studies. We point out that in the persisters literature also the Michor-Nowak mechanism would be interpreted as a phenotypic transition, so probably the point of the reviewer refers mostly to the scenario for this transition.

The authors need to provide a bit more detail and explanation on the stochastic modelling approach.

What proliferation parameters have been assumed for the quiescent cell population, how does mutation accumulation occur in quiescent cells do quiescent cells immediately transition into proliferating cells after they acquired a resistant mutation?

We thank the referee for pointing out this lack of information. In the simulations of the stochastic modeling, persister cells had a birth rate $b=0$ and a death rate $d=D_p$, where D_p is the effective growth rate estimated from growth assays (see Extended Data Table 4). It should be noted that the form of the estimator used to infer the mutation rate only depends on the difference $b-d=-D_p$, hence, to test the validity of our methods, the actual numerical value of b actually did not matter. Persister cells turn into resistant cells with a chronological rate μ_p (per cell, per unit of time). Once a persister cell has turned into a resistant one, it is assumed it immediately transitions into a normal cell, and its dynamics are described by proliferation rates of sensitive cells.

Decision Letter, third revision:

Our ref: NG-A57462R2

1st Mar 2022

Dear Dr. Bardelli,

First of all, I apologise for the delay in returning this decision to you. Thank you for your patience.

Thank you for submitting your revised manuscript "A modified fluctuation-test framework characterizes population dynamics and mutation rate of cancer persister cells" (NG-A57462R2). It has now been seen by Reviewer #3, and their comments are below. The reviewer finds that the paper has improved in revision, and therefore we'll be happy in principle to publish it in Nature Genetics, pending

minor revisions to comply with our editorial and formatting guidelines.

Sincerely,

Safia Danovi
Editor
Nature Genetics

Reviewer #3 (Remarks to the Author):

I want to thank the authors for their thoughtful rebuttal. I overall agree with the response. Especially the experimental observation that WiDr clones increase in number with high drug concentration is interesting.

I share the concern of reviewer 1 that these are results in only 2 cell lines and there is a lack of evidence for clinical relevance of the results.

However, the lack of a bi-phasic response to targeted treatment in CRC may be, because targeted treatments are administered at a late stage of tumour evolution, and relapse is dominated by pre-existing (genetic) resistance. Our current believe would be that efficacy of targeted therapy would increase if administered sufficiently early. However, if induced persister cells exist, this believe is wrong.

Although personally, I would need further evidence to truly believe in induced resistance, I think the experiments and results are important enough for publication.

Final Decision Letter:

In reply please quote: NG-A57462R3 Bardelli

25th May 2022

Dear Dr. Bardelli,

I am delighted to say that your manuscript "A modified fluctuation-test framework characterizes the population dynamics and mutation rate of colorectal cancer persister cells" has been accepted for publication in an upcoming issue of Nature Genetics.

Your paper will be published online after we receive your corrections and will appear in print in the next available issue. You can find out your date of online publication by contacting the Nature Press Office (press@nature.com) after sending your e-proof corrections. Now is the time to inform your Public Relations or Press Office about your paper, as they might be interested in promoting its publication. This will allow them time to prepare an accurate and satisfactory press release. Include your manuscript tracking number (NG-A57462R3) and the name of the journal, which they will need when they contact our Press Office.

Please note that *Nature Genetics* is a Transformative Journal (TJ). Authors may publish their research with us through the traditional subscription access route or make their paper immediately open access through payment of an article-processing charge (APC). Authors will not be required to make a final decision about access to their article until it has been accepted. [Find out more about Transformative Journals](https://www.springernature.com/gp/open-research/transformative-journals)

Authors may need to take specific actions to achieve [a](https://www.springernature.com/gp/open-research/funding/policy-compliance-)

faqs"> compliance with funder and institutional open access mandates. If your research is supported by a funder that requires immediate open access (e.g. according to Plan S principles) then you should select the gold OA route, and we will direct you to the compliant route where possible. For authors selecting the subscription publication route, the journal's standard licensing terms will need to be accepted, including https://www.nature.com/nature-portfolio/editorial-policies/self-archiving-and-license-to-publish. Those licensing terms will supersede any other terms that the author or any third party may assert apply to any version of the manuscript.

Please note that Nature Portfolio offers an immediate open access option only for papers that were first submitted after 1 January, 2021.

An online order form for reprints of your paper is available at https://www.nature.com/reprints/author-reprints.html. Please let your coauthors and your institutions' public affairs office know that they are also welcome to order reprints by this method.

If you have not already done so, we invite you to upload the step-by-step protocols used in this manuscript to the Protocols Exchange, part of our on-line web resource, natureprotocols.com. If you complete the upload by the time you receive your manuscript proofs, we can insert links in your article that lead directly to the protocol details. Your protocol will be made freely available upon publication of your paper. By participating in natureprotocols.com, you are enabling researchers to more readily reproduce or adapt the methodology you use. [Natureprotocols.com](https://natureprotocols.com) is fully searchable, providing your protocols and paper with increased utility and visibility. Please submit your protocol to <https://protocolexchange.researchsquare.com/>. After entering your [nature.com](https://www.nature.com) username and password you will need to enter your manuscript number (NG-A57462R3). Further information can be found at <https://www.nature.com/nature-portfolio/editorial-policies/reporting-standards#protocols>

Sincerely,

Safia Danovi
Editor
Nature Genetics